# Global covariation of forest age transitions with the net carbon balance

Simon Besnard [1] ✉, Viola H. A. Heinrich [1,2], Nuno Carvalhais [3,4,5], Philippe Ciais [6], Martin Herold [1], Ingrid Luijkx [7], Wouter Peters [7,8], Daniela Requena Suarez[1], Maurizio Santoro [9] & Hui Yang [3,10]

Forest age transitions are critical in shaping the global carbon balance, yet their influence on carbon stocks and fluxes remains poorly quantified. Here we analyse global forest age dynamics from 2010 to 2020 using the Global Age Mapping Integration v2.0 dataset, alongside satellite-derived aboveground carbon (AGC) and atmospheric inversion-derived net $CO_2$ flux data. We reveal widespread declines in forest age across the Amazon, Congo Basin, Southeast Asia and parts of Siberia, primarily driven by stand-replacing disturbances such as fire and harvest, leading to the replacement of older forests by younger stands. Meanwhile, forests in China, Europe and North America experienced net ageing. Globally, stand replacement resulted in substantial AGC losses, with old forests (>200 years, ~98.0 MgC ha$^{-1}$) transitioning to younger, carbon-poor stands (<20 years, ~43.5 MgC ha$^{-1}$), leading to a net AGC loss of ~0.14 PgC per year. Despite this, regions with high rates of young stands replacing old forests exhibited a temporary strengthening of the carbon sink, driven by the rapid regrowth of these young stands. Crucially, these young forests do not compensate for the long-term carbon storage of old forests. Our findings underscore the importance of protecting old forests while optimizing forest management strategies to maximize carbon gains and enhance climate mitigation.

Key international initiatives, such as the Forest and Land Use Declaration at the 26th United Nations Climate Change Conference of the Parties (COP26)[1], the EU's Nature Restoration Law[2], and the Regulation on Deforestation-Free Products[3], highlight the critical role of forests in global climate mitigation. In this study, forests encompass natural forests, managed stands and planted tree cover (for example, agroforestry, tree crop and plantations), characterized by at least 5 metres height[4]. Forests are essential to Earth's carbon cycle, acting as carbon sinks by absorbing about −3.5 ± 0.4 (where − denotes a net carbon gain) petagrams of carbon annually (PgC per year) in the 2010s[5]. Effectively

managing and conserving these forests is crucial for mitigating climate change, especially given the decline in forest resilience under climate change[6].

Several processes, including deforestation, degradation, afforestation, natural disturbances, management practices and regrowth, influence the distribution of forest ages worldwide[7]. These factors, in turn, affect the forests' capacity to sequester carbon[8]. European forests, for instance, have consistently acted as carbon sinks[9] due to their recovery from extensive clear-cutting after World War II[10]. Their growth exceeds harvest removals and natural mortality[9,11], although

[1]GFZ Helmholtz Centre for Geosciences, Potsdam, Germany. [2]School of Geographical Sciences, University of Bristol, Bristol, UK. [3]Max Planck Institute for Biogeochemistry, Jena, Germany. [4]ELLIS Unit Jena, Jena, Germany. [5]Departamento de Ciências e Engenharia do Ambiente, DCEA, Faculdade de Ciências e Tecnologia, FCT, Universidade Nova de Lisboa, Caparica, Portugal. [6]Laboratoire des Sciences du Climat et de l'Environnement, CEA-CNRS-UVSQ-UPSACLAY, Gif sur Yvette, France. [7]Meteorology and Air Quality Department, Wageningen University, Wageningen, The Netherlands. [8]Centre for Isotope Research, University of Groningen, Groningen, The Netherlands. [9]Gamma Remote Sensing, Gümligen, Switzerland. [10]College of Urban and Environmental Sciences, Peking University, Beijing, China. ✉e-mail: simon.besnard@gfz.de

**a**

Average forest age 2010–2020

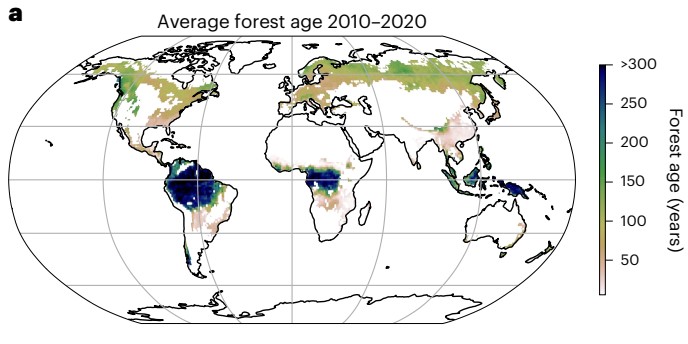

**b**

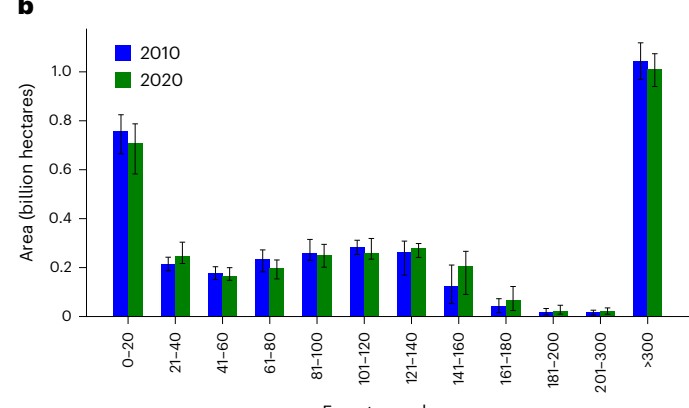

**c**

Anomaly difference map
relative to the expected 10-year ageing

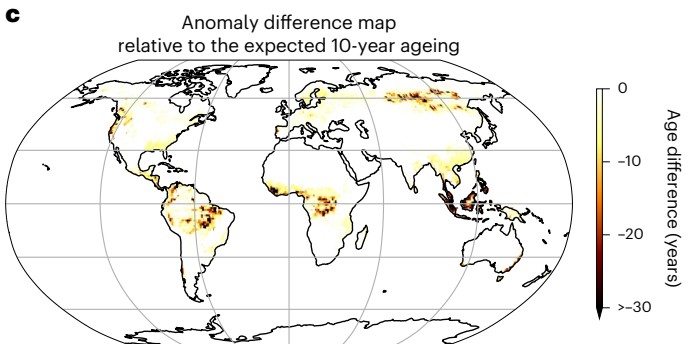

**Fig. 1 | Global patterns and shifts in forest age between 2010 and 2020.**
**a,b**, Average forest age for 2010 and 2020 at a one-degree pixel resolution
(**a**) and the total area of each forest age class for 2010 (blue) and 2020 (green)
(**b**). The total area represents the median values across the 20 ensemble
members, with error bars in **b** indicating the 5% and 95% quantiles of the total area
across the ensemble. **c**, The forest age difference map between 2020 and 2010,
relative to the expected 10-year ageing. This map was calculated by subtracting
ten years from the difference between 2020 and 2010. A value of zero indicates

that forests have aged as expected over the decade, consistent with the gradual
maturation of undisturbed forests. Positive values indicate slower-than-expected
ageing or ageing stagnation, potentially due to disturbances. By contrast,
negative values reflect forests becoming younger, probably due to stand
replacement, disturbances or shifts in forest composition favouring younger
stands. In **a** and **c**, the median estimates across the 20 ensemble members are
shown. One-degree grid cells with less than 20% forest cover have been masked.

the European carbon sink has shown a recent decline[12]. In regions
such as the Amazon Basin, old forests were once 'hotspots' of younger
forests, shaped substantially by ancient civilization activities[13]. Recent
regional transformations have created a diverse mosaic of forest age
classes (Fig. 1a,b). Complex interactions between stand-replacement
events, regrowth following disturbances, afforestation and the natural
ageing of established forests drive shifts in forest age distributions.
These factors collectively contribute to ecological transformations in
forest landscapes. Yet the implications of these changes on the global
carbon cycle remain unresolved[14].

Traditional methods, such as national forest inventories and
research plot measurements, offer valuable insights into local for-
ests but have limited spatial and temporal coverage. Several factors
constrain their effectiveness, including the absence, complexity or
limited availability of sample-based inventories in many regions, the
infrequency of repeated measurements and the variability in what
they measure. These limitations hinder their ability to comprehen-
sively capture global forest dynamics. Atmospheric $CO_2$ flux estimates
from inversion techniques offer an integrated view of biosphere–
atmosphere carbon exchange, considering various sources and sinks
across different spatial and temporal scales. Yet, their coarse spatial
resolution, typically around 110 km, limits their ability to distinguish
between forests and non-forests or accurately capture the different
dynamics of forests. Additionally, uncertainty in atmospheric trans-
port models and the lack of measurement stations in certain regions
limit their regional accuracy[15]. Efforts to map forest age at global and
regional levels have been made[16–18], but these typically result in static
snapshots of forest age distribution. A few satellite-derived maps

monitor temporal changes in forest age, but these are limited in space
and can only monitor forests since the beginning of the Landsat era[19,20].
This limitation impedes our ability to effectively track and understand
temporal changes in forest age across all age classes.

Using the Global Age Mapping Integration v2.0 (GAMIv2.0)
dataset[16,21] (Methods), our study analyses changes in forest age distribu-
tion from 2010 to 2020 and their implications for the carbon cycle. We
integrate forest age maps with independent atmospheric $CO_2$ inversion
flux data from the Global Carbon Project[22] and European Space Agency
Climate Change Initiative (ESA-CCI) satellite-based biomass maps[23].
Our analysis examines how shifts in forest age, caused by rejuvenation
from natural disturbances and clear-cutting, and the natural ageing
of established forests affect the net carbon balance. The study has
three main objectives: (1) to map global changes in forest age distri-
bution over the last decade, (2) to infer the impact of replacing older,
carbon-rich forests with younger ones on the net carbon balance, and
(3) to hypothesize the effects of these shifts on aboveground carbon
(AGC) stocks under a (1) business-as-usual (BAU) forest management
and (2) forest conservation scenario. Our findings highlight the inter-
play between forest age dynamics and the global carbon cycle, empha-
sizing the need for strategic, targeted forest management and informed
policies to optimize the role of forests in climate change mitigation.

## Results and discussion
### Changes in forest age distribution for 2010–2020
We analysed forest age distribution shifts from 2010 to 2020 using the
GAMIv2.0 dataset (Fig. 1a). We focused on this period due to the avail-
ability of ESA-CCI biomass data (2010, 2017–2020) and the rising

**Table 1 | Forest age classification and characteristics**

| Forest age class | Characteristics |
| --- | --- |
| Young forests (1–20 years) | Rapid post-disturbance growth, high net primary productivity[63,64], dominated by early-successional species and high canopy openness. |
| Maturing forests (21–80 years) | Transitioning towards structural stability, moderate carbon accumulation and increasing canopy closure. |
| Mature forests (81–200 years) | Substantial carbon storage, slower growth rates, structurally complex canopy, increased biodiversity. |
| Old forests (>200 years) | Mainly in intact tropical rainforests and inaccessible boreal regions represent complex ecosystems that may be sources or sinks of carbon[12,51] |

pressure on forests over the past decade[24,25]. Age classes were defined based on ecological stages (Table 1). From 2010 to 2020, global forest age shifts (Fig. 1b) were driven by stand replacement and the gradual ageing of established forests (definition in Table 2). Mature (81–200 years) forests increased by $+0.08^{+0.11}_{+0.05}$ billion hectares ($+8.0^{+12.6}_{+4.4}$%), whereas young (1–20 years), maturing (21–80 years) and old forests (>200 years) declined by $-0.04^{-0.01}_{-0.08}$ billion hectares ($-5.2^{-1.5}_{-12.0}$%), $-0.02^{+0.05}_{-0.05}$ billion hectares ($-2.7^{+7.4}_{-7.4}$%) and $-0.03^{-0.02}_{-0.04}$ billion hectares ($-2.6^{-2.1}_{-3.3}$%), respectively (Extended Data Table 1). Regional variations are noteworthy, though (Extended Data Fig. 1). For instance, the total area of young forests increased in Eurasian boreal from $0.039^{0.042}_{0.038}$ billion hectares in 2010 to $0.058^{0.059}_{0.058}$ billion hectares in 2020 (Extended Data Fig. 1a). By contrast, young forests remained relatively stable in European forests, with approximately $0.036^{0.042}_{0.032}$ billion hectares in 2010 and $0.037^{0.041}_{0.035}$ billion hectares in 2020 (Extended Data Fig. 1d). This stability suggests a balance between young forests maturing into older age classes and stand-replacement processes despite ongoing disturbances[26]. GAMIv2.0 estimates this relationship using a nonlinear model that considers the effects of local biogeographic and climatic conditions, enabling the representation of non-monotonic biomass accumulation across age classes (Extended Data Fig. 2). However, our analysis does not fully capture the impact of forest management practices on the forest age–biomass relationship. Consequently, shifts between age classes may be biased, especially in regions with substantial management influence on forest dynamics (Supplementary Fig. 1). To address this, we binned the age classes by 20 years, which smooths out variability and reduces sensitivity to errors in age estimation. Whereas this approach captures broader trends and minimizes noise from management-induced variations, it may mask the finer-scale impacts of different management practices. Therefore, more comprehensive forest management data are crucial to enhancing the accuracy and reliability of these models.

The reduction in young forests is probably due to their progression to older stages, outpacing the development of young forests through frequent forest clearings and afforestation. Additionally, the saturation of afforestation programmes in countries such as China[27], initiated in the 1980s and 1990s, and the possible exhaustion of old forests suitable for cutting in Southeast Asia due to increased protection efforts[28] may have contributed to this reduction. We observe an increase in the global area of forests aged between 21 and 40 years by approximately +0.03 billion hectares (+17%) over 2010–2020 (Extended Data Table 1). The increase could be a response to prior variations in natural or human-induced disturbances, such as clear-cutting, leading initially to an increase in this age class. These dynamics indicate a more complex forest regrowth pattern[29], where young forests mature and recover from earlier disturbances. Multiple factors, including the natural ageing of younger forests, enhanced conservation efforts and a shift towards sustainable practices such as thinning, probably drive the increase in mature forests. Additionally, land abandonment and migration to urban areas may indirectly contribute to this trend. The decline of old forests is alarming because it suggests the loss of

the oldest and most ecologically valuable forest stands, vital for their substantial carbon storage capabilities and rich biodiversity. These forests not only sequester large amounts of carbon but also enhance the recovery and growth of nearby degraded forests through processes such as seed dispersal and the creation of stable microclimates[20]. This underscores the critical importance of preserving old forests.

Beyond global changes in forest age distribution (Fig. 1b), we identified contrasting local and regional forest age transitions (Fig. 1c, Supplementary Fig. 2 and Supplementary Table 1). In regions such as Amazonia, the Congo Basin and Southeast Asia, marked decreases in forest age were evident, with variations at the one-degree pixel level showing up to 30% changes since 2010. Specifically, forests in South America and tropical Asia experienced net age decreases of $-4.7^{-3.4}_{-6.2}$ years (mean age of $251.9^{260.2}_{237.2}$ years in 2010, shifting to $247.1^{254.3}_{233.9}$ years in 2020) and $-7.3^{-5.0}_{-10.0}$ years (mean age of $139.4^{152.5}_{121.2}$ years in 2010, shifting to $132.4^{142.3}_{116.4}$ years in 2020), respectively (Supplementary Table 1). This trend is primarily attributed to increasing stand-replacing disturbances and mortality[30,31], indicating a shift towards younger forest stands and the replacement of old forests. Traditional activities such as slash-and-burn agriculture also contributed to this trend in the Amazon Basin[32]. Whereas young forests grow faster[27], their regrowth does not fully compensate for the carbon loss from older forests[28]. Transitions to younger stands often involve associated emissions fluxes—from direct emissions due to fire to delayed emissions resulting from increased substrate availability following the harvest of litter. In North America, especially in the Pacific Northwest, a mosaic of older stands and areas of stand replacement followed by regrowth is evident, probably influenced by clear-cutting, a high-burning frequency regime and other natural disturbances, such as insect outbreaks[33]. Forests in the North American boreal region experienced a net age increase of $+8.9^{+9.0}_{+8.8}$ years (mean age of $101.6^{111.9}_{85.7}$ years in 2010, shifting to $110.5^{120.7}_{94.6}$ years in 2020). The age increase is only marginally lower than the expected net ten-year increase if no disturbances have occurred. It suggests that disturbances were relatively limited in this region, allowing the forests to age naturally. We also observed that local patches of Canadian forests were getting younger, probably related to recent fire activity[34]. These localized disturbances indicate that while the overall impact on forest ageing was minimal, specific areas experienced substantial shifts in forest age distribution. Siberian forests, suspected to harbour an important fraction of the global carbon sink[12], predominantly sustained their older age class (that is, forests in Eurasian boreal regions showed a mean age of $108.0^{116.5}_{89.0}$ years in 2010, shifting to a mean age of $113.6^{122.0}_{95.1}$ years in 2020). However, substantial transitions towards younger forests (approximately 0.09 billion hectares, 7.2% of the Eurasia boreal region) indicate localized impacts from logging, increased wildfire events[35] and tree recruitment failure after fire[36].

In China, extensive reforestation and afforestation programs since the 1980s[37] have led to substantial shifts in the forest age, with younger planted forests maturing. As part of these large-scale afforestation efforts, the forest area in southern China increased from 9% to 35% between 1986 and 2018, reflecting a marked expansion in forest cover following the implementation of these policies[37]. These efforts have contributed to the overall net ageing of forests in Eurasia temperate by $+7.7^{+8.2}_{+6.4}$ years, from a mean age of $44.5^{53.2}_{36.5}$ years in 2010 to $52.5^{60.4}_{42.9}$ years in 2020 (Supplementary Table 1). In southwestern Australia, frequent wildfires have led to younger age classes, probably due to increased turnover or the severity of recent fire events[34]. Overall, forests in Australia aged by $+2.1^{+3.4}_{-0.3}$ years (a mean age of $67.7^{88.0}_{55.8}$ years in 2010, shifting to a mean age of $69.9^{87.7}_{57.9}$ years in 2020). This suggests that while fire events have caused localized rejuvenation, other areas have aged relatively undisturbed. Conversely, the Miombo woodlands have maintained consistent fire regimes[34], resulting in relatively unchanged turnover and gross primary productivity[38]. This stability indicates an adaptation to the prevailing fire regime, which preserves the understory and maintains the overall forest structure. In South

**Table 2 | Description of the stand-replaced and undisturbed ageing forests' processes derived from 100-m resolution data**

| | Stand-replaced forests | Undisturbed ageing forests |
|---|---|---|
| Process description | Stand-replaced forests emerge from substantial disturbances, resetting forest age and undergoing regrowth. Following disturbance, structural growth (biomass accumulation, canopy development) can be rapid in young forests. | Undisturbed ageing forests include forests that have not experienced stand-replacing disturbances but may be subject to stable management practices such as thinning. These practices do not involve resetting the stand age. |
| Forest age transition | Forests experience an age difference of less than ten years derived from 100-m resolution data, indicating stand-replacement followed by regrowth between 2010 and 2020. Before disturbance, these forests could have been young, maturing, mature or old, but after the stand-replacement event, they were reset to young stands and began regrowth. | For undisturbed forests between 2010 and 2020, a uniform ten-year age increase derived from 100-m resolution data is assumed. This category includes naturally ageing forests planted before 2010 and forests regenerating from pre-2010 disturbances. |

America, forests outside the Amazonia, such as the Atlantic forests, have experienced a decrease in native forest cover due to human activities and disturbances. However, recent conservation efforts have led to some recovery[39]. This combination of factors has resulted in a slight net age increase in the temperate forests of South America by $+3.9^{+6.3}_{+1.2}$ years (mean age of $87.4^{110.1}_{51.6}$ years in 2010, shifting to a mean age of $91.0^{110.5}_{57.7}$ years in 2020).

European forests, covering around 33% of the continent, are experiencing complex age dynamics due to management and natural disturbances, such as fire, drought and insect outbreaks since 2010[40,41]. Despite these disturbances, forests are generally ageing (mean age of $81.5^{91.2}_{72.1}$ years in 2010, shifting to a mean age of $89.5^{98.9}_{80.3}$ years in 2020, that is, $+8.0^{+8.2}_{+7.7}$ years), except in areas such as Portugal, where large fires have led to younger forests[34]. Climate-induced disturbances and subsequent salvage logging substantially impact northern and central Europe's boreal coniferous and dry forests in the Iberian Peninsula[42]. Consistent management and stable harvest rates[43] suggest that drastic forest age shifts due to human management are unlikely. However, intensified disturbances or increased harvesting could alter the future age structure of European forests, potentially amplifying the reported saturation of the European forest carbon sink[44]. The net ageing trend in Eurasian boreal forests (that is, forests aged by $+5.7^{+6.2}_{+5.4}$ years), especially in western Russia, is mainly due to large-scale forest abandonment since 1990, a scale surpassing the European forest area[45].

## Patterns of undisturbed ageing forests and forests replaced by young stands

Our analysis of global forest age shifts reveals distinct regional patterns, with a mix of undisturbed ageing (Fig. 2a) ($3.21^{3.46}_{2.84}$ billion hectares) and stand-replaced forests (Fig. 2b) ($0.23^{0.31}_{0.16}$ billion hectares) (Table 2 provides definitions, and Supplementary Table 3 and Supplementary Table 4 provide a summary of statistics). This variation across regions underscores the complex interplay (Supplementary Fig. 2) of natural processes and human influences (Fig. 2d) in shaping the age distributions of forests. Regionally, stand-replaced forests cover the largest absolute areas in tropical Asia ($0.26^{0.28}_{0.22}$ of total stand-replaced forest fraction, $0.059^{0.070}_{0.040}$ billion hectares), northern Africa ($0.22^{0.25}_{0.11}$ of total stand-replaced forest fraction, $0.048^{0.067}_{0.018}$ billion hectares) and South America tropical ($0.11^{0.13}_{0.10}$ of total stand-replaced forest fraction, $0.026^{0.030}_{0.020}$ billion hectares) (Supplementary Table 3). By contrast, undisturbed ageing forests are primarily located in South America tropical ($0.20^{0.21}_{0.20}$ of total undisturbed ageing forest fraction, $0.64^{0.64}_{0.63}$ billion hectares), Eurasia boreal ($0.17^{0.17}_{0.17}$ of total undisturbed ageing forest fraction, $0.54^{0.54}_{0.54}$ billion hectares), Europe ($0.10^{0.10}_{0.10}$ of total undisturbed ageing forest fraction, $0.32^{0.32}_{0.32}$ billion hectares) and North America temperate ($0.092^{0.092}_{0.090}$ of total undisturbed ageing forest fraction, $0.29^{0.29}_{0.28}$ billion hectares) (Supplementary Table 3), primarily characterized by a mix of forests often subject to management (for example, selective harvesting) and unmanaged forests (Fig. 1d). Due to its vast forest extent, South America tropical contains substantial absolute areas of both stand-replaced and undisturbed ageing forests, making both classes particularly prevalent in the region.

Boreal forests typically have an average pre-stand-replacement age of approximately 120 years ($123.75^{132.48}_{99.51}$ years and $113.05^{122.79}_{96.00}$ years for Eurasia boreal and North American boreal, respectively) (Fig. 2c). By contrast, forests in the Northern Hemisphere's temperate regions are replaced at younger ages. Management in these temperate regions, which usually involves naturally regenerated and planted forests, results in an average replacement age of $94.50^{102.87}_{78.94}$ years in Europe and $95.74^{101.63}_{80.64}$ years in North America (Fig. 2c). This variability is influenced by regional rotation lengths, with areas such as northern Spain, Portugal (that is, fast-growing species, such as Eucalyptus, feed into fast rotation forestry projects), Les Landes, Sweden and Finland having shorter rotation periods (Extended Data Fig. 3a,b). By contrast, other forests are harvested at an age over 100 years (Extended Data Fig. 3c). We observed short regeneration cycles in Eurasia's temperate ($33.55^{39.58}_{21.84}$ years) and South American temperate ($58.07^{75.47}_{38.46}$ years) forests. This forest age pre-stand replacement is probably influenced by frequent fires[34] and plantation forestry in these regions (Fig. 2d), where the typical rotation period is up to 15 years (ref. 46). In the South American tropics, stand replacement often occurs in older forests, averaging $190.71^{215.73}_{166.29}$ years, including old forests over 300 years old, usually characterized by continuous recruitment, though with considerable local variation. The age variability before stand-replacement in South America's tropical region suggests a mix of deforested old forests and young secondary forests harvested and burned before maturity[20] or part of a slash-and-burn agricultural system[32]. The scenario is more complex in northern ($91.97^{106.22}_{78.28}$ years) and southern ($78.92^{97.57}_{65.99}$ years) Africa regions, which experienced rapid replacement of young forests. These regions experience regular disturbances, primarily from fires[34], which can rapidly replace young forests. However, these fires do not always lead to complete stand replacement. Instead, they often leave some trees unburned, especially in lower-intensity or sub-canopy fires. Additionally, in some forest ecosystems, the lower tree density per hectare in certain areas[47] may limit the fire severity, reducing the likelihood of complete stand replacement. This highlights post-fire regeneration processes that differ from those of stand-replacing fires, allowing some trees to survive and creating a mix of age structures within the forests.

We observe a distinctive spatial pattern in the distribution of stand-replaced forests across various age classes as of 2010 (Fig. 3a, Extended Data Fig. 3 and Supplementary Table 3). Forests that were already young (that is 0–20 years) before stand replacement—covering $0.14^{0.20}_{0.083}$ billion hectares and accounting for $58.97^{66.55}_{46.14}$% of all stand-replaced forests— are broadly distributed but primarily concentrated in tropical regions (for example, South America's tropics, tropical Asia and northern Africa) and the Eurasian temperate zone (Fig. 3a). This widespread area of young forests being replaced by new stands suggests a high turnover rate, probably driven by plantations (Fig. 2d) and natural disturbances, such as fire events. Maturing forests (that is 20–80 years) replaced by younger stands constitute $12.51^{15.96}_{9.97}$% of the total stand-replaced forest areas. Mature forests (that is 81–200 years) that have undergone stand replacement, where younger stands have replaced previously mature forests, account for $14.45^{18.86}_{10.58}$% of the total stand-replaced forest area. Their distribution (Extended Data Fig. 3c)

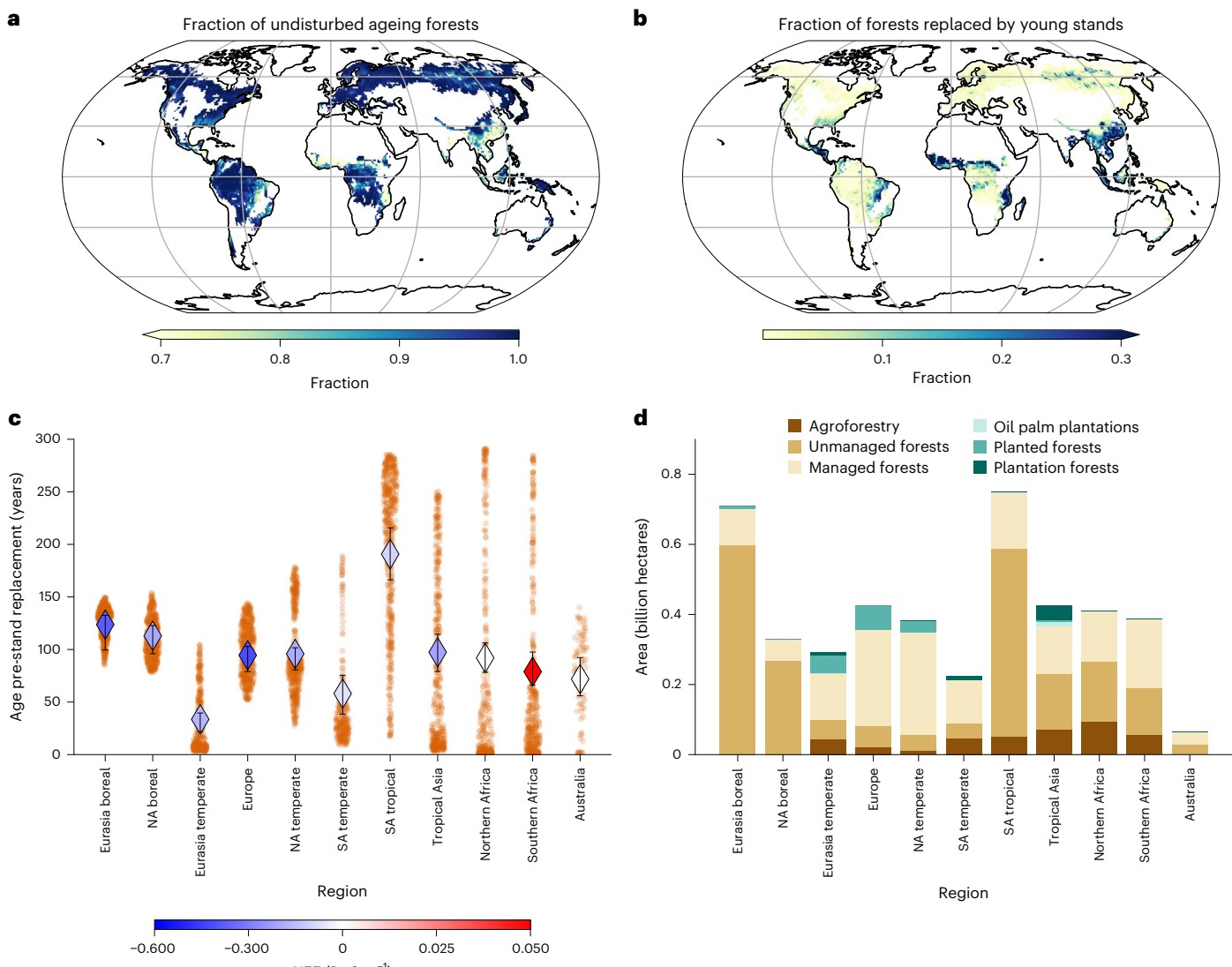

**Fig. 2 | Spatial patterns of forest ageing and stand-replacing disturbances.**
**a,b**, Spatial distribution of undisturbed ageing (**a**) and stand-replaced forest fraction (**b**) at one-degree pixel size. **c,d**, The forest age pre-stand-replacement (**c**) and the total area of forest management types[46] (**d**) are also shown across the 11 TRANSCOM-land regions (Supplementary Fig. 3). Unmanaged/intact forests are regenerating forests without any signs of management (including primary forests). Naturally regenerated forests are forests that have regenerated without signs of management (for example, logging, clear-cuts). Plantation forests are considered to have a rotation time of up to 15 years. In **a–c**, the median estimates of the 20 members are shown. The error bars in **c** represent the 5th and 95th percentiles of the forest age pre-stand-replacement quantiles across the 20 members. One-degree grid cells with less than 20% forest cover have been masked. NA, North American; SA, South America.

is sparser than that of younger forests, with large but dispersed patches in regions such as North American, European and Siberian forests. This pattern may indicate less frequent replacement events in mature forests, suggesting a period of relative stability or reduced human influence (Fig. 2d). Old forests (that is >200 years) were the least likely to undergo stand replacement, representing only $14.38^{19.92}_{11.78}$% of the total and are primarily located in tropical regions (Extended Data Fig. 3d and Supplementary Fig. 6a). By contrast, forests undisturbed during the study period display a more balanced distribution across different age classes: young ($19.01^{20.59}_{16.56}$%), maturing ($18.71^{20.76}_{16.22}$%), mature ($30.80^{32.98}_{26.30}$%) and old forests ($32.23^{34.41}_{30.06}$%) (Extended Data Fig. 4 and Fig. 3b).

**Influence of forests replaced by younger stands on the global carbon sink capacity**

Using the ESA-CCI biomass map for 2020[23], we observe apparent differences in AGC stocks where younger stands replaced carbon-rich older forests from 2010 to 2020 (Fig. 4c and Extended Data Fig. 5). After

replacement, the AGC stock of stand-replaced forests varies by prior age class: -2.3 MgC ha⁻¹ in young forests, increasing to 15.2 MgC ha⁻¹ in maturing forests, 31.1 MgC ha⁻¹ in mature and peaking at 31.9 MgC ha⁻¹ in old forests. These differences reflect two main factors. First, not all biomass is lost during stand-replacement—legacy trees and residual carbon often remain, especially in partially disturbed areas. Second, younger stands replacing old forests usually grow faster due to nutrient-rich soils, existing seed banks and structural legacies, with open canopies enhancing light availability and promoting rapid regrowth[48] (Extended Data Fig. 6).

Undisturbed ageing forests store substantially more carbon than recently disturbed ones across all age classes (Fig. 4c and Extended Data Fig. 7): -23.8 MgC ha⁻¹ in young forests, 29.5 MgC ha⁻¹ in maturing forests, 50.5 MgC ha⁻¹ in mature forests and 77.8 MgC ha⁻¹ in old forests. The comparable carbon storage capacity in stand-replaced and undisturbed mature forests suggests that stand replacement often involves partial rather than complete biomass removal through selective logging or natural disturbance. Whereas the GAMIv2.0 product does not

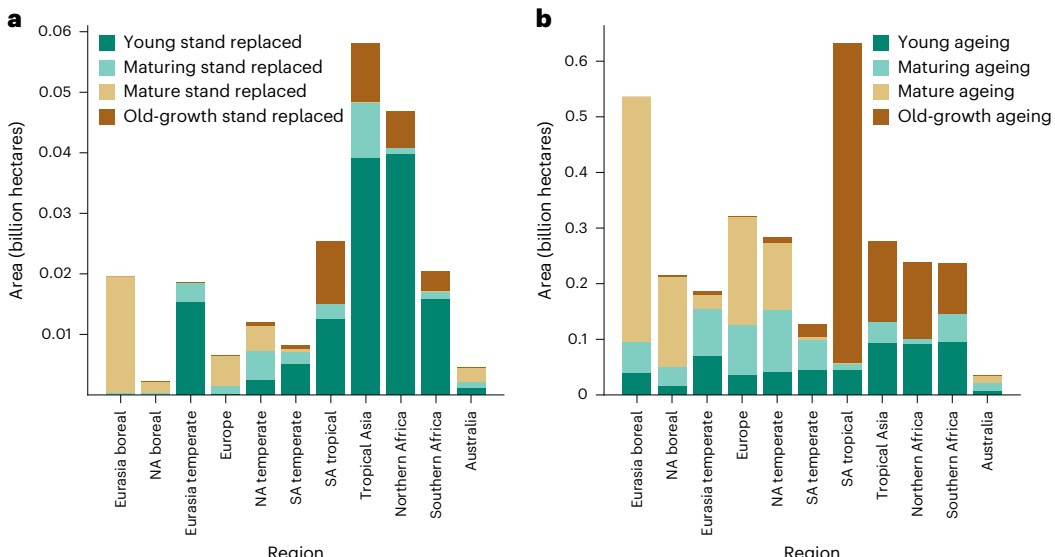

**Fig. 3 | Area of forest dynamics by age class across regions. a,b**, Total area (in billion hectares) of stand-replaced (**a**) and undisturbed ageing forests (**b**) per age class across the 11 TRANSCOM-land regions (Supplementary Fig. 3).

capture small-scale degradation or mortality effects, the ESA-CCI biomass map partially reflects these dynamics. This may explain the limited AGC gap observed in mature forests across disturbance types. However, ESA-CCI biomass data also have limitations, particularly in the calibration of AGC estimates from remote sensing observations, as the sensitivity of satellite-derived AGC retrievals decreases with increasing biomass, contributing to uncertainty in high-biomass forests[49].

Evaluating total carbon stocks across forest age classes (Supplementary Table 5) reveals that old, undisturbed forests store $127.39^{155.14}_{99.20}$ PgC. Conversely, young forests, including those that replaced older stands ($1.85^{3.17}_{1.35}$ PgC) and undisturbed ageing forests ($17.99^{21.32}_{16.02}$ PgC), collectively stored approximately 20 PgC in 2020 (approximately 9% of the total AGC stocks in 2020). Maturing and mature, undisturbed ageing forests cover large areas (Fig. 3b), substantially contributing to total forest AGC stocks ($31.29^{35.68}_{26.40}$ PgC and $60.51^{79.34}_{47.32}$ PgC, respectively). The total global AGC stock in 2020 was estimated at ~$235.75^{286.11}_{192.50}$ PgC in 2020, aligning with independent datasets but substantially lower than previous estimates[5] (~371.5 PgC), probably due to differences in disturbance representation, belowground biomass inclusion and forest definition (for example, a 20% forest fraction filter was applied in our study).

The transition from carbon-rich old forests to younger forests, resulting from stand replacement, substantially reduces the overall carbon storage of these ecosystems (Fig. 4b) and ecosystem function[50]. Between 2010 and 2020, forests undergoing stand replacement lost $+0.38^{+0.46}_{+0.35}$ PgC yr$^{-1}$ (where + denotes loss of AGC) (Supplementary Table 5 and Supplementary Fig. 4). Despite covering only a small fraction of the global forest area (Fig. 3a and Extended Data Fig. 3), these forests disproportionately influenced AGC losses, particularly where old forests were replaced by young stands, contributing to 0.10–0.16 PgC yr$^{-1}$ of global AGC loss (~1% of total forest area).

Despite the initial loss of AGC following stand replacement, young forests often transition quickly to a net carbon sink[51]. This is reflected in a significant correlation between the fraction of old forests (>200 years) replaced by young stands (<20 years) and the 10-year trend in net ecosystem exchange (NEE), based on GAMIv2.0 data and atmospheric inversion estimates from nine Global Carbon Budget 2023 (GCB2023) inversion models[22] (Fig. 4c) ($R^2$ = 0.28, slope = −34.32 gC m$^{-2}$ yr$^{-1}$ per % yr$^{-1}$, $P$ < 0.05, $N$ = 134). The negative slope indicates that for each 1% annual increase in the fraction of old forests replaced by young forests, the carbon sink strengthens by

34.32 gC m$^{-2}$ yr$^{-1}$. This suggests that regions with a higher proportion of old forests replaced by young forests tend to exhibit a stronger carbon sink. However, this pattern should not be misinterpreted as an argument in favour of stand replacement. The strong carbon sink observed in these areas primarily results from the long, undisturbed period before stand replacement, during which old forests accumulated large amounts of carbon and created conditions that facilitate rapid regrowth[52]. The time since the previous-to-last disturbance has a critical role in shaping carbon sink strength, as long-undisturbed forests develop favourable soil conditions, structural complexity and nutrient availability that enhance post-disturbance regrowth.

It is also essential to recognize the limitations of this analysis. At the spatial resolution of current atmospheric inversion models (~1°), it is beyond our capacity to disentangle NEE trends across all four age classes robustly. This limitation is particularly pronounced in regions with mixed-age forests and intensive management, where inversion signals integrate fluxes across diverse stand conditions. Whereas exploratory correlations by age class are presented in Extended Data Fig. 8a–c, these should be interpreted cautiously. More interpretable signals may emerge in tropical regions, where large areas of old-growth forest have been replaced by younger stands, resulting in more homogeneous age distributions within inversion grid cells. Still, robust separation of age-specific NEE responses remains challenging at this scale. Accordingly, our main interpretation focuses on the consistent NEE signal linked to old-to-young forest transitions in these areas (Extended Data Fig. 3d), which shows a consistent negative correlation with NEE trends across multiple inversion models (Fig. 4c). Still, broader factors such as $CO_2$ fertilization, temperature shifts and climate variability may also influence NEE trends. While we acknowledge these influences, our analysis isolates the relationship between stand replacement and NEE without attempting to disentangle climate-driven effects.

Young, fast-growing forests can sequester carbon at rates higher than old forests, particularly in tropical regions, where young regrowing forests (<20 years) can absorb $CO_2$ up to 20 times faster than old forests[52]. Yet they do not recover the AGC stocks of old forests within a meaningful timeframe for climate policy. Complete restoration of species composition can take centuries[53–55]. The ability of forests to accumulate carbon over centuries highlights the importance of preserving old forests, which store far more carbon than young regrowing stands. Although undisturbed ageing forests contribute to long-term carbon storage (Fig. 3b), the net $CO_2$ uptake trends

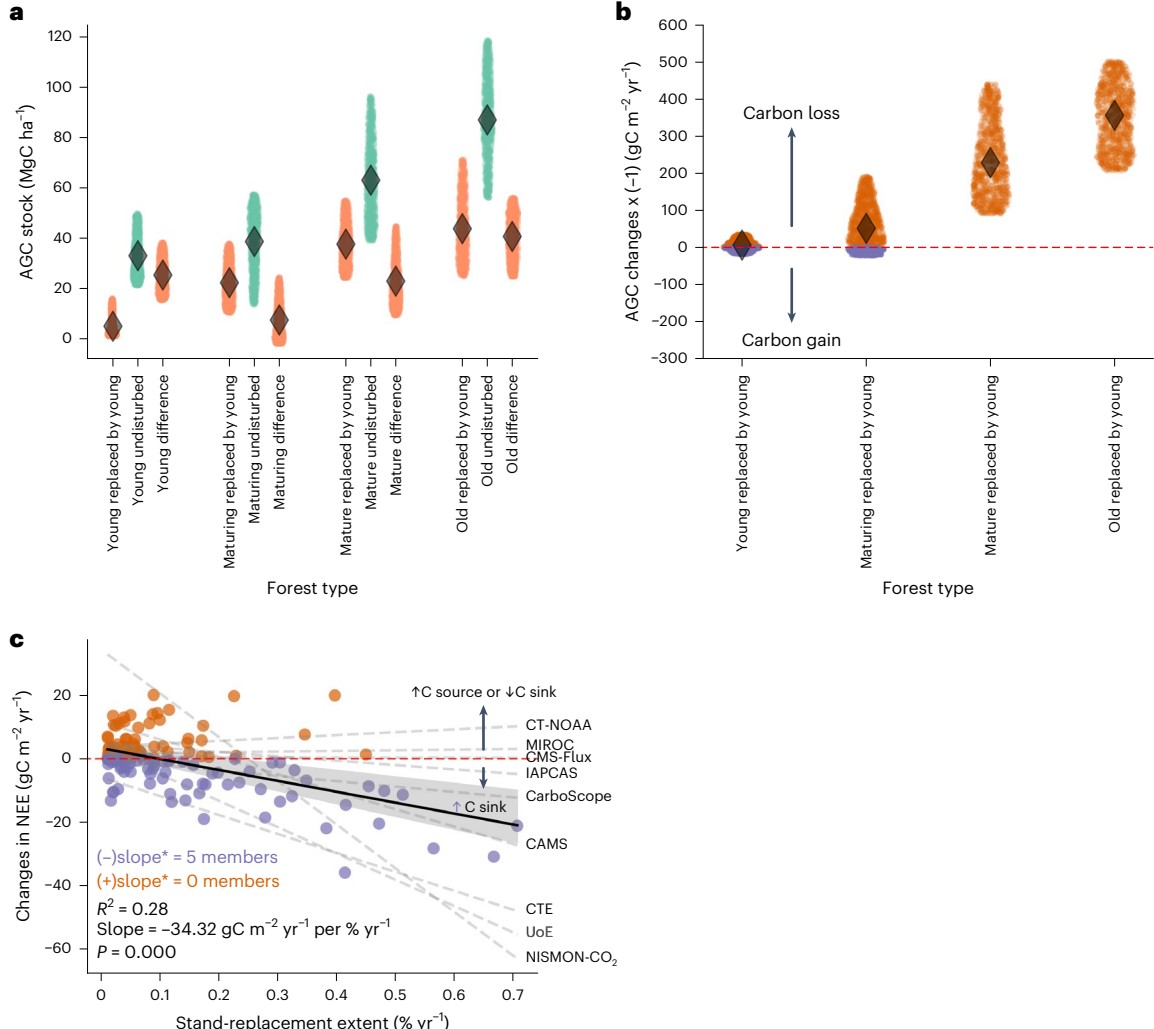

**Fig. 4 | Carbon stocks and net fluxes associated with forest age dynamics.**
**a**, AGCs stocks across forest age classes, distinguishing between stand-replaced forests and undisturbed ageing forests, expressed per unit area at a one-degree pixel level. The stand-replaced categories represent the AGC stock of forests at a given age class (young, maturing, mature or old) before stand replacement.
**b**, Net carbon changes for stand-replaced forests across different forest age categories. In **a** and **b**, the median values from the 20 biomass realizations are displayed, and the spread represents the spatial variation within a given age class.
**c**, The relationship between the fraction of old forests replaced by young stands (that is, stand-replacement extent) and changes in NEE from inversions between circa 2020 (average of 2019–2021) and 2010 (average of 2009–2011) is shown. The dark solid line represents the linear regression on the ensemble estimates, whereas the dashed grey lines indicate the regressions for the nine individual atmospheric inversion models. One-degree grid cells with less than 20% forest cover have been masked. To smooth the spatial distribution of net $CO_2$ fluxes, we applied a Gaussian filter (length = 500 km, equivalent to approximately four one-degree pixels). This smoothing technique reduces noise in the data and helps minimize the influence of local transport errors. To minimize noise and regional variability due to atmospheric transport errors, all spatial data were aggregated using area-weighted averaging over 5 × 5 grid cells (that is, a coarsening scale of 5° × 5°). CAMS, Copernicus Atmosphere Monitoring Service; CarboScope, Jena CarboScope Atmospheric Inversion System; CMS-Flux, Carbon Monitoring System Flux; CTE, CarbonTracker Europe; IAPCAS, Institute of Atmospheric Physics Carbon Assimilation System; ICT-NOAA, CarbonTracker NOAA; MIROC, Model for Interdisciplinary Research on Climate; NISMON-$CO_2$, NICAM-based Inverse Simulation for Monitoring of $CO_2$; UoE, University of Edinburgh Inversion System.

suggest that stand-replacement processes—not just naturally age-ing forests—are key drivers of recent flux changes. Nevertheless, it is essential to recognize that forest harvesting, leading to biomass loss, impacts atmospheric carbon levels in multiple ways. Harvesting does not immediately result in equivalent carbon emissions being released into the atmosphere. Forests continue to assimilate carbon, and the fate of harvested biomass—whether rapidly released through burning, gradually decomposed as coarse woody debris or stored in wood products for a long time—determines the timing and extent of carbon released into the atmosphere. The balance between immediate and delayed carbon release may explain the residuals between the proportion of forests replaced by young stands and net $CO_2$ fluxes (Fig. 4c), which are further influenced by variations in hydrometeorological conditions[56].

It is crucial to contextualize the impact of stand replacement on carbon flux dynamics within the broader picture of global forest dynamics. Global forests act as a net carbon sink, particularly in boreal and temperate regions[5,57]. However, deforestation and land-use conversion increasingly challenge this climate-mitigation role, where forests are permanently lost rather than replaced. In deforestation hotspots, atmospheric inversion data mainly show these areas as net $CO_2$ sources, primarily due to emissions from burning activities. For example, in regions such as the eastern edge of the Amazon, fire-related emissions can exceed the carbon uptake of remaining forests[58], reinforcing the dominant role of deforestation in increasing atmospheric $CO_2$. Thus, whereas stand-replacement processes contribute to enhanced $CO_2$ uptake in some regions, they should not be viewed in isolation from the broader issue of forest loss and the protection of old forests. Furthermore, the

relationship between old-stand replacement and NEE trends varies across atmospheric inversion models. Whereas five of the nine models indicate a significant negative correlation, others show no clear relationship (Fig. 4c), highlighting uncertainties in top-down $CO_2$ flux estimates[59]. This variability underscores the need for cautious interpretation when linking stand-replacement processes to regional NEE trends. Nevertheless, the consistent signal across multiple models and spatial windows (Supplementary Fig. 8) supports the conclusion that old forest replacement has been key in shaping recent $CO_2$ flux dynamics.

Our findings emphasize the complex role of forest age transitions in shaping the global carbon balance. Whereas old forests store the highest carbon stocks, young regrowing forests, particularly those replacing old forests, can contribute substantially to $CO_2$ uptake due to rapid biomass accumulation. However, this high sequestration rate primarily results from the favourable conditions left behind by the replaced old forests, including legacy soil carbon, nutrient availability and established seed banks. This does not imply that stand replacement is a viable climate mitigation strategy, as old forests take centuries to develop and cannot be restored within a policy-relevant timeframe. Young forests can never fully replace the long-term carbon storage capacity and ecological functions of old forests. Protecting existing forests remains the most effective and immediate strategy for mitigating climate change[60].

To explore the implications of age dynamics for future carbon storage, we projected forest age distributions and AGC stocks to 2050 under two idealized pathways: BAU and a forest conservation scenario that halts stand-replacement after 2030 (Methods, Extended Data Fig. 9 and Supplementary Fig. 5). Whereas BAU results in stable AGC stocks due to ongoing turnover, the conservation scenario projects an annual increase in carbon storage of +0.55 to +0.63 PgC yr$^{-1}$ relative to 2020 levels, primarily in maturing and mature forests. Despite reduced AGC in young stands, total projected AGC by 2050 is higher under conservation (209–306 PgC) than BAU (200–296 PgC), illustrating the mitigation potential of conserving ageing forests. Yet, the modest gain suggests that conservation alone will not be sufficient for achieving large-scale sequestration goals.

It is also crucial to distinguish between young forests emerging from stand replacement and those established through afforestation. Whereas afforestation represents an expansion of forest cover, stand replacement involves the replacement of existing forests, thereby reducing their long-term carbon storage potential. Although afforestation can enhance carbon sequestration[27,37], its effectiveness depends on the forest type; planted, homogeneous forests provide fewer benefits for carbon storage, biodiversity and local ecosystem resilience than naturally regenerating forests. Beyond carbon storage, natural forests provide irreplaceable ecosystem and social benefits compared to monoculture plantations. Yet, only 34% of global restoration commitments prioritize natural forests, whereas 45% focus on monocultures[61]. Our study highlights the need for targeted conservation and forest management policies that strike a balance between preserving old forests and implementing sustainable harvesting and regrowth strategies. We call for targeted management and conservation strategies that carefully consider forest age and type to ensure forests remain functional, land-based systems for mitigating anthropogenic climate change.

## Methods

### Satellite-based AGB product
We used the ESA-CCI biomass v4 data, which provide estimates of AGB density in woody vegetation for 2010 and 2020 (https://data.ceda.ac.uk/neodc/esacci/biomass/data/agb/maps/v4.0). Those data are derived from a combination of Earth observation data from the Copernicus Sentinel-1 mission, Envisat's ASAR instrument and JAXA's Advanced Land Observing Satellite (ALOS-1 and ALOS-2), along with additional information from Earth observation sources. We calculated the changes in AGB for 2010–2020 by taking the differences between the 2020 and 2010 maps. To quantify the uncertainties in the biomass

maps, we generated 20 realizations of the AGB data by introducing controlled perturbations to the mean biomass estimates. Specifically, for each time step and each member of our ensemble, we added a scaled version of the standard deviation of the biomass, with the scaling factor drawn from a normal distribution clipped to the range [−1, 1]. This approach allows us to simulate the natural variability and uncertainty in the biomass data. We then combined these perturbed maps to create a comprehensive dataset representing the possible range of biomass values. This enables us to provide confidence intervals and robust statistical measures of uncertainty in our estimates. The following equation can describe the perturbation applied:

$$\text{AGB}_{i,j,k}(t) = \max\left(\mu_{i,j}(t) + S_k \times \sigma_{i,j}(t), 0\right) \tag{1}$$

where $\text{AGB}_{i,j,k}(t)$ is the perturbed aboveground biomass at location $(i, j)$ and time $t$ for the $k$th member, $\mu_{i,j}(t)$ is the mean aboveground biomass, $\sigma_{i,j}(t)$ is the standard deviation of the aboveground biomass and $S_k$ is the scaling factor drawn from a normal distribution $N(0, \frac{1}{3})$ and clipped to the range [−1,1]. The use $N(0, \frac{1}{3})$ ensures that the perturbations are centred around zero, introducing symmetric and controlled variability. The standard deviation $\frac{1}{3}$ moderates the variability to prevent extreme perturbations, thus maintaining the realism of the biomass estimates. Clipping the values to [−1,1] further ensures that the introduced variations are within a reasonable range.

The AGB estimates, expressed as dry organic mass, were converted to AGC using the Intergovernmental Panel on Climate Change default carbon fraction of 0.47.

### Global Age Mapping Integration v2.0
GAMIv2.0[16,21] is an updated version of the Max Planck Institute for Biogeochemistry (MPI-BGC) forest age product, providing global forest age distributions for 2010 and 2020 at a 100-metre resolution. This version leverages the machine learning algorithm XGBoost to generate data-driven estimates of forest age, integrating over 40,000 forest inventory plots, biomass and height measurements, remote sensing observations and climate data. This multi-variable approach accounts for structural and ecological variations across climatic gradients, ensuring that forest age is not inferred from biomass alone but reflects broader growth patterns (Extended Data Fig. 2). Unlike previous global age products that assume monotonic or simplified growth trajectories, GAMIv2.0 is explicitly designed to represent the full diversity of age–biomass relationships, including saturation and declines in older age classes, across biomes. This improves alignment with patterns observed in ground-based studies and enhances the suitability of the dataset for carbon modelling applications.

A key improvement in GAMIv2.0 is the integration of Landsat-based disturbance history with machine-learning-based forest age estimates. The methodology for mapping time since the last disturbance and afforestation is based on Potapov et al.[4] and Hansen et al.[62]. The process involves the following steps:

- Mapping time since last disturbance and afforestation
  - Last disturbance layer: disturbed forest areas are identified using the 'forests affected by stand-replacement disturbances or degradation' layer from Potapov et al.[4]. The Hansen et al.[62] dataset is then used to determine the year of forest loss, which is subtracted from 2020 to obtain the time since the last disturbance. Negative values (indicating losses in 2021 and 2022) are masked.
  - Stable forest layer: the 'stable forest extent' layer from Potapov et al.[4] is used to identify undisturbed forest areas as of 2020.
  - Forest gain layer (Afforestation): afforested (regrown) forest areas are mapped using the 'forest extent gain' layer and the canopy height data from 2020.

- Integrating Landsat-based and machine-learning-based age estimates
  To improve forest age predictions, we merged the Landsat-based time since disturbance estimates with the machine-learning-derived forest age estimates through the following decision rules:

  - If the Landsat-based time since disturbance is ≤19 years and the machine-learning predicted age is higher, we assign the Landsat-based estimate.
  - If the Landsat-based time since disturbance is ≤19 years and the machine-learning predicted age is equal to or lower than the Landsat-based estimate, we retain the machine-learning estimate.
  - If the Landsat-based time since disturbance is ≥20 years, we retain the machine-learning estimate, regardless of its value.

This fusion approach refines the estimation of time since the last stand-replacement event over the past 20 years, addressing biases that tend to overestimate the age of young forests in regrowing or afforested areas (Supplementary Fig. 7). Despite these improvements, some biases remain, including a potential overestimation of young forest age and an underestimation of older forest age. This has important implications for our results: the observed decline in young forests may be less pronounced than suggested, whereas the reduction in older forests could be more substantial.

- Addressing uncertainties in age estimates
  The age maps were generated using an ensemble approach to account for uncertainties:

  - Aleatoric uncertainty: different biomass data realizations were incorporated into the model to capture variability in input data.
  - Epistemic uncertainty: multiple XGBoost models were trained with varying hyperparameter settings to assess variability in model performance and predictions.

By incorporating these uncertainty quantifications, we provide a more robust estimate of forest age distribution. We also derived age-class fraction products at one-degree resolution, categorizing forests into two-decade intervals up to 200 years, followed by 201–300 and >300-year age classes.

The GAMIv2.0 product and its associated uncertainties can be visualized in the following Google Earth Engine app: https://besnard-sim.users.earthengine.app/view/globalforestage.

## Atmospheric inversion data

We used the 1 × 1° gridded 'co2flux' output generated by nine (CAMS, CMS-Flux, CT-NOAA, CTE, CarboScope, IAPCAS, MIROC, NISMON-CO$_2$ and UoE) inversion models within the Regional Carbon Cycle Assessment and Processes Project (RECCAP-2)[22] (https://meta.icos-cp.eu/objects/FHbD8OTgCb7Tlvs99lUDApO0). These models incorporate CO$_2$ mole fraction measurements from surface stations and total column mole fraction data from satellites. The selected atmospheric inversion models include CAMS, sEXTocNEET, CTE2022, NISMON-CO$_2$, CMS-Flux, UoE, GONGGA, THU and CAMS-Satellite. We adjusted the net carbon fluxes by excluding lateral fluxes from the calculation. Such adjustments included removing riverine carbon export to estuaries and coastal oceans and accounting for carbon transfers in crops and emissions from cement production. By adjusting the data, we aligned the atmospheric inversion findings more closely with the forest age estimates inferred from the GAMIv2.0 product. We masked out non-forested pixels. Finally, atmospheric inversion-based net CO$_2$ fluxes estimates were smoothed with a Gaussian filter while maintaining the total intensity by redistributing values only among valid pixels. The NaN values in the input remain unchanged in the output. This approach uses a Gaussian distribution for intensity redistribution, considering only valid pixels. The filter's smoothing scale is set to a length of 500 km, defining the filter's physical length in kilometres. Sigma for the filter is calculated based on a specified degree of longitude at the equator. This approach effectively smooths the data while respecting its original structure and missing values. We created two net CO$_2$ fluxes, circa 2010 (average between 2009–2011) and 2020 (average between 2019–2021), from which we determined the NEE changes by subtracting the latter from the former.

## Estimating the changes in forest age distribution for 2010–2020

Estimating changes in forest age distribution between 2010 and 2020 involved several steps using the 100 m GAMIv2.0 product. First, we aggregated the data to a one-degree pixel resolution for 2010 and 2020 using an average resampling method. The resulting datasets provided estimates of the mean forest age at a one-degree scale for both years (Fig. 1a). In parallel, we calculated forest age differences at GAMIv2.0's native resolution (100 m). This difference map was then resampled to a one-degree resolution using the same averaging method to maintain spatial consistency (Fig. 1c). Additionally, we quantified the total area occupied by young (0–20 years old), maturing (21–80 years old), mature (81–200 years old) and old (>200 years old) forests in both 2010 and 2020 (Fig. 1b). This analysis was performed independently for each of the 20 forest age maps. In this study, it is essential to distinguish between forest ageing, which progresses at a constant rate over time, and forest growth, representing changes in biomass accumulation over time. Growth rates vary depending on stand dynamics, disturbance history and environmental conditions, whereas ageing itself remains uniform. However, because GAMI estimates forest age as a function of biomass, canopy height and climate variables, variations in modelled age primarily reflect inferred structural and ecological differences rather than the simple passage of time. This distinction is crucial for correctly interpreting trends in forest age distribution and their implications for carbon dynamics.

## Partitioning stand-replaced and undisturbed ageing forests

We first computed the difference in modelled forest age between 2010 and 2020 at 100-m resolution using the GAMIv2.0 product to classify forests into stand-replaced and undisturbed ageing categories. Undisturbed ageing forests were identified as pixels where the age difference was exactly 10 years, indicating that the forest remained undisturbed and aged naturally over the decade. Stand-replaced forests were identified as pixels where the age difference was less than 10 years, meaning that a stand-replacing disturbance occurred between 2010 and 2020. This includes negative values, where the 2020 forest age is lower than in 2010, reflecting a more recent disturbance that had little time for regrowth to occur.

To further analyse these patterns, we generated two derivative products for stand-replaced forests:

Age difference magnitude—capturing the extent of age reduction due to disturbance.
Binary classification—distinguishing stand-replaced (1) vs non-stand-replaced (0) pixels.

Similarly, we created equivalent products based on an age difference of exactly 10 years for undisturbed ageing forests.

To understand how initial forest age influences disturbance and ageing patterns, we stratified both stand-replaced and undisturbed ageing forests into four age classes, using their 2010 GAMIv2.0-estimated age: young forests (0–20 years), maturing forests (21–80 years), mature forests (81–200 years) and old forests (>200 years).

Finally, the age difference and binary classification products were resampled to a one-degree resolution using an average resampling

method, enabling large-scale analysis. This process was conducted independently for each of the 20 forest age maps to account for model variability. These resampled products served two purposes: the binary classification was used to compute the fraction of forest pixels in each one-degree grid cell that experienced stand replacement or undisturbed ageing. At the same time, the magnitude of the age difference captured the mean severity of age reduction or gain. These metrics were then used to quantify the spatial extent and intensity of age transitions (for example, Figs. 1c and 2a–c).

## Assessing the covariation of forest age shifts with changes in net CO$_2$ fluxes

First, we analysed carbon stock across forest age classes, focusing on stand-replaced and undisturbed ageing forests. This involved partitioning 100-m pixels based on age difference, as previously described, to identify distinct categories of forests that had undergone varying degrees of change over the study period. Each partitioned class was then assigned an estimate of carbon stock. Using an average resampling method, those carbon stock estimate products were resampled to a one-degree pixel resolution. In addition, we estimated the total carbon stock across age classes by multiplying the product of the carbon stock estimates and the area of each age-class partition. This was done independently for the stand-replaced and undisturbed ageing forest categories. Similarly, we calculated net carbon stock changes for stand-replaced forests by computing the difference in AGC stocks between 2010 and 2020. Specifically, for each spatial unit, the AGC stock 2020 was subtracted from the AGC stock 2010, yielding the net change over the decade ($\Delta AGC = AGC_{2020} - AGC_{2010}$). To express these changes in a flux-consistent sign convention, we multiplied the resulting values by −1 so that positive values represent carbon loss (biosphere-to-atmosphere flux, typically due to disturbance or decomposition). By contrast, negative values indicate carbon accumulation (biomass regrowth). This approach ensures consistency with commonly used flux representations while preserving the original stock change information. Finally, we estimate the average annual growth rate across age classes for stand-replaced and undisturbed ageing forests as follows:

$$\text{Ratio} = \frac{\text{biomass}_{2020}}{\text{biomass}_{2010}} \qquad (2)$$

$$\text{Average annual growth rate} = \text{ratio}^{\frac{1}{n}} - 1 \qquad (3)$$

where $n$ is the number of years between the two measurements. In this case, $n = 10$ years.

Such a procedure was done independently for the corresponding 20 forest age and biomass maps.

To assess the covariation between forest age shifts and net CO$_2$ flux changes, we analysed the relationship between stand-replacement fractions and changes in NEE derived from atmospheric inversions. To ensure the robustness of our analysis, we extract data within spatial windows of different resolutions (that is, 2° × 2°, 5° × 5° and 10° × 10°) before performing the regression analysis. This spatial window approach ensures that stand-replacement fractions and NEE changes remain spatially coherent within each inversion model. We compute the median stand-replacement fraction (20 members) and corresponding median NEE changes across all available inversion models (nine members) for each spatial window. To assess the robustness of our results, we apply Jackknife resampling, systematically excluding individual NEE inversion members to test their influence on the overall trend.

## Constructing future forest age distribution under different pathways

To explore how changes in forest management could influence future age-class dynamics and AGC storage, we developed projections of

forest age distributions to the year 2050 under two hypothetical management scenarios: BAU and forest conservation. These scenarios are inspired by, but do not strictly adhere to, the objectives of global forest policy initiatives, such as the New York Declaration on Forests, which aims to halt natural forest loss by 2030 and restore 350 million hectares of degraded landscapes and forests. We aim to provide a first-order estimate of how AGC stocks might evolve in response to shifts in forest age structures under idealized management pathways. Our projections do not account for climate-driven effects such as CO$_2$ fertilization or altered disturbance regimes.

**BAU scenario.** This scenario assumes that current forest management practices and policies remain unchanged for a period of 30 years. Consequently, the distribution of forest age classes is projected to mirror the pattern observed for the 2010–2020 period. As a result, the forest age structure in 2050 is expected to be similar to the one observed in 2020. To calculate the total carbon storage capacity for this scenario, we use a straightforward method: multiplying the total area of each age class in 2050 by its respective total carbon stock estimate and then dividing this product by the total area of each age class in 2020. In the BAU scenario, the total area of each age class in 2050 is expected to be the same as in 2020. Such carbon stock estimation can be estimated as follows:

Total AGC BAU$_{2050}$

$$= \sum \frac{\text{total area of age class BAU}_{i,2050} \times \text{total stock}_{i,2020}}{\text{total area of age class}_{i,2020}} \qquad (4)$$

Where:

The total area of age-class BAU$_{i,2050}$ is the area of the $i$th age class (in hectares) in 2050.

Total area of age class$_{i,2020}$ is the area of the $i$th age class (in hectares) in 2020.

Total stock$_{i,2020}$ is the estimated total carbon stock for the $i$th age class (in MgC per hectare) in 2020.

**Forest conservation scenario.** Aligning with the objectives of the New York Declaration on Forests, this scenario maintains the continuation of existing management practices until 2030, followed by a period of non-intervention in forest areas up to 2050. The aim is to halt BAU stand replacement by 2030 and allow forests to age naturally thereafter. In this scenario, the forest age-class distribution in 2030 is anticipated to be identical to that of 2020, whereas by 2050, all forests are expected to have matured by an additional 20 years. The total carbon stocks under this scenario are estimated by multiplying each age class's total area in 2050 by its corresponding total carbon stock estimates in 2020. This product will then be divided by the total area of each age class in 2020. The carbon stock changes estimation for the forest conservation scenario can be expressed as follows:

Total AGC conservation$_{2050}$

$$= \sum \frac{\text{total area of age class conservation}_{i,2050} \times \text{total stock}_{i,2020}}{\text{total area of age class}_{i,2020}} \qquad (5)$$

Where:

Total area of age-class conservation$_{2050}$ is the area of the $i$th age class (in hectares) in 2050.

Total area of age class$_{i,2020}$ is the area of the $i$th age class (in hectares) in 2020.

Total stock$_{i,2020}$ is the estimated total carbon stock for the $i$th age class (in MgC per hectare) in 2020.

## Exploring the role of forest management

First, to investigate the influence of forest management on the relationship between forest age and biomass (Supplementary Fig. 1), we used

mixed-effects models on a global scale. These models included the management category as a fixed effect and geographic coordinates as random effects to capture spatial heterogeneity. This approach allowed us to isolate the role of management while accounting for spatial variability across different regions. To manage regional variability effectively, the analysis was conducted within 2-degree latitude by 2-degree longitude windows, ensuring sufficient data points within each window for robust statistical analysis. The mixed-effects model can be expressed as:

$$\text{biomass} \sim \text{management type} \times \text{forest age} + (1 \mid \text{latitude/longitude}) \quad (6)$$

Where:

Biomass is the dependent variable, while management type (managed vs unmanaged) and forest age are fixed effects. Latitude and longitude are random effects that account for spatial variability.

Within each 2-degree latitude by 2-degree longitude window, we extracted the $P$ value of the interaction term between the management category and forest age to assess the influence of management practices on the relationship between forest age and biomass.

Second, we investigated the relationship between forest age and logging fraction (Supplementary Fig. 6). To do so, we used a forest management map[46] to calculate a metric related to the logging fraction as follows:

$$\text{Logging fraction} = \frac{\text{class20} + \text{class32}}{\text{class11} + \text{class20} + \text{class32} + \text{class31} + \text{class40} + \text{class53}} \quad (7)$$

Where:

class 11 – naturally regenerating forest without any signs of management, including primary forests;

class 20 – naturally regenerating forest with signs of management, for example, logging, clear-cuts and so on;

class 31 – planted forests;

class 32 – plantation forests (rotation time up to 15 years);

class 40 – oil palm plantations; and

class 53 – agroforestry.

## Reporting summary

Further information on research design is available in the Nature Portfolio Reporting Summary linked to this article.

## Data availability

The authors declare that the Methods section contains all the methods needed to evaluate the paper's conclusions.

## Code availability

All codes for the analysis of the data are available via GitHub at https://github.com/simonbesnard1/forest-age-upscale.git.

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

## Acknowledgements

We thank the Global Land Monitoring group members at the GFZ Helmholtz Centre for Geosciences for providing feedback on the presented results. We acknowledge funding support by the European Union through the FORWARDS (https://forwards-project.eu/) and OpenEarthMonitor (https://earthmonitor.org/) projects. V.H.A.H. acknowledges support from the Consultative Group on International Agricultural Research (CGIAR) MITIGATE+ project, the World Resources Institute (WRI) Land and Carbon Lab and the Open Earth Monitor Project funded by the European Union (grant agreement number 101059548). N.C and H.Y. acknowledge support by the Project Office BIOMASS (grant number 50EE1904) funded by the German Federal Ministry for Economic Affairs and Climate Action. M.S. acknowledges the support of the European Space Agency (ESA) under contract number 4000123662/18/I-NB (Climate Change Initiative BIOMASS). We recognize the use of OpenAI's ChatGPT and Grammarly AI tools to enhance the manuscript's sentence structure, conciseness and grammatical accuracy. Importantly, we emphasize that the conceptualization, research and results presented are entirely our own.

## Author contributions

S.B. designed the research, performed the analysis and drafted the manuscript. S.B. prepared the GAMIv2.0 dataset, M.S. prepared the ESA-CCI biomass product, I.L. prepared the outputs of GCP atmospheric inversions and P.C. provided the lateral flux products. All authors contributed to the interpretation of the results and revised the text.

## Funding

## Competing interests

The authors declare no competing interests.

## Additional information

**Extended data** is available for this paper at https://doi.org/10.1038/s41559-025-02821-5.

**Correspondence and requests for materials** should be addressed to Simon Besnard.

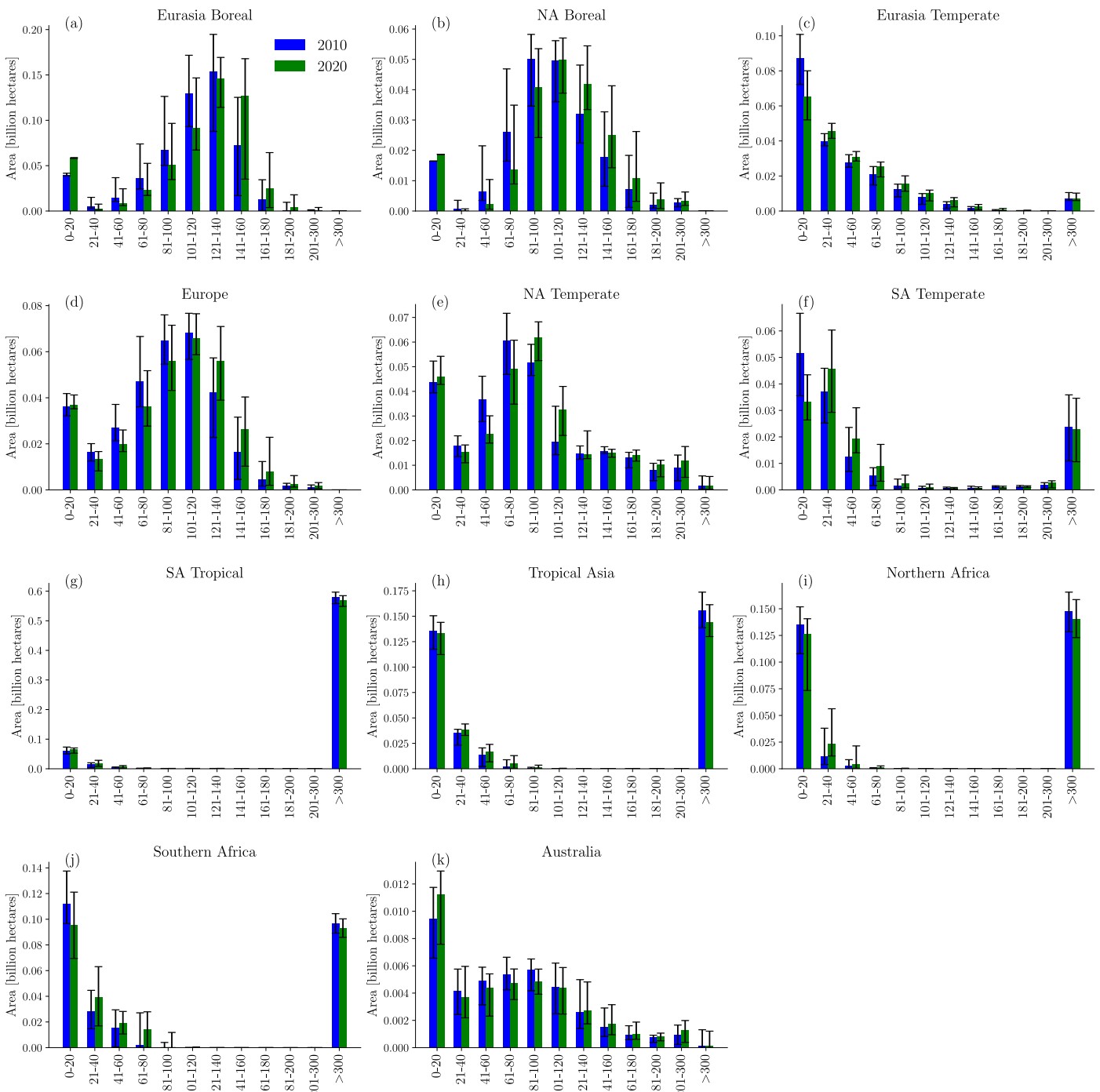

**Extended Data Fig. 1 | Forest age class distribution across TRANSCOM regions in 2010 and 2020. a–k**, Total forest area per age class in 2010 (blue) and 2020 (green) across the eleven TRANSCOM land regions (see Supplementary Fig. 3). Values represent the median estimates across 20 ensemble members, with error bars indicating the 5th and 95th percentiles.

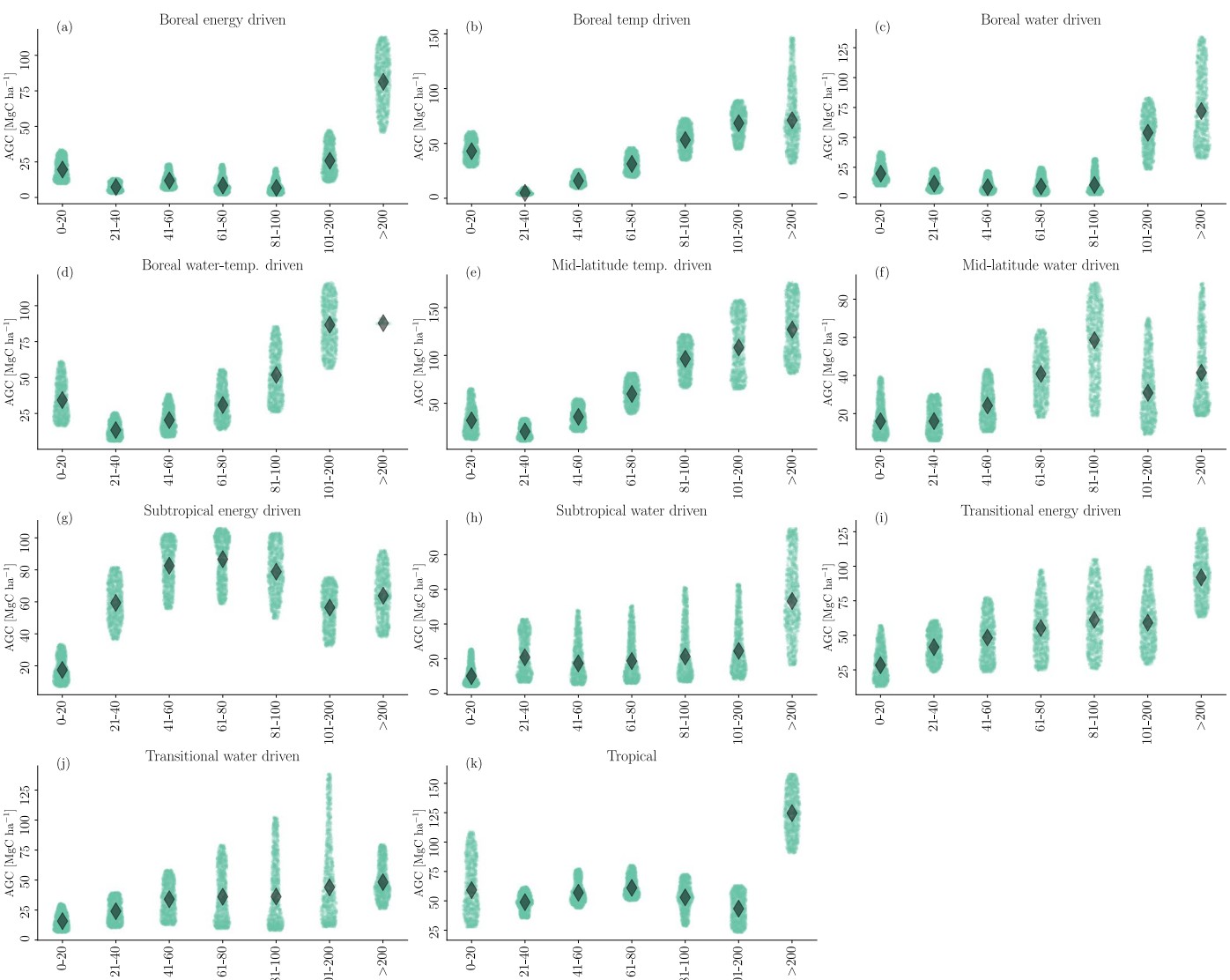

**Extended Data Fig. 2 | Biomass–age relationships across hydro-climate zones. a–k**, Relationship between forest biomass and age across hydro-climate zones, based on the GAMI dataset and ESA-CCI biomass v4 data for 2020. The figure illustrates consistent patterns of biomass accumulation with forest aging, reflecting broad ecological trends. Although the GAMI product incorporates ESA-CCI biomass data (see Methods), this comparison independently validates the robustness of the observed growth trajectories.

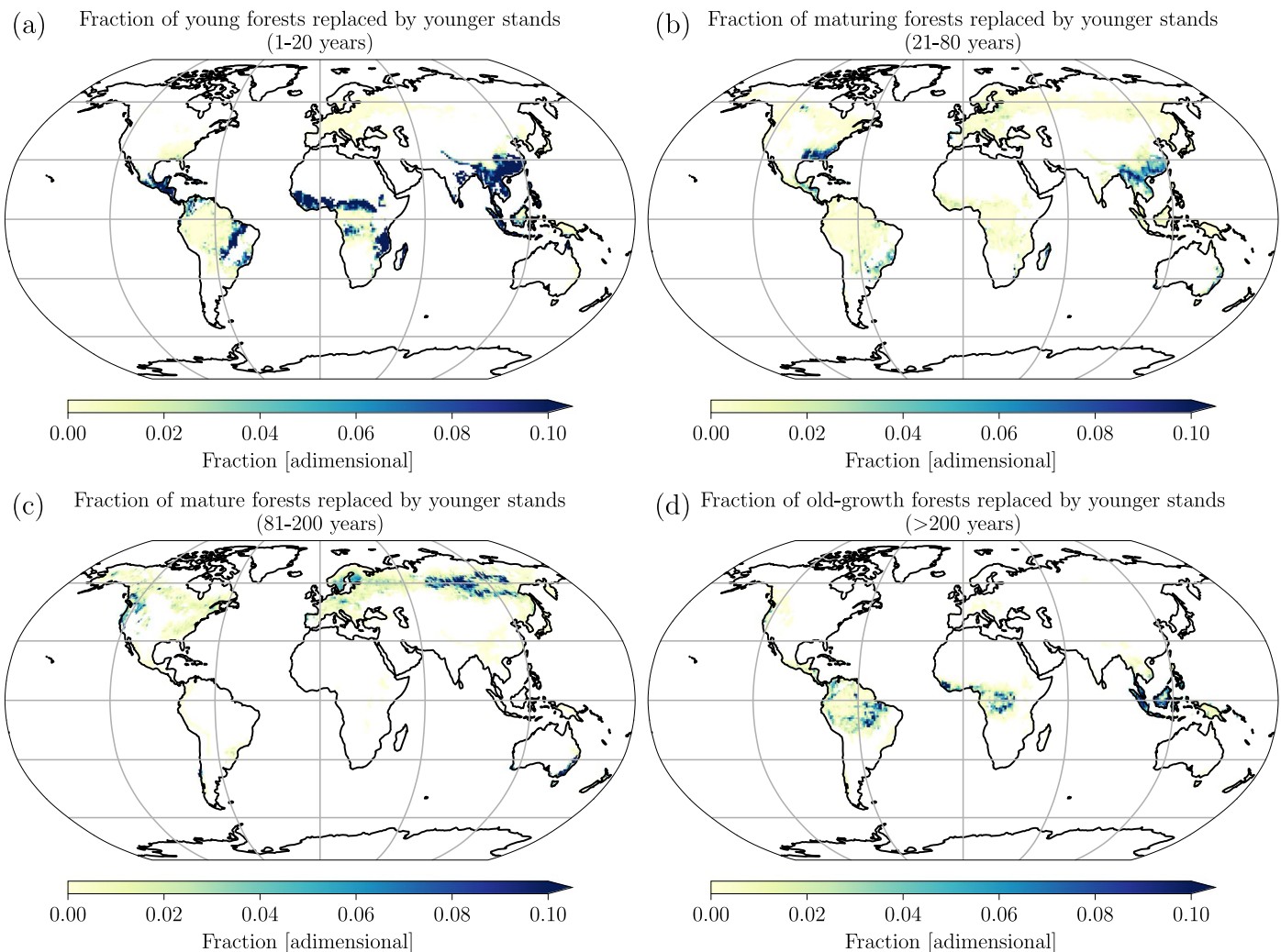

**Extended Data Fig. 3 | Spatial distribution of stand-replaced forests.** Spatial distribution of stand-replaced forests that were (**a**) young (forest age <= 20 years old), (**b**) intermediate (21–80 years old), (**c**) mature (81–200 years old), and (**d**) old forests (forest age > 200 years old) in 2010.

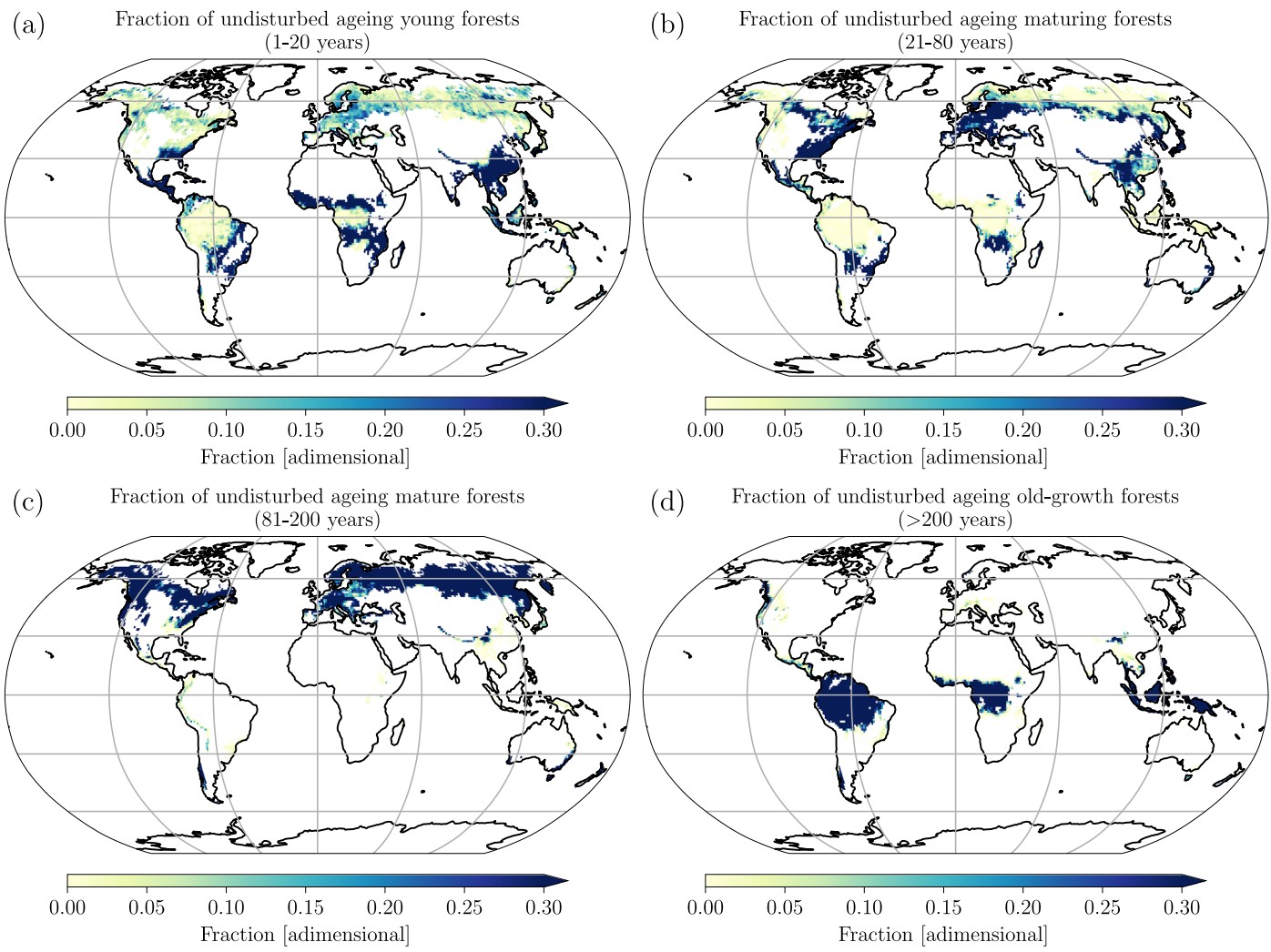

**Extended Data Fig. 4 | Spatial distribution of undisturbed ageing forests.** Spatial distribution of undisturbed ageing forests (**a**) young (forest age <= 20 years old), (**b**) intermediate (21–80 years old), (**c**) mature (81–200 years old), and (**d**) old forests (forest age > 200 years old) in 2010.

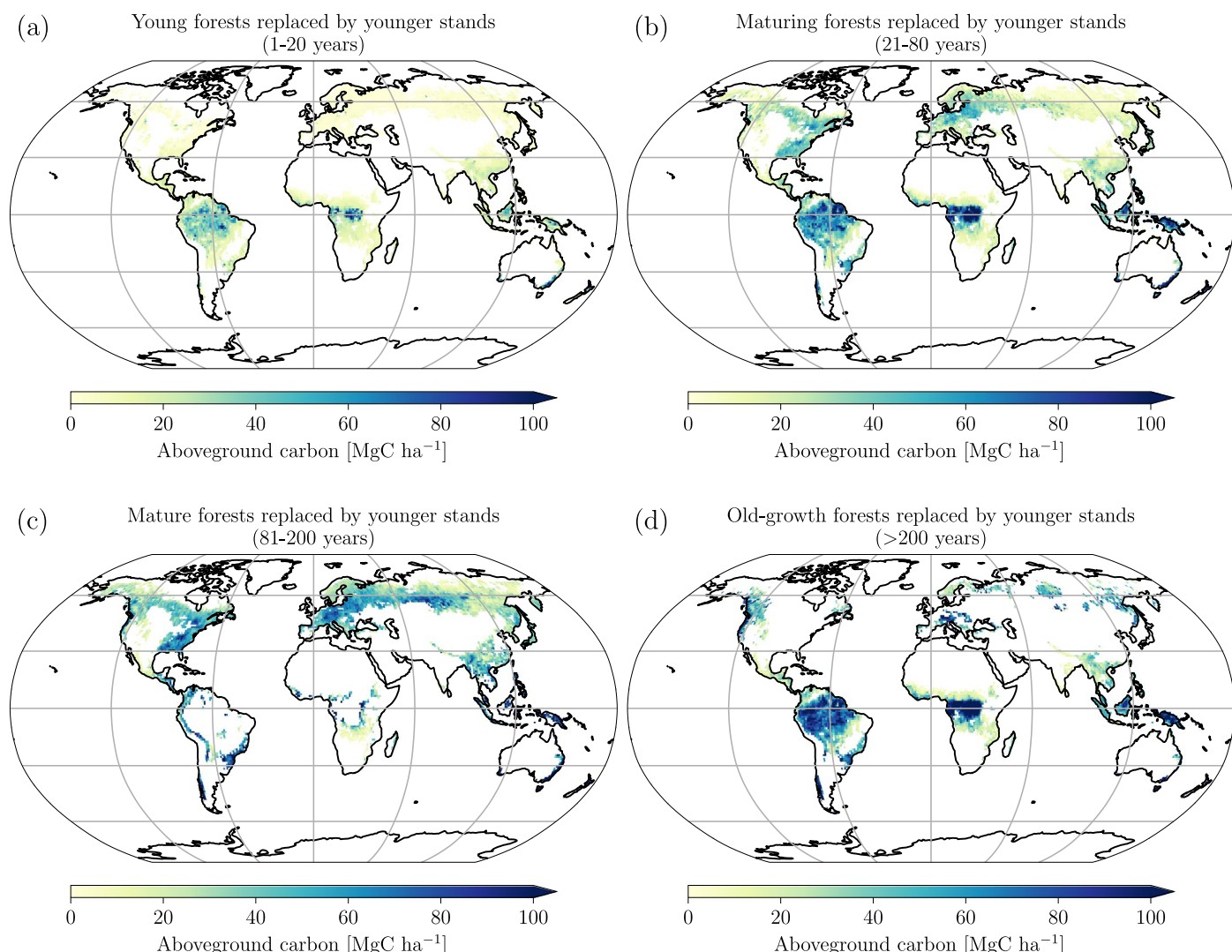

**Extended Data Fig. 5 | Spatial distribution of carbon stocks in stand-replaced forests.** Spatial distribution of carbon stocks (in MgC ha-1) in stand-replaced forests (**a**) young (forest age <= 20 years old), (**b**) intermediate (21–80 years old), (**c**) mature (81–200 years old), and (**d**) old forests (forest age > 200 years old) in 2010. Each pixel represents a median estimate of the 100 m pixels belonging to a specific category within each one-degree pixel.

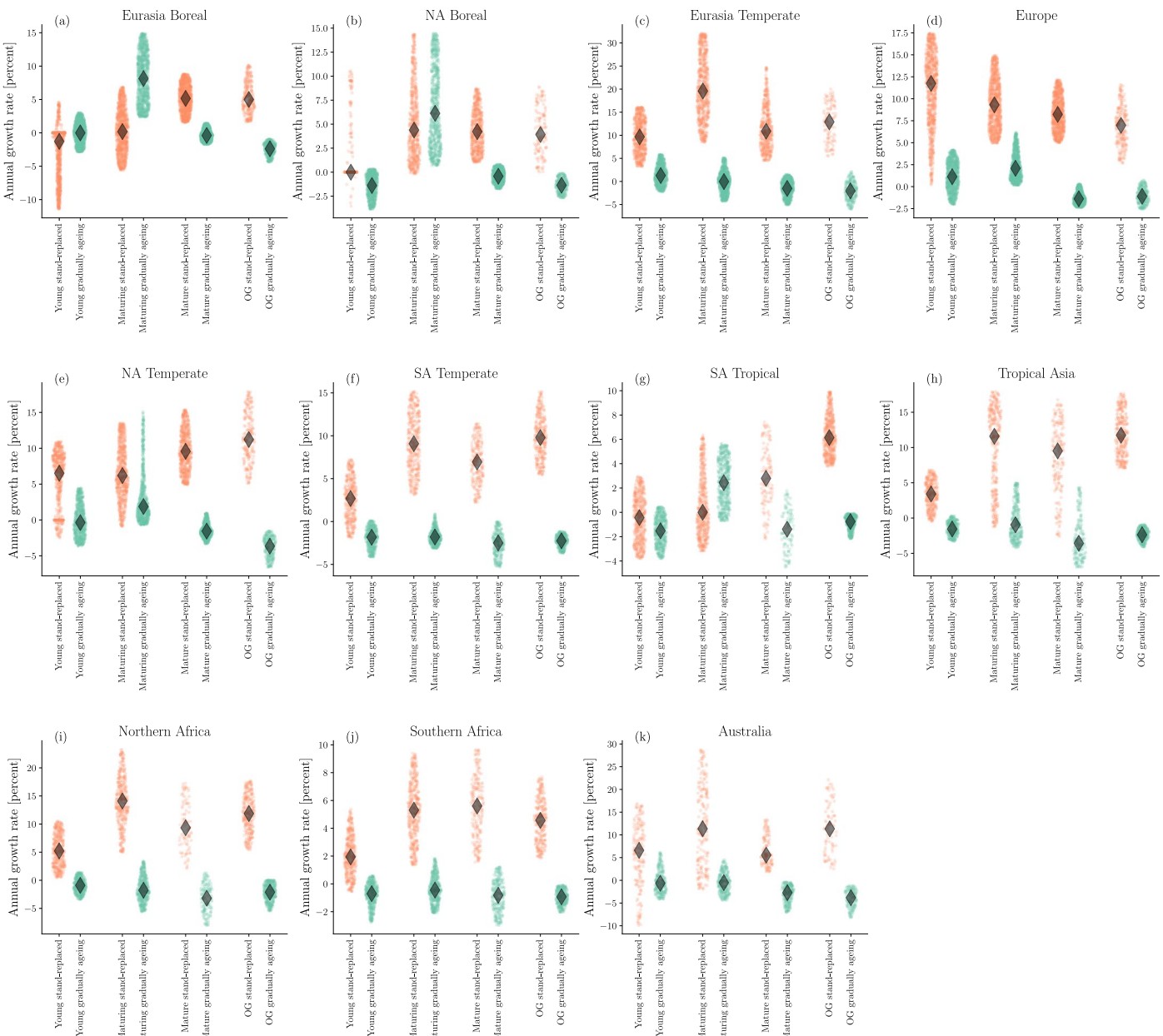

**Extended Data Fig. 6 | Annual growth rates across age classes. a–k,** Annual growth rates across age classes for forests replaced by young stands (in orange) and undisturbed ageing forests (in green) in each TRANSCOM-Land region. The annual growth rate represents the net change in biomass per year. Stand-replaced forests (orange) correspond to areas where older stands have been replaced by younger forests, which generally exhibit higher growth rates due to their early successional stage. In contrast, undisturbed ageing forests (green) represent forests that continue ageing without stand replacement, typically showing lower growth rates as they approach maturity.

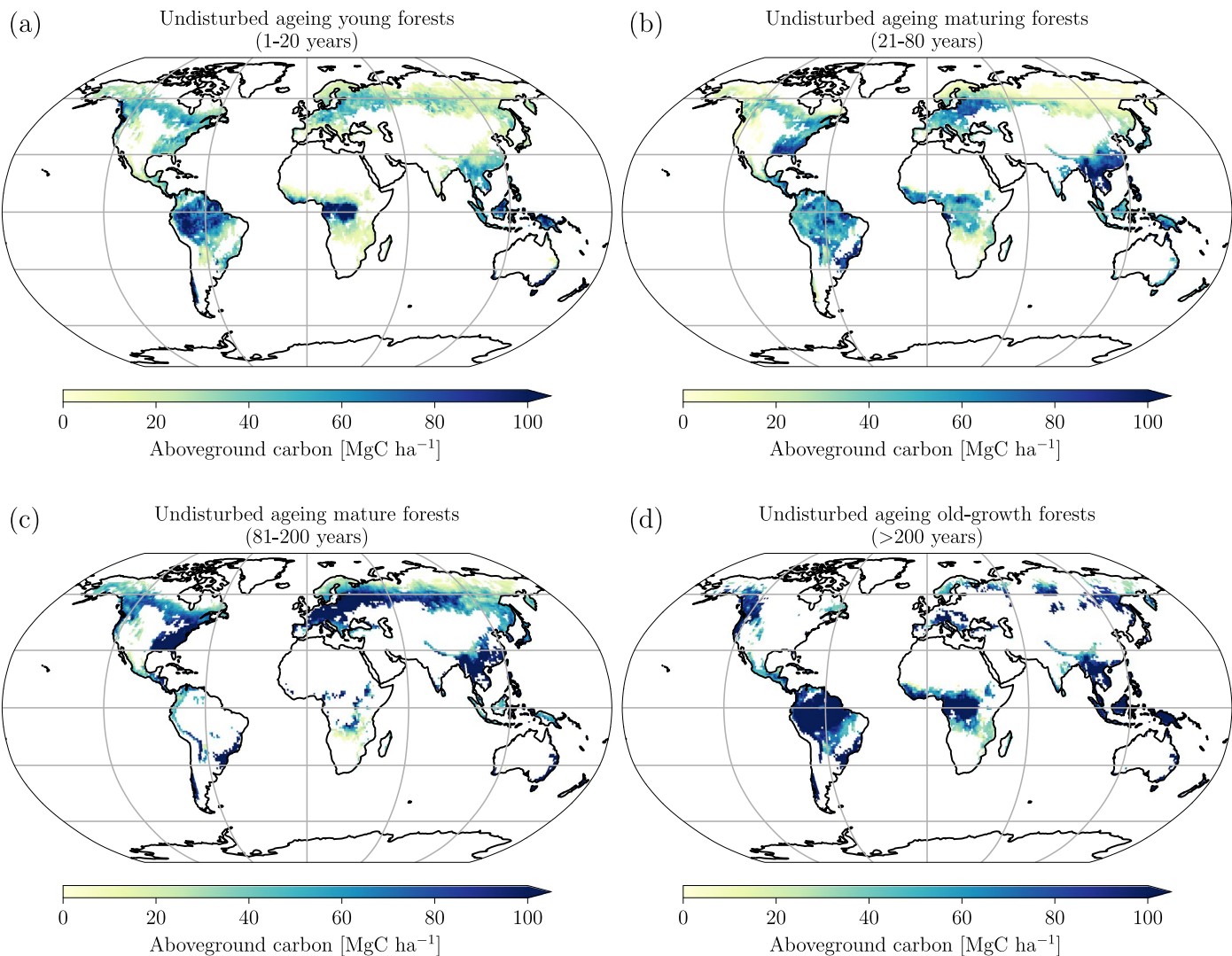

**Extended Data Fig. 7 | Spatial distribution of carbon stocks in undisturbed ageing established forest.** Spatial distribution of carbon stocks (in MgC ha-1) in undisturbed ageing established forests (**a**) young (forest age <= 20 years old), (**b**) intermediate (21–80 years old), (**c**) mature (81–200 years old), and (**d**) old forests (forest age > 200 years old) in 2010. Each pixel represents a median estimate of the 100 m pixels belonging to a specific category within each one-degree pixel.

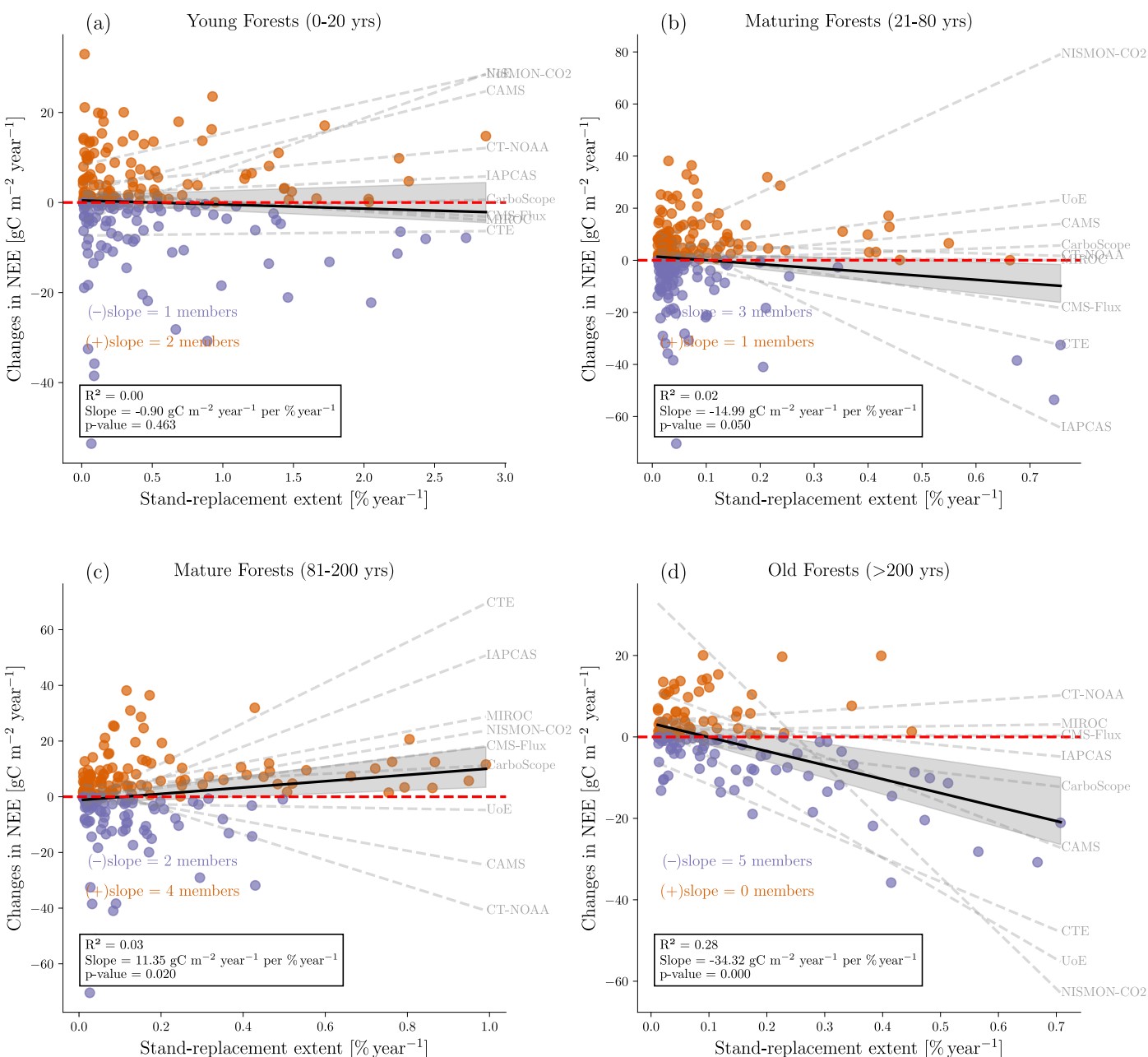

**Extended Data Fig. 8 | Relationship between the extent of stand-replaced forests and changes in net $CO_2$ fluxes.** Relationship between the fraction of forests replaced by young stands (that is, stand-replacement extent) and changes in net $CO_2$ fluxes between circa 2020 (average of 2019–2021) and 2010 (average of 2009–2011) for young (**a**), maturing (**b**), mature (**c**) and old (**d**) stand-replaced forests. The dark solid line represents the linear regression on the ensemble estimates, while the dashed grey lines indicate the regressions for the nine individual atmospheric inversion models. To smooth the spatial distribution of net CO2 fluxes, we applied a Gaussian filter (length = 500 km, equivalent to approximately four one-degree pixels). This smoothing technique reduces noise in the data and helps minimise the influence of local transport errors.

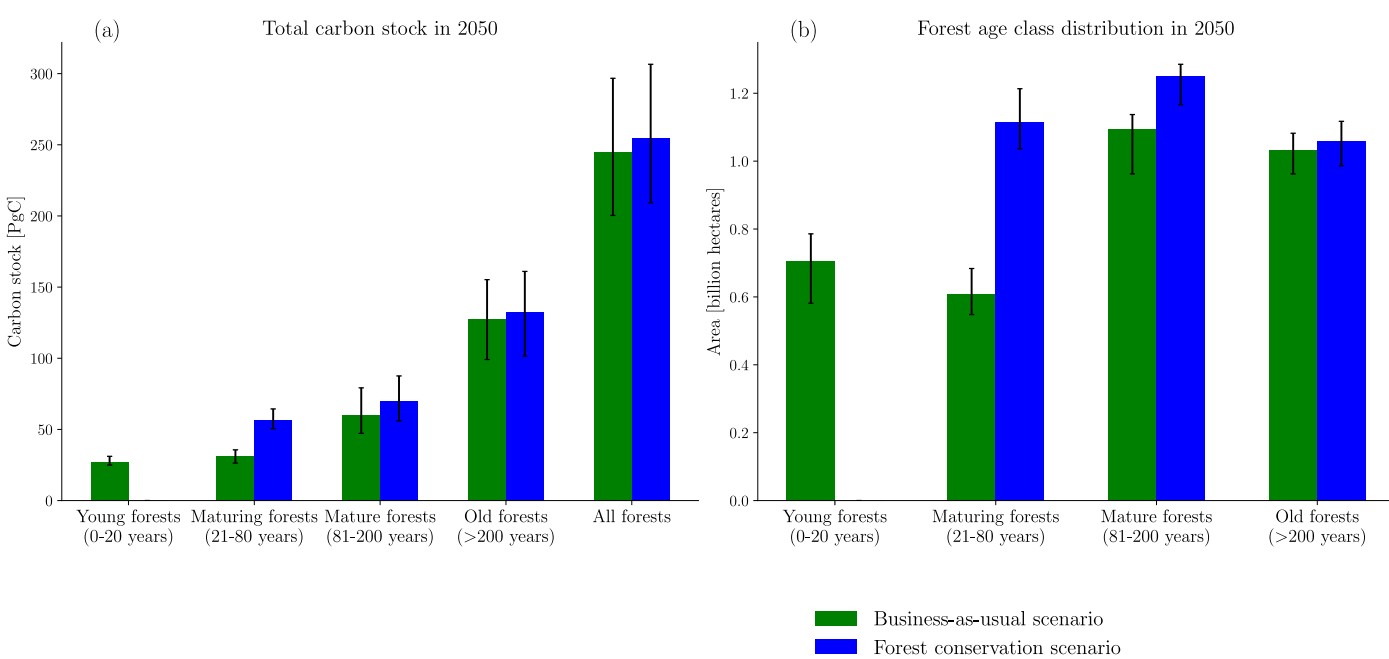

**Extended Data Fig. 9 | Potential total carbon stock in 2050.** Potential total carbon stock (**a**) in 2050 across age classes for the two scenarios. We also show the forest age class distribution in 2050 (**b**) under BAU and forest conservation scenarios. The young forest age class include the stand-replaced and undisturbed ageing forests.

**Extended Data Table 1 | Global forest area by age class in 2010 and 2020 with net changes over the decade**

| | Young forests | Maturing forests | | | Mature forests | | | | | | | Old forests | |
|---|---|---|---|---|---|---|---|---|---|---|---|---|---|
| | 0-20 | 21-40 | 41-60 | 61-80 | 81-100 | 101-120 | 121-140 | 141-160 | 161-180 | 180-200 | 201-300 | >300 |
| **Total area in 2010** | $0.76_{0.66}^{0.82}$ | $0.21_{0.18}^{0.24}$ | $0.18_{0.15}^{0.20}$ | $0.23_{0.18}^{0.27}$ | $0.26_{0.23}^{0.31}$ | $0.28_{0.25}^{0.31}$ | $0.26_{0.17}^{0.31}$ | $0.13_{0.05}^{0.21}$ | $0.043_{0.016}^{0.073}$ | $0.017_{0.0075}^{0.033}$ | $0.018_{0.0061}^{0.026}$ | $1.044_{0.97}^{1.12}$ |
| **Total area in 2020** | $0.71_{0.58}^{0.79}$ | $0.25_{0.22}^{0.30}$ | $0.17_{0.15}^{0.20}$ | $0.20_{0.15}^{0.23}$ | $0.25_{0.20}^{0.29}$ | $0.26_{0.23}^{0.32}$ | $0.28_{0.24}^{0.30}$ | $0.20_{0.09}^{0.27}$ | $0.065_{0.024}^{0.12}$ | $0.024_{0.009}^{0.046}$ | $0.023_{0.0095}^{0.035}$ | $1.01_{0.94}^{1.07}$ |
| **Absolute net changes** | -0.040 | +0.034 | -0.0098 | -0.023 | -0.023 | -0.019 | +0.014 | +0.055 | +0.022 | +0.0077 | +0.0059 | -0.034 |
| **Relative net changes** | -5.20% | +16.61% | -5.71% | -14.54% | -8.11% | -7.19% | +5.49% | +58.96% | +55.00% | +41.30% | +34.96% | -3.26% |

Table shows the total global forest area distributed across age classes for the years 2010 and 2020, along with net area changes over the 2010–2020 period. Area estimates are reported in billions of hectares. The table includes the median and the 5th and 95th percentiles across 20 ensemble members.

# Reporting Summary

## Statistics

For all statistical analyses, confirm that the following items are present in the figure legend, table legend, main text, or Methods section.

| n/a | Confirmed | |
|---|---|---|
| ☐ | ☒ | The exact sample size ($n$) for each experimental group/condition, given as a discrete number and unit of measurement |
| ☒ | ☐ | A statement on whether measurements were taken from distinct samples or whether the same sample was measured repeatedly |
| ☒ | ☐ | The statistical test(s) used AND whether they are one- or two-sided *Only common tests should be described solely by name; describe more complex techniques in the Methods section.* |
| ☐ | ☒ | A description of all covariates tested |
| ☐ | ☒ | A description of any assumptions or corrections, such as tests of normality and adjustment for multiple comparisons |
| ☐ | ☒ | A full description of the statistical parameters including central tendency (e.g. means) or other basic estimates (e.g. regression coefficient) AND variation (e.g. standard deviation) or associated estimates of uncertainty (e.g. confidence intervals) |
| ☐ | ☒ | For null hypothesis testing, the test statistic (e.g. $F$, $t$, $r$) with confidence intervals, effect sizes, degrees of freedom and $P$ value noted *Give P values as exact values whenever suitable.* |
| ☒ | ☐ | For Bayesian analysis, information on the choice of priors and Markov chain Monte Carlo settings |
| ☒ | ☐ | For hierarchical and complex designs, identification of the appropriate level for tests and full reporting of outcomes |
| ☐ | ☒ | Estimates of effect sizes (e.g. Cohen's $d$, Pearson's $r$), indicating how they were calculated |

*Our web collection on statistics for biologists contains articles on many of the points above.*

## Software and code

Policy information about availability of computer code

| Data collection | All code used in our study are available in the provided Gitlab repository |
|---|---|
| Data analysis | All code used in our study are available in the provided Gitlab repository |

For manuscripts utilizing custom algorithms or software that are central to the research but not yet described in published literature, software must be made available to editors and reviewers. We strongly encourage code deposition in a community repository (e.g. GitHub). See the Nature Portfolio guidelines for submitting code & software for further information.

## Data

Policy information about availability of data

All manuscripts must include a data availability statement. This statement should provide the following information, where applicable:
- Accession codes, unique identifiers, or web links for publicly available datasets
- A description of any restrictions on data availability
- For clinical datasets or third party data, please ensure that the statement adheres to our policy

The authors declare that the Methods section contains all the methods needed to evaluate the paper's conclusions.

# Research involving human participants, their data, or biological material

Policy information about studies with human participants or human data. See also policy information about sex, gender (identity/presentation), and sexual orientation and race, ethnicity and racism.

| | |
|---|---|
| Reporting on sex and gender | Use the terms sex (biological attribute) and gender (shaped by social and cultural circumstances) carefully in order to avoid confusing both terms. Indicate if findings apply to only one sex or gender; describe whether sex and gender were considered in study design; whether sex and/or gender was determined based on self-reporting or assigned and methods used. Provide in the source data disaggregated sex and gender data, where this information has been collected, and if consent has been obtained for sharing of individual-level data; provide overall numbers in this Reporting Summary. Please state if this information has not been collected. Report sex- and gender-based analyses where performed, justify reasons for lack of sex- and gender-based analysis. |
| Reporting on race, ethnicity, or other socially relevant groupings | Please specify the socially constructed or socially relevant categorization variable(s) used in your manuscript and explain why they were used. Please note that such variables should not be used as proxies for other socially constructed/relevant variables (for example, race or ethnicity should not be used as a proxy for socioeconomic status). Provide clear definitions of the relevant terms used, how they were provided (by the participants/respondents, the researchers, or third parties), and the method(s) used to classify people into the different categories (e.g. self-report, census or administrative data, social media data, etc.) Please provide details about how you controlled for confounding variables in your analyses. |
| Population characteristics | Describe the covariate-relevant population characteristics of the human research participants (e.g. age, genotypic information, past and current diagnosis and treatment categories). If you filled out the behavioural & social sciences study design questions and have nothing to add here, write "See above." |
| Recruitment | Describe how participants were recruited. Outline any potential self-selection bias or other biases that may be present and how these are likely to impact results. |
| Ethics oversight | Identify the organization(s) that approved the study protocol. |

Note that full information on the approval of the study protocol must also be provided in the manuscript.

# Field-specific reporting

Please select the one below that is the best fit for your research. If you are not sure, read the appropriate sections before making your selection.

☐ Life sciences ☐ Behavioural & social sciences ☒ Ecological, evolutionary & environmental sciences

For a reference copy of the document with all sections, see nature.com/documents/nr-reporting-summary-flat.pdf

# Life sciences study design

All studies must disclose on these points even when the disclosure is negative.

| | |
|---|---|
| Sample size | Describe how sample size was determined, detailing any statistical methods used to predetermine sample size OR if no sample-size calculation was performed, describe how sample sizes were chosen and provide a rationale for why these sample sizes are sufficient. |
| Data exclusions | Describe any data exclusions. If no data were excluded from the analyses, state so OR if data were excluded, describe the exclusions and the rationale behind them, indicating whether exclusion criteria were pre-established. |
| Replication | Describe the measures taken to verify the reproducibility of the experimental findings. If all attempts at replication were successful, confirm this OR if there are any findings that were not replicated or cannot be reproduced, note this and describe why. |
| Randomization | Describe how samples/organisms/participants were allocated into experimental groups. If allocation was not random, describe how covariates were controlled OR if this is not relevant to your study, explain why. |
| Blinding | Describe whether the investigators were blinded to group allocation during data collection and/or analysis. If blinding was not possible, describe why OR explain why blinding was not relevant to your study. |

# Behavioural & social sciences study design

All studies must disclose on these points even when the disclosure is negative.

| | |
|---|---|
| Study description | Briefly describe the study type including whether data are quantitative, qualitative, or mixed-methods (e.g. qualitative cross-sectional, quantitative experimental, mixed-methods case study). |
| Research sample | State the research sample (e.g. Harvard university undergraduates, villagers in rural India) and provide relevant demographic information (e.g. age, sex) and indicate whether the sample is representative. Provide a rationale for the study sample chosen. For studies involving existing datasets, please describe the dataset and source. |

| Sampling strategy | *Describe the sampling procedure (e.g. random, snowball, stratified, convenience). Describe the statistical methods that were used to predetermine sample size OR if no sample-size calculation was performed, describe how sample sizes were chosen and provide a rationale for why these sample sizes are sufficient. For qualitative data, please indicate whether data saturation was considered, and what criteria were used to decide that no further sampling was needed.* |
|---|---|
| Data collection | *Provide details about the data collection procedure, including the instruments or devices used to record the data (e.g. pen and paper, computer, eye tracker, video or audio equipment) whether anyone was present besides the participant(s) and the researcher, and whether the researcher was blind to experimental condition and/or the study hypothesis during data collection.* |
| Timing | *Indicate the start and stop dates of data collection. If there is a gap between collection periods, state the dates for each sample cohort.* |
| Data exclusions | *If no data were excluded from the analyses, state so OR if data were excluded, provide the exact number of exclusions and the rationale behind them, indicating whether exclusion criteria were pre-established.* |
| Non-participation | *State how many participants dropped out/declined participation and the reason(s) given OR provide response rate OR state that no participants dropped out/declined participation.* |
| Randomization | *If participants were not allocated into experimental groups, state so OR describe how participants were allocated to groups, and if allocation was not random, describe how covariates were controlled.* |

# Ecological, evolutionary & environmental sciences study design

All studies must disclose on these points even when the disclosure is negative.

| Study description | Assess the impact of forest age transitions on the global carbon balance from 2010-2020 |
|---|---|
| Research sample | Global |
| Sampling strategy | *Note the sampling procedure. Describe the statistical methods that were used to predetermine sample size OR if no sample-size calculation was performed, describe how sample sizes were chosen and provide a rationale for why these sample sizes are sufficient.* |
| Data collection | *Describe the data collection procedure, including who recorded the data and how.* |
| Timing and spatial scale | 2010 and 2020 at 100m spatial resolution |
| Data exclusions | *If no data were excluded from the analyses, state so OR if data were excluded, describe the exclusions and the rationale behind them, indicating whether exclusion criteria were pre-established.* |
| Reproducibility | *Describe the measures taken to verify the reproducibility of experimental findings. For each experiment, note whether any attempts to repeat the experiment failed OR state that all attempts to repeat the experiment were successful.* |
| Randomization | *Describe how samples/organisms/participants were allocated into groups. If allocation was not random, describe how covariates were controlled. If this is not relevant to your study, explain why.* |
| Blinding | *Describe the extent of blinding used during data acquisition and analysis. If blinding was not possible, describe why OR explain why blinding was not relevant to your study.* |

Did the study involve field work? ☐ Yes  ☒ No

## Field work, collection and transport

| Field conditions | *Describe the study conditions for field work, providing relevant parameters (e.g. temperature, rainfall).* |
|---|---|
| Location | *State the location of the sampling or experiment, providing relevant parameters (e.g. latitude and longitude, elevation, water depth).* |
| Access & import/export | *Describe the efforts you have made to access habitats and to collect and import/export your samples in a responsible manner and in compliance with local, national and international laws, noting any permits that were obtained (give the name of the issuing authority, the date of issue, and any identifying information).* |
| Disturbance | *Describe any disturbance caused by the study and how it was minimized.* |

# Reporting for specific materials, systems and methods

We require information from authors about some types of materials, experimental systems and methods used in many studies. Here, indicate whether each material, system or method listed is relevant to your study. If you are not sure if a list item applies to your research, read the appropriate section before selecting a response.

## Materials & experimental systems

| n/a | Involved in the study |
|-----|------------------------|
| ☒ | Antibodies |
| ☒ | Eukaryotic cell lines |
| ☒ | Palaeontology and archaeology |
| ☒ | Animals and other organisms |
| ☒ | Clinical data |
| ☒ | Dual use research of concern |
| ☒ | Plants |

## Methods

| n/a | Involved in the study |
|-----|------------------------|
| ☒ | ChIP-seq |
| ☒ | Flow cytometry |
| ☒ | MRI-based neuroimaging |

# Antibodies

| Antibodies used | Describe all antibodies used in the study; as applicable, provide supplier name, catalog number, clone name, and lot number. |
|---|---|
| Validation | Describe the validation of each primary antibody for the species and application, noting any validation statements on the manufacturer's website, relevant citations, antibody profiles in online databases, or data provided in the manuscript. |

# Eukaryotic cell lines

Policy information about cell lines and Sex and Gender in Research

| Cell line source(s) | State the source of each cell line used and the sex of all primary cell lines and cells derived from human participants or vertebrate models. |
|---|---|
| Authentication | Describe the authentication procedures for each cell line used OR declare that none of the cell lines used were authenticated. |
| Mycoplasma contamination | Confirm that all cell lines tested negative for mycoplasma contamination OR describe the results of the testing for mycoplasma contamination OR declare that the cell lines were not tested for mycoplasma contamination. |
| Commonly misidentified lines (See ICLAC register) | Name any commonly misidentified cell lines used in the study and provide a rationale for their use. |

# Palaeontology and Archaeology

| Specimen provenance | Provide provenance information for specimens and describe permits that were obtained for the work (including the name of the issuing authority, the date of issue, and any identifying information). Permits should encompass collection and, where applicable, export. |
|---|---|
| Specimen deposition | Indicate where the specimens have been deposited to permit free access by other researchers. |
| Dating methods | If new dates are provided, describe how they were obtained (e.g. collection, storage, sample pretreatment and measurement), where they were obtained (i.e. lab name), the calibration program and the protocol for quality assurance OR state that no new dates are provided. |

☐ Tick this box to confirm that the raw and calibrated dates are available in the paper or in Supplementary Information.

| Ethics oversight | Identify the organization(s) that approved or provided guidance on the study protocol, OR state that no ethical approval or guidance was required and explain why not. |
|---|---|

Note that full information on the approval of the study protocol must also be provided in the manuscript.

# Animals and other research organisms

Policy information about studies involving animals; ARRIVE guidelines recommended for reporting animal research, and Sex and Gender in Research

| Laboratory animals | For laboratory animals, report species, strain and age OR state that the study did not involve laboratory animals. |
|---|---|
| Wild animals | Provide details on animals observed in or captured in the field; report species and age where possible. Describe how animals were caught and transported and what happened to captive animals after the study (if killed, explain why and describe method; if released, say where and when) OR state that the study did not involve wild animals. |

| | |
|---|---|
| Reporting on sex | *Indicate if findings apply to only one sex; describe whether sex was considered in study design, methods used for assigning sex. Provide data disaggregated for sex where this information has been collected in the source data as appropriate; provide overall numbers in this Reporting Summary. Please state if this information has not been collected. Report sex-based analyses where performed, justify reasons for lack of sex-based analysis.* |
| Field-collected samples | *For laboratory work with field-collected samples, describe all relevant parameters such as housing, maintenance, temperature, photoperiod and end-of-experiment protocol OR state that the study did not involve samples collected from the field.* |
| Ethics oversight | *Identify the organization(s) that approved or provided guidance on the study protocol, OR state that no ethical approval or guidance was required and explain why not.* |

Note that full information on the approval of the study protocol must also be provided in the manuscript.

# Clinical data

Policy information about clinical studies

All manuscripts should comply with the ICMJE guidelines for publication of clinical research and a completed CONSORT checklist must be included with all submissions.

| | |
|---|---|
| Clinical trial registration | *Provide the trial registration number from ClinicalTrials.gov or an equivalent agency.* |
| Study protocol | *Note where the full trial protocol can be accessed OR if not available, explain why.* |
| Data collection | *Describe the settings and locales of data collection, noting the time periods of recruitment and data collection.* |
| Outcomes | *Describe how you pre-defined primary and secondary outcome measures and how you assessed these measures.* |

# Dual use research of concern

Policy information about dual use research of concern

## Hazards

Could the accidental, deliberate or reckless misuse of agents or technologies generated in the work, or the application of information presented in the manuscript, pose a threat to:

No | Yes
- ☒ ☐ Public health
- ☒ ☐ National security
- ☒ ☐ Crops and/or livestock
- ☒ ☐ Ecosystems
- ☒ ☐ Any other significant area

## Experiments of concern

Does the work involve any of these experiments of concern:

No | Yes
- ☒ ☐ Demonstrate how to render a vaccine ineffective
- ☒ ☐ Confer resistance to therapeutically useful antibiotics or antiviral agents
- ☒ ☐ Enhance the virulence of a pathogen or render a nonpathogen virulent
- ☒ ☐ Increase transmissibility of a pathogen
- ☒ ☐ Alter the host range of a pathogen
- ☒ ☐ Enable evasion of diagnostic/detection modalities
- ☒ ☐ Enable the weaponization of a biological agent or toxin
- ☒ ☐ Any other potentially harmful combination of experiments and agents

# Plants

Seed stocks
*Report on the source of all seed stocks or other plant material used. If applicable, state the seed stock centre and catalogue number. If plant specimens were collected from the field, describe the collection location, date and sampling procedures.*

Novel plant genotypes
*Describe the methods by which all novel plant genotypes were produced. This includes those generated by transgenic approaches, gene editing, chemical/radiation-based mutagenesis and hybridization. For transgenic lines, describe the transformation method, the number of independent lines analyzed and the generation upon which experiments were performed. For gene-edited lines, describe the editor used, the endogenous sequence targeted for editing, the targeting guide RNA sequence (if applicable) and how the editor was applied.*

Authentication
*Describe any authentication procedures for each seed stock used or novel genotype generated. Describe any experiments used to assess the effect of a mutation and, where applicable, how potential secondary effects (e.g. second site T-DNA insertions, mosiacism, off-target gene editing) were examined.*

# ChIP-seq

## Data deposition

☐ Confirm that both raw and final processed data have been deposited in a public database such as GEO.

☐ Confirm that you have deposited or provided access to graph files (e.g. BED files) for the called peaks.

Data access links
*May remain private before publication.*
*For "Initial submission" or "Revised version" documents, provide reviewer access links. For your "Final submission" document, provide a link to the deposited data.*

Files in database submission
*Provide a list of all files available in the database submission.*

Genome browser session
(e.g. UCSC)
*Provide a link to an anonymized genome browser session for "Initial submission" and "Revised version" documents only, to enable peer review. Write "no longer applicable" for "Final submission" documents.*

## Methodology

Replicates
*Describe the experimental replicates, specifying number, type and replicate agreement.*

Sequencing depth
*Describe the sequencing depth for each experiment, providing the total number of reads, uniquely mapped reads, length of reads and whether they were paired- or single-end.*

Antibodies
*Describe the antibodies used for the ChIP-seq experiments; as applicable, provide supplier name, catalog number, clone name, and lot number.*

Peak calling parameters
*Specify the command line program and parameters used for read mapping and peak calling, including the ChIP, control and index files used.*

Data quality
*Describe the methods used to ensure data quality in full detail, including how many peaks are at FDR 5% and above 5-fold enrichment.*

Software
*Describe the software used to collect and analyze the ChIP-seq data. For custom code that has been deposited into a community repository, provide accession details.*

# Flow Cytometry

## Plots

Confirm that:

☐ The axis labels state the marker and fluorochrome used (e.g. CD4-FITC).

☐ The axis scales are clearly visible. Include numbers along axes only for bottom left plot of group (a 'group' is an analysis of identical markers).

☐ All plots are contour plots with outliers or pseudocolor plots.

☐ A numerical value for number of cells or percentage (with statistics) is provided.

## Methodology

Sample preparation
*Describe the sample preparation, detailing the biological source of the cells and any tissue processing steps used.*

Instrument
*Identify the instrument used for data collection, specifying make and model number.*

Software
*Describe the software used to collect and analyze the flow cytometry data. For custom code that has been deposited into a community repository, provide accession details.*

| Cell population abundance | *Describe the abundance of the relevant cell populations within post-sort fractions, providing details on the purity of the samples and how it was determined.* |
|---|---|
| Gating strategy | *Describe the gating strategy used for all relevant experiments, specifying the preliminary FSC/SSC gates of the starting cell population, indicating where boundaries between "positive" and "negative" staining cell populations are defined.* |

☐ Tick this box to confirm that a figure exemplifying the gating strategy is provided in the Supplementary Information.

# Magnetic resonance imaging

## Experimental design

| Design type | *Indicate task or resting state; event-related or block design.* |
|---|---|
| Design specifications | *Specify the number of blocks, trials or experimental units per session and/or subject, and specify the length of each trial or block (if trials are blocked) and interval between trials.* |
| Behavioral performance measures | *State number and/or type of variables recorded (e.g. correct button press, response time) and what statistics were used to establish that the subjects were performing the task as expected (e.g. mean, range, and/or standard deviation across subjects).* |

## Acquisition

| Imaging type(s) | *Specify: functional, structural, diffusion, perfusion.* |
|---|---|
| Field strength | *Specify in Tesla* |
| Sequence & imaging parameters | *Specify the pulse sequence type (gradient echo, spin echo, etc.), imaging type (EPI, spiral, etc.), field of view, matrix size, slice thickness, orientation and TE/TR/flip angle.* |
| Area of acquisition | *State whether a whole brain scan was used OR define the area of acquisition, describing how the region was determined.* |

Diffusion MRI    ☐ Used    ☐ Not used

## Preprocessing

| Preprocessing software | *Provide detail on software version and revision number and on specific parameters (model/functions, brain extraction, segmentation, smoothing kernel size, etc.).* |
|---|---|
| Normalization | *If data were normalized/standardized, describe the approach(es): specify linear or non-linear and define image types used for transformation OR indicate that data were not normalized and explain rationale for lack of normalization.* |
| Normalization template | *Describe the template used for normalization/transformation, specifying subject space or group standardized space (e.g. original Talairach, MNI305, ICBM152) OR indicate that the data were not normalized.* |
| Noise and artifact removal | *Describe your procedure(s) for artifact and structured noise removal, specifying motion parameters, tissue signals and physiological signals (heart rate, respiration).* |
| Volume censoring | *Define your software and/or method and criteria for volume censoring, and state the extent of such censoring.* |

## Statistical modeling & inference

| Model type and settings | *Specify type (mass univariate, multivariate, RSA, predictive, etc.) and describe essential details of the model at the first and second levels (e.g. fixed, random or mixed effects; drift or auto-correlation).* |
|---|---|
| Effect(s) tested | *Define precise effect in terms of the task or stimulus conditions instead of psychological concepts and indicate whether ANOVA or factorial designs were used.* |

Specify type of analysis:    ☐ Whole brain    ☐ ROI-based    ☐ Both

| Statistic type for inference<br>(See Eklund et al. 2016) | *Specify voxel-wise or cluster-wise and report all relevant parameters for cluster-wise methods.* |
|---|---|
| Correction | *Describe the type of correction and how it is obtained for multiple comparisons (e.g. FWE, FDR, permutation or Monte Carlo).* |

## Models & analysis

| n/a | Involved in the study |
|-----|------------------------|
| ☐ | ☐ Functional and/or effective connectivity |
| ☐ | ☐ Graph analysis |
| ☐ | ☐ Multivariate modeling or predictive analysis |

**Functional and/or effective connectivity**

*Report the measures of dependence used and the model details (e.g. Pearson correlation, partial correlation, mutual information).*

**Graph analysis**

*Report the dependent variable and connectivity measure, specifying weighted graph or binarized graph, subject- or group-level, and the global and/or node summaries used (e.g. clustering coefficient, efficiency, etc.).*

**Multivariate modeling and predictive analysis**

*Specify independent variables, features extraction and dimension reduction, model, training and evaluation metrics.*

