## [Peer Review File · Nature Ecology & Evolution]

Global covariation of forest age transitions with the net carbon balance

Corresponding Author: Dr Simon Besnard

Version 0:

Decision Letter:

20th September 2024

Dear Dr Besnard,

Your Article, "Global covariation of forest age transitions with the net carbon balance" has now been seen by 3 reviewers. You will see from their comments copied below that while they find your work of considerable potential interest, they have raised quite substantial concerns that must be addressed. In light of these comments, we cannot accept the manuscript for publication, but would be very interested in considering a revised version that addresses these serious concerns.

We hope you will find the reviewers' comments useful as you decide how to proceed. If you wish to submit a substantially revised manuscript, please bear in mind that we will be reluctant to approach the reviewers again in the absence of major revisions.

If you choose to revise your manuscript taking into account all reviewer comments, please highlight all changes in the manuscript text file.

* Include a "Response to reviewers" document detailing, point-by-point, how you addressed each referee comment. If no action was taken to address a point, you must provide a compelling argument. This response will be sent back to the referees along with the revised manuscript.

* If you have not done so already we suggest that you begin to revise your manuscript so that it conforms to our Article format instructions at <http://www.nature.com/natecolevol/info/final-submission>. Refer also to any guidelines provided in this letter.

Link Redacted

If you wish to submit a suitably revised manuscript we would hope to receive it within 6 months. If you cannot send it within this time, please let us know. We will be happy to consider your revision so long as nothing similar has been accepted for publication at Nature Ecology & Evolution or published elsewhere.

Nature Ecology & Evolution is committed to improving transparency in authorship. As part of our efforts in this direction, we are now requesting that all authors identified as 'corresponding author' on published papers create and link their Open

Researcher and Contributor Identifier (ORCID) with their account on the Manuscript Tracking System (MTS), prior to acceptance. This applies to primary research papers only. ORCID helps the scientific community achieve unambiguous attribution of all scholarly contributions. You can create and link your ORCID from the home page of the MTS by clicking on 'Modify my Springer Nature account'. For more information please visit www.springernature.com/orcid.

Thank you for the opportunity to review your work.

[redacted]

Reviewer expertise:

Reviewer #1: remote sensing, carbon cycle

Reviewer #2: remote sensing, biomass modelling, forest dynamics

Reviewer #3: carbon cycle, land use, remote sensing

Reviewers' comments:

Reviewer #1 (Remarks to the Author):

General comments:

Although the importance of forest age structure in quantifying regional and global forest carbon balance has long been recognized, quantitative assessment of this importance has been lacking. Making use of a global forest age map series at the 100-m resolution in conjunction with a global biomass map and atmospheric inversion results of the land carbon flux, you conducted a comprehensive analysis on the carbon stock and the growth rate of forests in various age classes at the global scale, and attempted to establish the relationship between forest age transitions (mainly from forests in various age classes to stand-replaced young forests) and the net carbon sink. Although similar results are already available at regional scales (see comment 2 below), such results at the global scale may be considered as a large step forward. In this regard, this paper may have substantially advanced our understanding of the global forest carbon cycle to deserve publication in *Nature Ecology and Evolution*. The manuscript is mostly clearly written and well illustrated with good quality figures. However, there are some key points that prevented me from a sound assessment of the results and there are some improvements to be made to the key phrases used.

1. Figure 3a shows the key result of your study. However, it is not clear how you obtain the data points in this figure. I understand that the "stand replacement extent" on the horizontal axis is obtained from the forest age maps in 2010 and 2020, but I don't understand how the "changes in NEE" are obtained. You seem to imply that each point is the mean of 9 atmospheric inversion results, but the regression lines of the individual inversion models are mostly below the points, suggesting that the points are not from inversion results. NEE could be from estimates based on biomass change, but NEE should also include soil carbon change. I am therefore puzzled by these results. Although it is not clearly written in the text, I guess that atmospheric inversion is the only source for your NEE estimates. Also, the fact that the responses of inverted NEE by individual models to stand replacement differ so greatly gives a very low confidence in the relationship that you established in this figure after some steps of averaging processes. In essence, atmospheric inversion is typically unreliable at high spatial resolutions as the reliable ground-based CO₂ observations are sparse and/or satellite column observations have large uncertainties and transport errors increases greatly with increasing spatial resolution. Stand-replacing disturbances often occur in small patches. The scale mismatch issue could have played a large role in the relationship given in Figure 3a, although you have made great efforts in spatial matching of the various datasets.

2. There has been a fair amount of literature on forest growth curves (NPP vs age) and carbon sinks at different ages (NEP vs age) for various regions of the world. These results could be used to strength or evaluate your findings. Some of them are:

- Chen, J. M., W. Ju, J. Cihlar, D. Price, J. Liu, W. Chen, J. Pan, T. A. Black, and A. Barr, 2003. Spatial distribution of carbon sources and sinks in Canada's forests. *Tellus B*, 55(2): 622-642.
- Wang, S., Zhou, L., Chen, J. M., et al. (2011). Relationships between net primary productivity and stand age for several forest types and their influence on China's carbon balance. *Journal of Environmental Management* 92, 1651-1662, 10.1016/j.jenvman.2011.01.024.
- He, L., Chen, J.M., Pan, Y., et al. (2012). Relationships between net primary productivity and forest stand age in U.S. forests. *Global Biogeochemical Cycles* 26, 2010GB003942, 10.1029/2010GB003942.
- Pan, Y., J. M. Chen, R. Birdsey, K. McCullough, L. He, and F. Deng. 2011. Age Structure and Disturbance Legacy of North American Forests. *Biogeosciences*, 8: 715-732.

Minor comments:

- Figure 3c: The meanings of the pairs of grey and green columns are not clear to me. For example, what does the "Maturing stand-replaced" AGC stock mean? Does it mean the AGC stock of replaced stands growing to the maturing age or the AGC stock of replaced-stands from maturing forests? It is likely the latter. In this case, why the stocks of replaced stands are so different among the age classes. I can't find any explanation for this.
- Figure S2 caption: "within a 2x2 spatial window". What do you mean by 2x2? In degrees or 2 pixels by 2 pixels?

- Figure S3: It would be clearer to label the color bars with “forest age change” rather than “forest age”.
- Figure S5 and Figure S9: I guess “stand-replaced mature forests” means mature forests replaced by young stands, but it could also mean replaced stands ageing to mature forests. I wonder if “young stands replacing mature forests” is exactly what you mean. If so, similar changes would be required throughout the paper including similar phrases for young, maturing and old growth forests.
- Figure S8: I am concerned by the large contrast between managed and unmanaged forests in terms of the change of NEE due to stand replacement. Although the reasons, such as diverse forest species and ages, are given in the main text, the reality is that plantations would generally establish forests stands faster than natural regeneration although they have less shrubs and grass, resulting in similar sinks as the unmanaged forests. I wonder if your data processing stream and/or the large uncertainty in the atmospherically inverted carbon flux have anything to do with this large contrast.
- Figure S11: Similar to Figure 3c, I could not understand the pairs in orange and in green. What does the “Maturing stand-replaced” annual growth rate mean? Also, in about half the cases in this figure, the annual growth rates in different age classes do not follow the unusual patterns of forest growth curves, i.e. initial increase, reaching a peak and then gradual decrease, as reported in the papers given in major comment 2. This raises an alarm for me on the data quality.
- Figure S12: Because I can not understand the phrases like “stand-replaced mature forests”, I could not understand the content of this figure. For example, if “stand-replaced old growth” means the old growth that is replaced by young stands, then why does carbon stock mostly increase? If it means replaced stands growing to old growth (we would not have data for that), then why do we have negative changes in the carbon stock.

Reviewer #2 (Remarks to the Author):

General comments

The manuscript is based on global modeling of forest age from a modeled biomass map and environmental data at 100m. Changes in forest age are then assessed and classified based on difference in modeled value between 2010 and 2020. These changes are analyzed spatially, and compared to provide insight into carbon fluxes, recent forest management practices, and two different future scenarios regarding hypothetical forest management pathways.

The findings can be partitioned into two groups: 1) relating to the modeled forest age and spatial patterns, etc. and 2) the impact of age changes in carbon stocks and fluxes. In my opinion, the first group of findings is more interesting (spatial distribution of modeled ages, etc.), with a large caveat that I have serious reservations about the premise that something as complex as the age distribution of thousands of trees within a pixel a) could be represented by a single number and b) that number could/should be modeled from biomass. The predicted age maps look very similar to global maps of biomass (regardless of remote sensing platform used to create the maps, e.g. CCI, GEDI, Harris et al 2020, etc.). That reservation aside, the use of forest structure as a proxy for forest longevity is highly relevant in literature today, and this work is an interesting addition, even if performed at a coarse, global scale.

The second group of findings (impact on changes in forest age / disturbances on carbon stocks and fluxes) seems intuitive and to my knowledge does not provide new information in this field. While important, the finding that an increase in stand replaced forest fraction is correlated with an increased C source and/or decreased C sink doesn't seem to add new information. Perhaps I am missing the significance of these findings, and if so, the authors should do more to highlight the importance of this second group of findings.

Comments:

Please reconsider the term ‘old-growth’, which is a loaded term that usually has additional constraints in addition to the ‘age’ of a forest (e.g. structural or functional characteristics, little-no human impact, etc.). Since age is the only factor considered when classifying forests, the term ‘old’ should suffice. In other words, there needs to be far more nuance attributed to identifying “old-growth” than a single modeled age value that does not reflect the distribution of tree ages within a 100m pixel or mean 1° aggregation. High biomass does not necessarily mean old-growth, nor does low-biomass necessarily mean young forest.

This work relies heavily on the GAM, and some updates to the product were made to create an updated version GAMv2. A clearer overview of that product (~line 520 in methods) would be helpful in the interpretation of lines 520-567. Furthermore, lines 529-550 are difficult to understand without intimate knowledge of the GAM product and Landsat products. For example, as written, it states “The Hansen data determines forest loss years, subtracted from 20 to obtain the time since the last disturbance in 2020.” I understand the authors subtracted 20 from the Hansen lossyear estimates to obtain the time since disturbance, but this is not what is written. The Hansen data represents the year of forest lost after 2000, and the authors the subtracted this value from 20. Another example, in line 542, its unclear what the authors are referring to by ‘the landsat-based estimates’. Is it the Hansen lossyear, or the time since last disturbance layer explained in line 529? It seems it's the time since last disturbance layer, but again this isn't explicitly stated. The justification for items 1-3 (lines 542-550) is missing, and difficult to infer. Thus, a clear and precise explanation of the methods in this section would help readers understand the improvements made in the creation of GAMv2.

The framing of “gradually ageing” is odd to me, as well as a “rapid transition” in stand replacing forests from young growth to mature stages. Aging happens at the same rate for any tree or forest, in that time doesn't pass at different rates in different

pixels. Is a “gradual ageing” forest one that did not experience a disturbance is this has an age different of 10 years between 2010 and 2020? Similarly, growth (height, biomass, etc.) can be rapid in young forests, but aging can’t occur faster in young forests after a disturbance than in the “gradual ageing” forests. Two factors seems to be adding to confusion here:

1) Growth rates and ageing rates seem to be conflated here, or, if this is not the case, then the distinction between the two is not well explained by the authors. Growth (an increase in some forest attribute, e.g. biomass) can vary based on age, as noted throughout the paper and in table S1. Growth seems to be confused with a change in age, or “ageing”. The rate of ageing of forests, regardless if its immediately after a disturbance, or in the ‘gradually ageing’ class that did not experience a disturbance, does not change, because time passes at a constant rate. This is complicated, however, because in GAMI, age is modeled from biomass. While the GAMI product itself isn’t the main focus of this work, I have concerns with the premise of modeling forest age from outputs from a modeled biomass surface. (I’ve expressed these concerns above).

2) The methods section “Partitioning stand-replaced and gradually ageing forests” does not make sense as explained, or at least context is missing. An age difference of 10 years means no disturbance, which is straightforward. If there was a stand replacing disturbance between 2010 and 2020, the age difference would be then be negative, as the 2020 age would be less than the 2010 age. So, if there are other possibilities outside of 1) a 10 year age difference between 2010 and 2020 or 2) a negative difference, indicating the 2020 age is less than the 2010 age due to disturbance, then it is not clear or well explained how this could be possible. If not, this section could be more intuitively explained. Further, for the 1° resampling, a mean different of less than 10 years really represents two things: the fraction of pixels that were disturbed than thus the 2020 age is less than the 2010 age, and the time of the disturbance and thus how much regrowth was able to occur between the time of disturbance and 2020. If this interpretation is correct, it should be explicitly explained in the text. If not, the section needs to be clarified.

South America tropical is listed as a region with “more prevalent” stand replaced forests and gradually aging forests, which seems at least counter intuitive, and potentially incorrect, if these two forests class fractions sum to 1.

Reviewer #2 (Remarks on code availability):

I'm not proficient enough in python to provide a meaningful or fair assessment of the code.

Reviewer #3 (Remarks to the Author):

This manuscript presents a study that combines empirical estimates of forest age with top-down NEE estimates, revealing a significant positive correlation between the fraction of stand-replaced forests and the inversely derived trend in carbon sink strength from 2010 to 2020 on a global scale. The research addresses critical and complex topics in the carbon cycle, particularly in light of the significant impact of land-use activities ongoing. Two elements in this study are quite challenging to quantify: forest age and NEE, of which uncertainties could affect the conclusion made.

The first concern is the uncertainty associated with both forest age and CO₂ flux estimates. Forest age is a complex term, and its definition depends on the context. In this manuscript, I believe the definition of age in this study appears to be related to carbon, so providing a clear and precise definition is still crucial. However, accurately estimating forest age to differentiate between age classes with a 20-year interval is challenging due to the substantial uncertainties in biomass data used for age estimation. While this uncertainty does not invalidate the analysis, it is essential to include age validation results and account for this uncertainty when assessing the relationship between age and NEE (e.g., Fig 3). Additionally, the top-down inversion estimates themselves vary significantly, and uncertainty is a very large grid scale scale. This study incorporates nine estimates in an effort to address these uncertainties; however, the results seem highly dependent on the subset of estimates used. For instance, in Fig 3a, one inversion product shows a larger positive slope compared to the others, which heavily influences the overall trend represented by the dark line and is central to the paper's conclusion. Therefore, I recommend testing the robustness of the results by excluding this outlier to ensure the conclusions are not overly dependent on a single product.

My second major concern is the scale mismatch between the 100 m spatial resolution of the forest age GAMI v2.0 data and the 1-degree resolution of the top-down CO₂ fluxes. The method for identifying CO₂ flux by age class (e.g., Figure 3b) is not sufficiently explained. Since each 1-degree grid likely contains multiple age classes, it is unclear how the flux is partitioned among these classes. There is no clear explanation and evaluation of this process given the significant spatial heterogeneity of forest age at fine scales, needs to be addressed.

The third concern is the inherent influence of climate variability and other factors on the NEE estimates used to represent ~2010 and ~2020. It is unclear how the study isolates the effects of climate variability from ageing on NEE. This becomes even more problematic when considering projections for 2050, as major drivers such as CO₂ fertilization and warming are not explicitly considered in the projection. How these drivers affect the study's conclusions related to forest conservation. Is the estimate optimistic or conservative?

The fourth concern relates to the uncertainty in the projections for the two hypothetical scenarios. The manuscript assumes that the age distribution remains unchanged under the BAU scenario and, based on this, concludes that biomass is nearly

unchanged. However, this assumption is flawed. The analysis presented between lines 140-220 for the period 2011-2020 clearly shows that forest age shifts under current practices, which effectively represent BAU conditions. Additionally, forests continue to accumulate biomass within each age class, meaning the method used in this study likely underestimates biomass gain. A simple way to test the validity of the method is to check whether it can accurately reproduce AGB changes using ESA Biomass data. Including this validation, or at least acknowledging the potential limitations, would provide clearer insight into the reliability of the study's results.

My other specific comments are listed below:

In fig 1, the negative sign is missing in ">30". Additionally, it would be helpful to clarify what the negative values in fig 1c imply. Does this imply that forests are getting younger due to compositional changes, with a higher proportion of younger forests emerging?

In fig 3a, how robust is the trend shown by the black line, considering the outlier with a steep positive slope? Since this outlier seems to have a significant influence on the overall trend, testing the sensitivity of the black line by excluding this outlier would provide greater confidence in the robustness of the conclusions. This sensitivity test is crucial to ensure that the results are not overly dependent on a single inversion product.

For fig 3b, it is unclear whether the age class is based on 2011 or 2020. If it is based on 2011, why isn't there a positive NEE (carbon source), which is expected due to legacy emissions from previous disturbances? This also ties back to my major concern regarding the method used to derive NEE by age class. How was the NEE by age classes derived? The CO₂ flux at one-deg contains multiple age classes.

In P4, L55, it is unclear how Figure 3b supports the statement being made.

In P9, L273-274, the statement "younger forests typically have lower net CO₂ uptake because of emissions related to stand-replacement (Fig. 3b)" is unclear. Does this refer to legacy emissions (CO₂ losses from delayed tree mortality, deadwood decomposition, and soil respiration) from themselves, or to immediate emissions from the aboveground biomass of forested areas surrounding younger forests?

L9, L277, the data source of unmanaged and managed layers is missing.

In P9, L284-285, the statement, "The natural regeneration and diverse age structure in these forests may help absorb the impacts of such disturbances without significantly affecting net CO₂ fluxes," raises a question about the role of forest tree cover. To what extent is it due to the high tree cover in unmanaged forests? The classification of unmanaged versus managed forests seems to correlate with tree cover. This suggests that, at the spatial scale of Figure 3a and S8, grids classified as unmanaged forests may simply have higher tree cover, which leads to greater carbon uptake, thereby offsetting losses from stand-replacement and making the slope insignificant. This implies that the result may not necessarily be due to a diverse age structure.

In P11, L374-383, I am not sure how the carbon stocks partition among age classes from L370-383 is related to the former part of this paragraph. And also, providing stock changes over the interval is also very important.

In P11, L387, how exactly was the BAU scenario defined? Assuming the current rate of deforestation continues? Providing a clearer definition or outlining the key assumptions used to establish the BAU scenario is very necessary.

In P18, L511, it is unclear what "mean" refers to. Mean over what (time or space)? Additionally, how is the standard deviation of aboveground biomass (AGB) derived? Is it calculated over space, and if so, why? Does the spread in fig 3c reflect both spatial variation across pixels within the group as well as AGB variation due to perturbations?

In P19, L537, it is unclear how the gain layer was used in the age estimation process. As I know that this data does not specify when the gain occurred, so how was this information utilized? Was the gain layer just used to provide a gridded fraction, or for another purpose?

In P19, L544, the method for estimating "forest age from Landsat-based estimates" is unclear. Are you referring to the "time since disturbance" layer? If so, it's important to note that this measure does not necessarily reflect the year the forest began to recover.

In P19, L565, it is unclear how the uncertainty data were incorporated into the analysis, particularly in Fig 3a. Could you clarify how these uncertainties were factored into the results, and how they influenced the trends shown in the figure?

In P20, L600-615, what was the consideration for not using the Hansen forest loss and gain layers, which are closer to direct observation, and instead opting for modeled forest age? How consistent are these two approaches? Are there areas where the Hansen data do not indicate loss or gain, but the model shows an age change of less than 10 years, resulting in classification as stand-replaced forests?

In P21, L630-640, the method for calculating NEE by age class is not clearly explained. How is the "ratio between the fraction of stand-replaced and gradually aging forests" related to the age classes shown in fig 3b?

Version 1:

Decision Letter:

12th May 2025

Dear Dr Besnard,

Your manuscript entitled "Global covariation of forest age transitions with the net carbon balance" has now been seen by two of the original three reviewers. Their comments are attached. Thank you for your patience while we gathered these reviews. We had been waiting to hear back from Reviewer 3 but haven't heard from them, and have decided we can proceed with the two reviews we have.

The reviewers find that the manuscript has improved substantially in revision, but they still have some concerns which will need to be addressed before we can offer publication in Nature Ecology & Evolution. We will therefore need to see your responses to these comments, along with a revised manuscript, before we can reach a final decision regarding publication.

We therefore invite you to revise your manuscript taking into account all reviewer comments. Please highlight all changes in the manuscript text file.

* If you have not done so already please begin to revise your manuscript so that it conforms to our Article format instructions at <http://www.nature.com/natecolevol/info/final-submission>. Refer also to any guidelines provided in this letter.

* Extended Data Figures - please ensure that any supplementary figures and tables that are crucial to the manuscript's conclusions are converted into Extended Data figures and tables to increase visibility of these data. Extended Data figures and tables are online-only (present in the online PDF and full-text HTML versions of the paper), peer-reviewed display items that provide essential background to the article but are not included in the main article due to space constraints. A maximum of ten Extended Data display items (figures and tables) is permitted.

Link Redacted

Nature Ecology & Evolution is committed to improving transparency in authorship. As part of our efforts in this direction, we are now requesting that all authors identified as 'corresponding author' on published papers create and link their Open Researcher and Contributor Identifier (ORCID) with their account on the Manuscript Tracking System (MTS), prior to acceptance. ORCID helps the scientific community achieve unambiguous attribution of all scholarly contributions. You can create and link your ORCID from the home page of the MTS by clicking on 'Modify my Springer Nature account'. For more information please visit www.springernature.com/orcid.

[redacted]

Reviewer comments:

Reviewer #1 (Remarks to the Author):

General comments:

I appreciate your great effort in revising your manuscript according to comments from all three reviewers, in particular you conducted a large amount of work in recomputing some of the results in response to my concerns. My main concerns are mostly well addressed, and your manuscript is much improved.

However, the new results (Figure 3) regarding to NEE based on atmospheric inversion models become quite different from previous results in the original manuscript. For example, NEE from most models now decreases (meaning increasing sinks) with stand replacement extent (Figure 3c) instead of increasing in the previous result. You provided an explanation for this new and different result, which is reasonable as younger stands could indeed grow faster than the replaced old stands, although AGB of younger stands is smaller. However, this large change in the results due to different ways in manipulating the same data causes concern on the reliability of the inversion results for the purpose of your analysis. A more problematic issue is the set of results in Figure 3d, which shows very different responses of NEE to stand replacement extent for different age cohorts. In particular, the responses between maturing (decrease) and mature (increase) age cohorts to stand replacement is opposite, while the old forests show a decreasing trend. Logically, we would expect the mature cohort would be a case between maturing and old forest cohorts. Although you provided a long explanation to these results in a speculative tone (lines 332-349), I feel the explanation is faint and unconvincing. While I fully appreciate your great effort in pooling the data together and producing these new outputs, I feel that parts of the results are too uncertain to publish. I would suggest to scrub all results from the atmospheric inversion models, but it would water down the content of the paper significantly. As a compromise, Figure 3c may be retained if you have some confidence on the results, but the uncertainty associated with NEE results should be emphasized.

Minor edits:

L80-82: add "to" to the three objectives;

L91: change "increase" to "increased";

L291: change "slightly" to "substantially"

Reviewer #2 (Remarks to the Author):

Overall, the authors have done a sufficient job clarifying various aspect of the manuscript. Below are response to my original comments, although I have read the responses to the other reviewer's comments as well and feel the authors have done an adequate job in general responding to all reviews.

Comments in response to the Authors' rebuttals to Reviewer 2's comments:

Page 14 major comment 1: Excellent, thank you.

Page 14 major comment 2: The methods section on GAMI starting on line 523 in the revised manuscript version reads much more clearly. Furthermore, fig S15 that shows non-linear, and more importantly, non-monotonic relationships between biomass and age is highly important, and in my opinion a marked improvement over existing age products at coarser resolutions. While I understand the perspective and audience of this work is primary for the global modeling and carbon dynamics community, there is a lot of skepticism regarding the validity and utility of mean forest age mapping in the ground-based forest ecology community. These results demonstrate an advance in global-scale representations of biomass as a function of age that are more in line with ground-based observations, which is highly promising. If appropriate, the authors should consider highlighting this result more than is currently done in the manuscript.

Page 15 major comment 3: This is very helpful, and alleviates most of my confusion. I would recommend the authors be explicit and precise in this area of the manuscript however (e.g bottom panels in table 2 of revised manuscript, i.e. "forests experience a mean age difference of ..." and "a uniform ten-year mean age increase ..."). When talking about the forest age transitions at 1 degree, please be explicit you're referring to the mean age from the higher resolution data.

Page 15 major comment 4: The methods section starting on line 596 is much improved – the distinction between aging and growth is well explained and the authors do a good job giving context to how variations in modeled age are primarily driven by growth (structural, ecological changes). I suggest some form of the information communicated in lines 609-613 belongs in the main body of the manuscript, perhaps somewhere in the discussion of age transitions (130-160). Above, in the response to comment 2 on pg. 14, I comment on figure S15, which I think is a fantastic figure that goes a long way towards bolstering the validity of the GAMI v2 product. I appreciate the responses to comments #2 and #4 and believe these improvements have increased the rigor and merit of the manuscript.

Page 17 major comment 5: This is helpful explanation, but I still have questions: How were the data products referenced in lines 634-636 integrated into the analysis? It seems to me quantifying the contribution of both factors (the magnitude of age difference vs the number of pixels with an age change) would be valuable information to report and use to contextualize results. Perhaps this is what is being referenced on lines 625-627, but it isn't clear how these products feed into the analysis or results.

*****END*****

Version 2:

Decision Letter:

11th June 2025

Dear Simon,

Thank you for submitting your revised manuscript "Global covariation of forest age transitions with the net carbon balance" (NATECOLEVOL-24082130B). It has now been seen again by the remaining reviewer, whose comments are below. In light of this reviewer's final comments, and those of the other reviewers previously, we are happy in principle to publish it in Nature Ecology & Evolution, pending minor revisions to satisfy the reviewers' final requests and to comply with our editorial and formatting guidelines.

If you have not done so already, please ensure that you also email us completed copies of the Reporting summary and Editorial policy checklists:

Reporting summary: https://www.nature.com/documents/nr-reporting-summary.pdf
Editorial policy checklist: https://www.nature.com/documents/nr-editorial-policy-checklist.pdf

[redacted]

Reviewer #1 (Remarks to the Author):

Your effort in addressing my concerns is much appreciated. I recommend your manuscript be accepted for publication.

Major Comments

1. **Comment:** Figure 3a shows the key result of your study. However, it is not clear how you obtain the data points in this figure. I understand that the “stand replacement extent” on the horizontal axis is obtained from the forest age maps in 2010 and 2020, but I don’t understand how the “changes in NEE” are obtained. You seem to imply that each point is the mean of 9 atmospheric inversion results, but the regression lines of the individual inversion models are mostly below the points, suggesting that the points are not from inversion results. NEE could be from estimates based on biomass change, but NEE should also include soil carbon change. I am therefore puzzled by these results. Although

it is not clearly written in the text, I guess that atmospheric inversion is the only source for your NEE estimates. Also, the fact that the responses of inverted NEE by individual models to stand replacement differ so greatly gives a very low confidence in the relationship that you established in this figure after some steps of averaging processes. In essence, atmospheric inversion is typically unreliable at high spatial resolutions as the reliable ground-based CO₂ observations are sparse and/or satellite column observations have large uncertainties and transport errors increases greatly with increasing spatial resolution. Stand-replacing disturbances often occur in small patches. The scale mismatch issue could have played a large role in the relationship given in Figure 3a, although you have made great efforts in spatial matching of the various datasets.

Response: We appreciate the reviewer's careful examination of Figure 3a and their request to clarify the derivation of NEE changes. We confirm that the NEE changes in this analysis are derived exclusively from atmospheric inversions, not from biomass-based estimates. The reason why the ensemble mean appears above the regression lines of individual inversion models is due to the method initially used to compute the ensemble values. In our original approach, we first created an ensemble map of NEE by averaging the inversion products spatially at one degree before extracting data points for the figure. A more appropriate approach is first to extract data points from each inversion product and then compute the mean or median at each data point, ensuring that the ensemble better reflects the distribution of model outputs. We acknowledge this oversight and have corrected our approach to maintain consistency with the individual inversion products. The revised analysis yields an ensemble trend that is more centred among the regression lines of individual models (see new Fig. 3c).

Additionally, we acknowledge that our previous approach, which relied on binning the ratio of stand-replacement to gradually ageing forests at the one-degree pixel level, may not have been the most appropriate way to assess the relationship between stand-replacement and NEE trends. The binning process inherently disrupted the spatial coherence of regional fluxes, potentially introducing inconsistencies in how the inversion models were represented across different bins. Furthermore, we may have overlooked critical spatial variations in NEE responses by averaging across all models within each bin. To improve the robustness of our analysis, we have now adopted a spatial window approach, where we extract data within coarse spatial windows (e.g., 2°×2°, 5°×5° or 10°×10° regions) before performing the regression analysis. This ensures that the comparison between stand-replacement and NEE changes remains spatially coherent within each inversion model. By maintaining the spatial structure of the inversion data, this approach avoids the potential distortions introduced by binning.

Furthermore, given the variability in responses across different atmospheric inversion models, we recognise the reviewer's concerns regarding the robustness of previously

reported relationships between stand-replacement and NEE. We agree that our previous approach made it difficult to interpret the inferred relationships confidently. To address this, we have conducted a more systematic and rigorous analysis, explicitly investigating the relationship across all individual NEE inversion members rather than relying solely on ensemble means.

To assess the stability and robustness of the observed trends, we have applied Jackknife resampling, systematically excluding individual NEE members one at a time. Unlike our previous approach, which considered the total stand-replaced fraction, our revised analysis now partitions stand replacement by age class. This refinement shows that when a significant relationship is found, regions with a higher fraction of stand-replaced old forests consistently exhibit an increasing carbon sink across different NEE ensemble members (see Figure 3c and Figure S16 in revised manuscript). In contrast, other age classes show no strong or systematic correlation with NEE trends. These results indicate that the loss of old forests and their replacement with young stands play a crucial role in shaping atmospheric CO₂ fluxes. Moreover, our revised approach demonstrates that the observed correlation is not merely an artefact of averaging but a robust and persistent signal across multiple inversion datasets.

Additionally, we acknowledge the reviewer's concerns regarding scale mismatches between atmospheric inversions and stand-replacing disturbances, which often occur at finer spatial scales than those resolved by inversions. To mitigate potential biases from this mismatch, we have conducted a sensitivity analysis aggregating stand-replacement extents and NEE changes to coarser spatial windows (e.g., 2°x2°, 5°x5°, and 10°x10° resolution) and assessing how this affects the inferred relationships (Fig. S17). We concluded that our current approach does not suffer strongly from this bias at coarser resolutions. Still, we acknowledge that we could not test this for finer resolutions at which replacement occurs.

Combining these spatial aggregation tests and Jackknife resampling, we ensure our conclusions are not overly dependent on resolution mismatches, individual models, or stand-replacement estimates.

Revisions made:

- We updated Figure 3 and its caption.
- We added Fig S16, which replicates Fig 3c for each age class.
- We added Fig S17 to the supplements, showing how the relationship is impacted using different spatial window aggregation.
- After re-running the analysis behind Fig 3a (now Fig 3c in revised manuscript) more robustly, the results section on the **Influence of forests stand-replaced by**

younger stands on global carbon sink capacity has been completely rewritten to match the updated results of Figure 3. (L256-366)

- We have also updated the method section related to this analysis: *“To assess the covariation between forest age shifts and net CO₂ flux changes, we analysed the relationship between stand-replacement fractions and changes in NEE derived from atmospheric inversions. To ensure the robustness of our analysis, we extract data within spatial windows of different resolutions (i.e., 2°×2°, 5°×5° and 10°×10°) before performing the regression analysis. This spatial window approach ensures that stand-replacement fractions and NEE changes remain spatially coherent within each inversion model. We compute the median stand-replacement fraction (20 members) and corresponding median NEE changes across all available inversion models (9 members) for each spatial window. To assess the robustness of our results, we apply Jackknife resampling, systematically excluding individual NEE inversion members to test their influence on the overall trend.”* (L660-669).
- Based on these new results, our analysis's abstract (L18-34) and conclusions (L 410-432) have been slightly re-written.

Fig. 3. Above ground carbon stocks across forest age classes, distinguishing between stand-replaced forests and undisturbed ageing forests, expressed per unit area at a one-degree pixel level. The stand-replaced categories represent the AGC stock of forests at a given age class (Young, Maturing, Mature, or Old) before stand replacement. In (b), net carbon changes for stand-replaced forests across different forest age categories are shown. In (a) and (b), the median values from the 20 biomass realisations are shown and that the spread represents the spatial variation within a given age class. In (c), the relationship between the fraction of old forests replaced by young stands (i.e., stand-replacement extent) and changes in NEE from inversions between circa 2020 (average of 2019–2021) and 2010 (average of 2009–2011) is shown. The dark solid line represents the linear regression on the ensemble estimates, while the dashed grey lines indicate the regressions for the nine individual atmospheric inversion models. Similar relationships are shown in (d) for the four age classes based on the median ensemble estimates.

To smooth the spatial distribution of net CO₂ fluxes, we applied a Gaussian filter (length = 500 km, equivalent to approximately four one-degree pixels). This smoothing technique reduces noise in the data and helps minimise the influence of local transport errors.

Fig. S16. Relationship between the fraction of forests replaced by young stands (i.e., stand-replacement extent) and changes in net CO₂ fluxes between circa 2020 (average of 2019–2021) and 2010 (average of 2009–2011) for young (a), maturing (b), mature (c) and old (d) stand-replaced forests. The dark solid line represents the linear regression on the ensemble estimates, while the dashed grey lines indicate the regressions for the nine individual atmospheric inversion models. To smooth the spatial distribution of net CO₂ fluxes, we applied a Gaussian filter (length = 500 km, equivalent to approximately four one-degree pixels). This smoothing technique reduces noise in the data and helps minimise the influence of local transport errors.

Fig S17. Relationship between the fraction of old forests replaced by young stands (i.e., stand-replacement extent) and changes in net CO_2 fluxes between circa 2020 (average of 2019–2021) and 2010 (average of 2009–2011) using different spatial windows: none (a), 2x2 degree (b), 5x5 degree (c) and 10x10 degree (d). The dark solid line represents the linear regression on the ensemble estimates, while the dashed grey lines indicate the regressions for the nine individual atmospheric inversion models. To smooth the spatial distribution of net CO_2 fluxes, we applied a Gaussian filter (length = 500 km, equivalent to approximately four one-degree pixels). This smoothing technique reduces noise in the data and helps minimise the influence of local transport errors.

- Comment:** There has been a fair amount of literature on forest growth curves (NPP vs age) and carbon sinks at different ages (NEP vs age) for various regions of the world. These results could be used to strength or evaluate your findings.

Response: We thank the reviewer for these suggestions and have included citations to the recommended studies (e.g., Chen et al., Wang et al.) to strengthen our discussion and evaluate our findings in the context of prior regional-scale research.

Revisions made:

- References and discussion were added in the Introduction and Discussion sections. For instance, we have added the reference to Pan et al. (2024) in L42 as well as Wang et al. (2011) and He et al. (2012) in Table 1.

Minor Comments

- 1. Comment:** Figure 3c: The meanings of the pairs of grey and green columns are not clear to me. For example, what does the “Maturing stand-replaced” AGC stock mean? Does it mean the AGC stock of replaced stands growing to the maturing age or the AGC stock of replaced-stands from maturing forests? It is likely the latter. In this case, why the stocks of replaced stands are so different among the age classes. I can’t find any explanation for this

Response: Thank you for your comment. We understand that the labels in Figure 3c were not entirely clear, and have revised the figure caption and manuscript text for clarity.

Revisions made:

- We have added an explanation in the figure caption to make it explicit that these stocks represent the AGC of forests that had reached their respective age classes before experiencing stand-replacing disturbances (L 476-490).
- Additionally, we have expanded the discussion to explain why the AGC stocks of replaced stands differ across age classes (L 262-268). This variation arises due to several factors:
 - Stand-replacement events do not always lead to complete biomass removal, as some trees may remain, preserving portions of the pre-disturbance AGC.
 - Old-growth forests that undergo stand-replacement may experience faster regrowth due to favourable conditions such as legacy trees, seed banks, and nutrient-rich soils, leading to higher residual AGC stocks than younger stand-replaced forests.

- 2. Comment:** Figure S2 caption: “within a 2x2 spatial window”. What do you mean by 2x2? In degrees or 2 pixels by 2 pixels?

Response: Thank you for your comment. We acknowledge that the description in the figure caption was unclear and have revised it for clarity.

Revisions made: In the figure caption, we have explicitly stated that "2×2" refers to a 2-degree × 2-degree spatial window.

3. **Comment:** Figure S3: It would be clearer to label the color bars with “forest age change” rather than “forest age”.

Response: Thank you for your suggestion. We agree that "forest age change" better represents the data shown in Figure S3 and have updated the colour bar title accordingly.

Revisions made: The colour bar title in Figure S3 has been changed from "forest age" to "forest age change" to more accurately reflect the data.

4. **Comment:** Figure S5 and Figure S9: I guess “stand-replaced mature forests” means mature forests replaced by young stands, but it could also mean replaced stands ageing to mature forests. I wonder if “young stands replacing mature forests” is exactly what you mean. If so, similar changes would be required throughout the paper including similar phrases for young, maturing and old growth forests.

Response: Thank you for your careful reading and insightful comment. We acknowledge that the terminology used for stand-replaced forests could be ambiguous and have revised it throughout the manuscript to ensure clarity.

Revisions made: We have explicitly reworded phrases such as "stand-replaced mature forests" to "young stands replacing mature forests" (and similarly for other age classes) to avoid ambiguity. These changes have been applied consistently throughout the text, figure captions, and legends, including Figures S5 and S9, to ensure a clear distinction between forests being replaced and forests ageing into a given age class.

5. **Comment:** Figure S8: I am concerned by the large contrast between managed and unmanaged forests in terms of the change of NEE due to stand replacement. Although the reasons, such as diverse forest species and ages, are given in the main text, the reality is that plantations would generally establish forests stands faster than natural regeneration although they have less shrubs and grass, resulting in similar sinks as the unmanaged forests. I wonder if your data processing stream and/or the large uncertainty in the atmospherically inverted carbon flux have anything to do with this large contrast.

Response: We appreciate the reviewer’s concern regarding the large contrast between managed and unmanaged forests regarding NEE changes due to stand replacement. Our revised analysis adopted a more spatially coherent approach to analysing the relationship between stand replacement and NEE trends. Unlike our previous method, which relied on binning, our updated methodology extracts data within spatial windows and evaluates relationships at a coarser scale while systematically assessing the robustness of results across different atmospheric inversion products.

With this refined approach, we no longer find a consistent or robust signal differentiating managed and unmanaged forests regarding NEE response to stand replacement. Given the substantial uncertainties in atmospheric inversion estimates and the complexities of disentangling climate-driven variability from stand-replacement effects, we have decided to remove this particular analysis. Instead, we now focus on the broader relationship between old forest stand replacement and NEE trends, where we find a more robust and consistent signal across multiple inversion products. We acknowledge that forest management practices, including plantation establishment and natural regeneration dynamics, influence carbon sequestration rates, but resolving these effects using current top-down CO₂ flux estimates remains challenging.

6. **Comment:** Figure S11: Similar to Figure 3c, I could not understand the pairs in orange and in green. What does the “Maturing stand-replaced” annual growth rate mean? Also, in about half the cases in this figure, the annual growth rates in different age classes do not follow the unusual patterns of forest growth curves, i.e. initial increase, reaching a peak and then gradual decrease, as reported in the papers given in major comment 2. This raises an alarm for me on the data quality.

Response: We appreciate the reviewer’s concerns regarding Figure S11. The term “*Maturing stand-replaced*” annual growth rate refers to the growth rate of forests that have undergone stand replacement and are now maturing. To clarify how these growth rates are estimated, we refer the reviewer to Equations 2 and 3 in the Methods section. Regarding the observed growth patterns, the orange pairs correspond to forests replaced by younger stands. These young forests generally exhibit higher growth rates than the older, gradually ageing forests (shown in green). Since stand-replaced forests are, by definition, young, the different age classes in this figure represent forests with varying pre-disturbance ages rather than a temporal sequence of forest development. It is important to note that this figure presents annual growth rates at a regional scale (TRANSCOM-Land regions) and is derived from spatial analysis rather than a traditional chronosequence. Consequently, a direct comparison with classical forest growth curves—where growth initially increases, reaches a peak, and then gradually declines—is not entirely appropriate. In contrast, naturally ageing forests (i.e., those unaffected by stand replacement) may exhibit trends more closely aligned with traditional growth trajectories across age classes. Some degree of variability in the reported growth rates is expected due to the heterogeneous nature of forests across regions, differences in environmental conditions, and inherent limitations of remote sensing-based observations. To further support confidence in data quality, we have added a supplementary plot (see Fig. S17) comparing the biomass-age relationships derived from the GAMI dataset with those from ESA-CCI biomass data across different climate zones. While we acknowledge a degree of circularity, as the GAMI dataset implicitly incorporates biomass data signals

(see Methods), the results provide ecological consistency. Specifically, we observe an expected increase in biomass with forest age, reaching a saturation point in older forests. However, biomass data tend to overestimate biomass at low biomass levels, which can introduce some uncertainty in young forest estimates (e.g., boreal temp. Driven, tropical). Additionally, integrating Landsat disturbance data into GAMI involves aggregating 30m resolution disturbance information to 100m biomass pixels by assuming that a 100m pixel has been stand-replaced if most of its underlying 30m pixels were disturbed. As a result, some residual biomass may remain within the 100m pixel, potentially affecting biomass-age relationships in recently disturbed areas. Despite these sources of variability, the overall trends align with ecological expectations.

Fig. S15. Relationship between biomass and forest age across different hydro-climate zones, derived from the GAMI dataset and ESA-CCI biomass (v4) data for 2020. The figure compares biomass accumulation patterns with increasing forest age, highlighting broad ecological trends. While the GAMI dataset inherently includes biomass

information from the ESA-CCI products (see Methods), this comparison confirms consistency of observed growth patterns.

Revisions made:

- Updated the caption of Figure S11 to “*Annual growth rates across age classes for forests replaced by young stands (in orange) and undisturbed ageing forests (in green) in each TRANSCOM-Land region. The annual growth rate represents the net change in biomass per year. Stand-replaced forests (orange) correspond to areas where older stands have been replaced by younger forests, which generally exhibit higher growth rates due to their early successional stage. In contrast, undisturbed ageing forests (green) represent forests that continue ageing without stand replacement, typically showing lower growth rates as they approach maturity.*”
- The supplements were supplemented with a figure (Fig S15) showing the relation between forest age and biomass across hydro-climatic space derived from GAMI and ESA-CCI biomass products, respectively.

7. **Comment:** Figure S12: Because I can not understand the phrases like “stand-replaced mature forests”, I could not understand the content of this figure. For example, if “stand-replaced old growth” means the old growth that is replaced by young stands, then why does carbon stock mostly increase? If it means replaced stands growing to old growth (we would not have data for that), then why do we have negative changes in the carbon stock.

Response: We appreciate the reviewer’s feedback regarding the terminology in Figure S12 and acknowledge that the phrasing may have been unclear. In this context, “*stand-replaced*” forests refer to areas where young stands have replaced forests due to disturbance. The classification into young, intermediate, mature, and old-growth forests (as shown in panels a–d) reflects the pre-stand-replacement age. Regarding the direction of carbon stock changes, the values in this figure represent carbon stock changes over time ($\text{gC m}^{-2} \text{ year}^{-1}$), which we express with the convention used for carbon fluxes by multiplying them by -1. As a result, positive values indicate carbon loss (biosphere-to-atmosphere flux, typically due to disturbance or decomposition), while negative values indicate carbon accumulation (net biomass growth). To improve clarity, we have revised the figure caption to explicitly state that these values represent carbon stock changes while the sign convention follows that of fluxes. We appreciate the reviewer’s suggestion and believe this adjustment enhances the interpretability of the

figure.

Revisions made:

- Updated the title of Figure S12 (now Figure S11 in revised manuscript) to *Spatial distribution of carbon stock changes (in $\text{gC m}^{-2} \text{ year}^{-1}$) in stand-replaced forests categorised by age before stand-replacement: (a) young (≤ 20 years), (b) intermediate (21–80 years), (c) mature (81–200 years), and (d) old (> 200 years) in 2010. Each pixel represents the median carbon stock change for 100m resolution pixels belonging to a specific forest age category, aggregated at a 1-degree scale. Positive values indicate carbon loss (biosphere-to-atmosphere flux) to maintain consistency with flux sign conventions, while negative values indicate carbon accumulation.*
- We also elaborated on how the carbon stock changes have been estimated in the method section: “Similarly, we calculated net carbon stock changes for stand-replaced forests by computing the difference in AGC stocks between 2010 and 2020. Specifically, for each spatial unit, the AGC stock 2020 was subtracted from the AGC stock 2010, yielding the net change over the decade ($\Delta\text{AGC} = \text{AGC}_{2020} - \text{AGC}_{2010}$). To express these changes in a flux-consistent sign convention, we multiplied the resulting values by -1 so that positive values represent carbon loss (biosphere-to-atmosphere flux, typically due to disturbance or decomposition). In contrast, negative values indicate carbon accumulation (biomass regrowth). This approach ensures consistency with commonly used flux representations while preserving the original stock change information” (L 646-654).

Reviewer #2

General response:

We sincerely appreciate the time and effort the reviewer has put into evaluating our manuscript. Their insightful comments and critical perspective on our approach and findings have been highly valuable in refining our analysis and improving the clarity of our arguments. We acknowledge the reviewer’s concerns regarding representing forest age as a single number within a pixel and the challenges associated with modelling age from biomass. We also appreciate the constructive feedback on interpreting carbon flux implications and the need to better articulate these findings’ novelty. In our responses below, we clarify our methodological choices, discuss the relevance of our approach in the context of existing literature, and make the necessary revisions to better highlight the significance of our contributions. We hope these improvements address the reviewer’s concerns and strengthen the manuscript.

Major Comments

- 1. Comment:** Please reconsider the term ‘old-growth’, which is a loaded term that usually has additional constraints in addition to the ‘age’ of a forest (e.g. structural or functional characteristics, little-no human impact, etc.). Since age is the only factor considered when classifying forests, the term ‘old’ should suffice. In other words, there needs to be far more nuance attributed to identifying “old-growth” than a single modeled age value that does not reflect the distribution of tree ages within a 100m pixel or mean 1° aggregation. High biomass does not necessarily mean old-growth, nor does low-biomass necessarily mean young forest.

Response: We appreciate the reviewer’s comment and acknowledge the need for more precise terminology. We have replaced the term *old-growth* with *old forests* throughout the manuscript, including figures and text, to reflect better the classification based solely on modelled forest age.

- 2. Comment:** This work relies heavily on the GAM, and some updates to the product were made to create an updated version GAMiv2. A clearer overview of that product (~line 520 in methods) would be helpful in the interpretation of lines 520-567. Furthermore, lines 529-550 are difficult to understand without intimate knowledge of the GAM product and Landsat products. For example, as written, it states “The Hansen data determines forest loss years, subtracted from 20 to obtain the time since the last disturbance in 2020.” I understand the authors subtracted 20 from the Hansen lossyear estimates to obtain the time since disturbance, but this is not what is written. The Hansen data represents the year of forest lost after 2000, and the authors the subtracted this value from 20. Another example, in line 542, its unclear what the authors are referring to by ‘the landsat-based estimates’. Is it the Hansen lossyear, or the time since last disturbance layer explained in line 529? It seems it’s the time since last disturbance layer, but again this isn’t explicitly stated. The justification for items 1-3 (lines 542-550) is missing, and difficult to infer. Thus, a clear and precise explanation of the methods in this section would help readers understand the improvements made in the creation of GAMiv2.

Response: We appreciate the reviewer’s feedback on needing a more precise explanation of the GAMI data construction process and the methodology behind GAMiv2. We have revised the relevant section (L523-574) to provide a more structured and detailed overview of the GAMiv2 product. We have clarified the time derivation since the last disturbance from the Hansen loss year data, ensuring the methodology is described unambiguously. Additionally, we have explicitly stated what “Landsat-based estimates” means and refined the justification for items 1-3 to improve readability and logical flow. These revisions aim to enhance the transparency and comprehensibility of the

methodology for readers unfamiliar with the GAMI product and Landsat-derived datasets.

- 3. Comment:** The framing of “gradually ageing” is odd to me, as well as a “rapid transition” in stand replacing forests from young growth to mature stages. Aging happens at the same rate for any tree or forest, in that time doesn’t pass at different rates in different pixels. Is a “gradual ageing” forest one that did not experience a disturbance is this has an age different of 10 years between 2010 and 2020? Similarly, growth (height, biomass, etc.) can be rapid in young forests, but aging can’t occur faster in young forests after a disturbance than in the “gradual ageing” forests.

Response: We appreciate the reviewer’s feedback and recognise the need to clarify the terminology used in describing forest age dynamics. We confirm that all forests age at the same rate (i.e., time progresses uniformly across all pixels), and our use of "gradually ageing" was not intended to suggest differences in the speed of ageing.

Revisions made: To avoid confusion, we have revised the terminology and now refer to “undisturbed ageing forests” rather than “gradually ageing forests.” This term more accurately reflects that these forests did not experience stand-replacing disturbances between 2010 and 2020 and, as a result, show an age difference of precisely 10 years during this period. Similarly, we have adjusted the wording around the transition of stand-replaced forests to clarify that while biomass accumulation and structural growth may be rapid in young forests, ageing remains constant and independent of growth rate. These refinements ensure that the distinction between ageing and growth is clearly articulated. In addition, we have improved the definition of these two processes in Table S1 (now Table 2 in the updated manuscript).

- 4. Comment:** Growth rates and ageing rates seem to be conflated here, or, if this is not the case, then the distinction between the two is not well explained by the authors. Growth (an increase in some forest attribute, e.g. biomass) can vary based on age, as noted throughout the paper and in table S1. Growth seems to be confused with a change in age, or “ageing”. The rate of ageing of forests, regardless if its immediately after a disturbance, or in the ‘gradually ageing’ class that did not experience a disturbance, does not change, because time passes at a constant rate. This is complicated, however, because in GAMI, age is modeled from biomass. While the GAMI product itself isn’t the main focus of this work, I have concerns with the premise of modeling forest age from outputs from a modeled biomass surface. (I’ve expressed these concerns above).

Response: We appreciate the reviewer’s careful consideration of the distinction between growth and ageing rates and acknowledge the need to clarify these concepts in the manuscript. We agree that ageing occurs constantly as time progresses, whereas growth—defined as increased biomass or structural attributes—varies with age and environmental factors. In response to this comment, we revised the manuscript to distinguish these concepts clearly. Additionally, we recognise the concern regarding the

premise of modelling forest age from a modelled biomass surface. However, age in GAMI is not solely derived from biomass; instead, it is estimated as a function of biomass, canopy height, and climate variables using an extensive forest inventory age dataset. This multivariate approach accounts for structural and ecological differences across climate gradients, allowing us to approximate the complex relationship between forest age and forest structure rather than directly equating biomass with age. While we acknowledge the limitations inherent in this approach, we emphasise that GAMI provides a data-driven framework to infer large-scale forest age distributions where direct age measurements are unavailable. In response to the reviewer’s concerns, we have revised the methodology section to clarify these assumptions, discuss the implications of using modelled biomass to infer forest age, and explicitly address potential sources of bias and uncertainty. Additionally, we include Fig. S15 in the supplementary material, illustrating how biomass and age covary across different hydro-climatic gradients.

Fig. S15. Relationship between biomass and forest age across different hydro-climate zones, derived from the GAMI dataset and ESA-CCI biomass (v4) data for 2020. The

figure compares biomass accumulation patterns with increasing forest age, highlighting broad ecological trends. While the GAMI dataset inherently includes biomass information from the ESA-CCI products (see Methods), this comparison confirms consistency of observed growth patterns.

Revisions made:

- We have adapted the method section **Estimating the changes in forest age distribution for 2010-2020**: *“Estimating changes in forest age distribution between 2010 and 2020 involved several steps using the 100m GAMIV2.0 product. First, we aggregated the data to a one-degree pixel resolution for both 2010 and 2020 using an average resampling method. The resulting datasets provided estimates of the mean forest age at a one-degree scale for both years (Fig. 1a). In parallel, we calculated forest age differences at GAMIV2.0’s native resolution (100m). This difference map was then resampled to a one-degree resolution using the same averaging method to maintain spatial consistency (Fig. 1c). Additionally, we quantified the total area occupied by young (0-20 years old), maturing (21-80 years old), mature (81-200 years old), and old (>200 years old) forests in both 2010 and 2020 (Fig. 1b). This analysis was performed independently for each of the 20 forest age maps. In this study, it is important to distinguish forest ageing, which progresses at a constant rate as time passes, from forest growth, representing changes in biomass accumulation over time. Growth rates vary depending on stand dynamics, disturbance history, and environmental conditions, whereas ageing itself remains uniform. However, because GAMI estimates forest age as a function of biomass, canopy height, and climate variables, variations in modelled age primarily reflect inferred structural and ecological differences rather than the simple passage of time. This distinction is crucial for correctly interpreting trends in forest age distribution and their implications for carbon dynamics.”* (L 596-613).
 - Added Fig S15 that shows the relationship between biomass and forest age across different hydro-climate zones.
 - The description of the GAMI method has been improved, as discussed in the response to a previous comment. (L523-574)
5. **Comment:** The methods section “Partitioning stand-replaced and gradually ageing forests” does not make sense as explained, or at least context is missing. An age difference of 10 years means no disturbance, which is straightforward. If there was a stand replacing disturbance between 2010 and 2020, the age difference would be then be negative, as the 2020 age would be less than the 2010 age. So, if there are other possibilities outside of 1) a 10 year age difference between 2010 and 2020 or 2) a

negative difference, indicating the 2020 age is less than the 2010 age due to disturbance, then it is not clear or well explained how this could be possible. If not, this section could be more intuitively explained. Further, for the 1° resampling, a mean difference of less than 10 years really represents two things: the fraction of pixels that were disturbed than thus the 2020 age is less than the 2010 age, and the time of the disturbance and thus how much regrowth was able to occur between the time of disturbance and 2020. If this interpretation is correct, it should be explicitly explained in the text. If not, the section needs to be clarified.

Response: We appreciate the reviewer’s detailed feedback and recognise the need to clarify how stand-replaced and gradually ageing forests are classified.

Revisions made: We have revised the methods section to explicitly state that an age difference of precisely 10 years corresponds to undisturbed ageing forests, while an age difference of less than 10 years (including negative values) indicates a stand-replacing disturbance between 2010 and 2020. Negative values occur when disturbances happen later in the decade, resulting in limited regrowth. Additionally, we have clarified the interpretation of the one-degree resampling process, where a mean age difference of less than 10 years reflects both:

- The fraction of pixels experienced stand-replacing disturbances (lowering the mean age difference below 10 years).
- The timing of disturbances and subsequent regrowth (earlier disturbances allow for more regrowth, leading to higher mean differences).

These clarifications ensure that the methodology is more intuitive and explains possible scenarios explicitly. The revised text can be found in the updated methods section (L614-636).

6. **Comment:** South America tropical is listed as a region with “more prevalent” stand replaced forests and gradually aging forests, which seems at least counter intuitive, and potentially incorrect, if these two forests class fractions sum to 1.

Response: We appreciate the reviewer’s observation and recognise the need to clarify our statement. The key point is that our results are expressed in total area rather than relative fractions. While the sum of stand-replaced and undisturbed ageing forest fractions within a given region equals 1, this does not mean that the absolute area of both classes cannot be high compared to other regions. In tropical South America , the total forest extent is extensive, meaning that both stand-replaced forests and naturally/undisturbed ageing forests occupy substantial absolute areas, even if their fractional composition remains constrained within the region.

Revisions made: To avoid confusion, we have revised the wording in the results section to ensure that the comparison with other regions is interpreted in terms of absolute area rather than fraction: “*Due to its vast forest extent, South America Tropical contains*

substantial absolute areas of both stand-replaced and undisturbed ageing forests, making both classes particularly prevalent in the region.” (L 208-210).

Reviewer #3

General response:

We appreciate the reviewer’s careful assessment of our manuscript and their recognition of the study’s contribution to understanding the carbon cycle in the context of land-use change. We acknowledge that both forest age and net ecosystem exchange (NEE) are inherently challenging to quantify, and we appreciate the reviewer’s concern regarding how uncertainties in these estimates might influence our conclusions. In the responses below, we provide further details on the methods used to derive forest age and NEE, discuss how uncertainties are accounted for, and clarify the robustness of our findings. We also ensure that the limitations of our approach are more explicitly addressed in the manuscript. We hope these revisions strengthen our analysis and provide greater transparency in interpreting the results.

Major Comments

- 1. Comment:** The first concern is the uncertainty associated with both forest age and CO₂ flux estimates. Forest age is a complex term, and its definition depends on the context. In this manuscript, I believe the definition of age in this study appears to be related to carbon, so providing a clear and precise definition is still crucial. However, accurately estimating forest age to differentiate between age classes with a 20-year interval is challenging due to the substantial uncertainties in biomass data used for age estimation. While this uncertainty does not invalidate the analysis, it is essential to include age validation results and account for this uncertainty when assessing the relationship between age and NEE (e.g., Fig 3). Additionally, the top-down inversion estimates themselves vary significantly, and uncertainty is a very large grid scale scale. This study incorporates nine estimates in an effort to address these uncertainties; however, the results seem highly dependent on the subset of estimates used. For instance, in Fig 3a, one inversion product shows a larger positive slope compared to the others, which heavily influences the overall trend represented by the dark line and is central to the paper's conclusion. Therefore, I recommend testing the robustness of the results by excluding this outlier to ensure the conclusions are not overly dependent on a single product.

Response: We appreciate the reviewer’s thoughtful feedback regarding the uncertainties associated with forest age estimation and CO₂ flux estimates from atmospheric inversions. We fully acknowledge that forest age is a complex variable with different definitions depending on the context. In this study, forest age is estimated as a function of

biomass, canopy height, and climate variables rather than being solely derived from biomass. This allows our approach to capture structural and ecological differences across climate gradients rather than relying on a direct biomass-to-age conversion. However, we recognise that biomass uncertainties inherently propagate into age estimates, which can impact age class differentiation. Currently, full validation of our forest age product is not feasible, as no consistent, independent dataset exists that would allow direct validation of such an EO-derived product. A rigorous validation would require human-interpreted data or National Forest Inventory (NFI) datasets with precise geographic coordinates, which we currently do not have access to. The best we can do at this stage is to report our cross-validation results (Fig. S13) and evaluate whether the forest age data exhibit ecologically sound patterns, such as the expected relationship between biomass and forest age (e.g., Fig. S15). Additionally, a comparison with existing forest age products was conducted in the original dataset publication (<https://essd.copernicus.org/articles/13/4881/2021/>), revealing some discrepancies. These differences highlight the inherent challenges in mapping forest age globally, emphasising the uncertainties and complexities associated with remote sensing-based forest age estimates.

Fig. S13. Cross-validated results of the old-forest vs. non-old-forest classification (a) and comparison of predicted vs. observed forest age estimates from the regression model (b). The quantile-quantile plot (c) and the model residuals across age classes (d) are also shown.

Fig. S15. Relationship between biomass and forest age across different hydro-climate zones, derived from the GAMI dataset and ESA-CCI biomass (v4) data for 2020. The figure compares biomass accumulation patterns with increasing forest age, highlighting broad ecological trends. While the GAMI dataset inherently includes biomass information from the ESA-CCI products (see Methods), this comparison confirms consistency of observed growth patterns.

Yet, we used an ensemble of 20 forest age members that reflect uncertainties from the biomass input data to account for uncertainties in forest age estimation (see method section in L502-520 and L563-572). This ensemble approach ensures that all analyses incorporate a range of plausible age estimates. Additionally, by binning the data into coarse forest age classes (young, maturing, mature, old-growth), we substantially mitigate uncertainties in the GAMI dataset by reducing the sensitivity to small variations in individual age estimates.

Furthermore, we acknowledge the reviewer’s concern regarding uncertainties in top-down CO₂ flux estimates from atmospheric inversions. We recognise that individual

inversion models exhibit significant variability at coarse spatial scales, which can influence the inferred relationships. In our initial approach, we relied on an ensemble estimate of nine inversion models, which could lead to averaging effects that obscure spatial variations within individual models. To improve the robustness of our analysis, we have now conducted a systematic Jackknife resampling approach, where we systematically exclude individual NEE inversion models one at a time. This approach allows us to quantify how dependent the observed trends are on specific inversion products and ensures that any single model does not dominate our findings. Importantly, this new analysis revealed that the relationship between the fraction of old forests replaced by young stands and NEE trends is consistently present across multiple NEE models. However, we find no consistent or significant relationship for younger age classes, reinforcing that old forests replaced by young stands are the key drivers of carbon flux changes in our study.

Additionally, we acknowledge the scale mismatch between fine-scale stand-replacing disturbances and the coarse resolution of atmospheric inversions. To address this, we have implemented a spatial window approach that aggregates both stand-replacement and NEE changes within $2^{\circ}\times 2^{\circ}$, $5^{\circ}\times 5^{\circ}$ and $10^{\circ}\times 10^{\circ}$ spatial windows rather than relying on pixel-based binning. This ensures that our analysis preserves spatial coherence between stand-replacement estimates and NEE fluxes.

By integrating cross-validation results, age ensemble members, Jackknife resampling, and spatial window aggregation, we ensure our conclusions are not overly dependent on a single dataset or modelling choice. This refined approach strengthens confidence in our findings and highlights the robust and spatially consistent relationship between the fraction of old forests replaced by young stands and NEE trends. We refer to the response to the first comment of reviewer on for more details.

- 2. Comment:** My second major concern is the scale mismatch between the 100 m spatial resolution of the forest age GAMI v2.0 data and the 1-degree resolution of the top-down CO₂ fluxes. The method for identifying CO₂ flux by age class (e.g., Figure 3b) is not sufficiently explained. Since each 1-degree grid likely contains multiple age classes, it is unclear how the flux is partitioned among these classes. There is no clear explanation and evaluation of this process given the significant spatial heterogeneity of forest age at fine scales, needs to be addressed.

Response: We acknowledge the referee's concern regarding the scale mismatch between the 100 m forest age data (GAMIV2.0) and the 1-degree resolution of the CO₂ fluxes. Given the spatial heterogeneity of forest age at fine scales, we have removed Figure 3b from the manuscript to avoid potential misinterpretations. The revised analysis now primarily focuses on the relationship between the fraction of old forests replaced by

young forests and NEE trends, ensuring a clearer and more robust assessment of how forest age transitions influence carbon fluxes.

- 3. Comment:** The third concern is the inherent influence of climate variability and other factors on the NEE estimates used to represent ~2010 and ~2020. It is unclear how the study isolates the effects of climate variability from ageing on NEE. This becomes even more problematic when considering projections for 2050, as major drivers such as CO₂ fertilization and warming are not explicitly considered in the projection. How these drivers affect the study's conclusions related to forest conservation. Is the estimate optimistic or conservative?

Response: We acknowledge that climate variability, CO₂ fertilisation, and other environmental factors influence NEE estimates, and we do not attempt to isolate these effects from forest stand replacement dynamics. Instead, our analysis is designed to assess whether there is a covariation between stand replacement and NEE trends rather than attributing causality to a single driver. The purpose of this study is not to quantify the direct impact of forest ageing on NEE in isolation but rather to investigate whether regions with higher stand-replacement rates exhibit distinct NEE trends compared to areas with stable, gradually ageing forests.

Regarding projections for 2050, we agree that future climate-driven changes (e.g., CO₂ fertilisation, warming, and altered precipitation patterns) will strongly influence forest carbon dynamics. However, our approach does not explicitly model these effects. Instead, our projection assumes that past forest stand-replacement patterns continue at similar rates, providing a baseline for understanding how continued forest transitions could influence future carbon fluxes. Given the substantial uncertainties associated with climate-driven changes in forest carbon uptake, our estimates should indicate the potential impact of forest age dynamics rather than a precise forecast of future NEE. Whether these estimates are optimistic or conservative depends on how future climate conditions modify the capacity of regrowing forests to act as carbon sinks. For example, CO₂ fertilisation could enhance growth rates, making our estimates conservative, while increased drought frequency and fire disturbance could limit carbon uptake, making our estimates optimistic. Further work is needed to fully disentangle the interactions between stand-replacement, forest ageing, and climate-driven processes.

Revisions made: We have explicitly discussed these limitations in the revised discussion section to acknowledge these uncertainties. We highlight how our results should be interpreted in the broader context of multiple drivers influencing NEE trends. Here the addition to the discussion: *'While strengthening the carbon sink in regions with high stand-replacement rates can be partly attributed to the transition from old to younger forests, additional factors such as CO₂ fertilisation, temperature changes, and climate*

variability may also influence NEE trends. Increased atmospheric CO₂ can enhance forest productivity, while warming and drought may limit carbon uptake, particularly in specific ecosystems. Although these broader influences are acknowledged, our analysis focuses explicitly on the relationship between stand replacement and NEE trends without attempting to disentangle the effects of climate variability and CO₂ fertilisation.’ (L 342-349)

4. **Comment:** The fourth concern relates to the uncertainty in the projections for the two hypothetical scenarios. The manuscript assumes that the age distribution remains unchanged under the BAU scenario and, based on this, concludes that biomass is nearly unchanged. However, this assumption is flawed. The analysis presented between lines 140-220 for the period 2011-2020 clearly shows that forest age shifts under current practices, which effectively represent BAU conditions. Additionally, forests continue to accumulate biomass within each age class, meaning the method used in this study likely underestimates biomass gain. A simple way to test the validity of the method is to check whether it can accurately reproduce AGB changes using ESA Biomass data. Including this validation, or at least acknowledging the potential limitations, would provide clearer insight into the reliability of the study's results.

Response: We acknowledge the referee's concern regarding the assumptions in the BAU scenario and the potential underestimation of biomass accumulation within age classes. We agree that explicitly validating our projections against observed ESA-CCI biomass changes would provide valuable insight into the method's reliability. However, due to time constraints, we did not perform this validation. Instead, we have clearly acknowledged this methodological limitation in the revised manuscript (L670-707).

- a. Clarification of the BAU Assumption:

The BAU scenario does not assume that forests will not age or that biomass accumulation will cease. Instead, it assumes that the proportional distribution of forest age classes remains similar to that observed between 2010 and 2020. This approach allows us to estimate future carbon stocks while continuing current disturbance and stand-replacement rates rather than simulating precise age transitions for individual pixels.

- b. Why the BAU Scenario Reflects Current Practices:

The 2010-2020 period provides an empirical basis for understanding how forest age distributions evolve under present-day conditions. Our methodology ensures that forests continue to age within their respective age classes while experiencing stand-replacement dynamics, maintaining a consistent age-class distribution

pattern over-time. This approach reflects ongoing harvesting and disturbance trends rather than assuming static forest conditions.

c. Potential Underestimation of Biomass Accumulation:

The referee correctly points out that forests within each age class will continue accumulating biomass. Our approach accounts for this by using observed biomass values for each age class in 2020 and applying them to 2050 projections. However, we acknowledge that this method does not explicitly model age-class transitions at finer temporal resolutions, which may lead to conservative estimates of carbon accumulation. While this simplification introduces some uncertainty, it provides a first-order estimate of how carbon stocks evolve under different disturbance scenarios.

Revisions made: We have revised the discussion section to clearly state that our approach does not explicitly simulate forest succession processes beyond age-class transitions. Instead, it provides a first-order estimate of how forest age dynamics under different scenarios could influence future carbon storage. Here is the added text to the discussion: *'While these scenarios do not explicitly model the effects of climate-driven factors (e.g., CO₂ fertilisation or altered disturbance regimes), they provide a first-order estimate of how carbon stocks evolve under different forest age dynamics.'* (L377-380)

Minor Comments

5. **Comment:** In fig 1, the negative sign is missing in ">30". Additionally, it would be helpful to clarify what the negative values in fig 1c imply. Does this imply that forests are getting younger due to compositional changes, with a higher proportion of younger forests emerging?

Response: Thank you for pointing this out. We have corrected the missing negative sign in ">30" in Figure 1. Regarding the negative values in Figure 1c, we have now clarified their meaning in the figure caption (L454-464). Negative values indicate a net shift towards younger forests within a given region, meaning that the proportion of younger classes has increased relative to older ones. This could result from higher rates of stand replacement, increased disturbances (e.g., harvesting, wildfires, or insect outbreaks), or shifts in forest composition favouring younger stands.

6. **Comment:** In fig 3a, how robust is the trend shown by the black line, considering the outlier with a steep positive slope? Since this outlier seems to have a significant influence on the overall trend, testing the sensitivity of the black line by excluding this outlier would provide greater confidence in the robustness of the conclusions. This sensitivity

test is crucial to ensure that the results are not overly dependent on a single inversion product.

Response: We acknowledge the reviewer's concern regarding the potential influence of individual inversion products on the overall trend shown in Fig. 3a (now Fig. 3c in revised manuscript). In our revised analysis, we have adopted a more rigorous approach to assessing the robustness of the observed relationship between stand replacement and NEE changes (see above comment and response to referee 1). Specifically, we now systematically evaluate the relationship across all individual inversion members rather than relying solely on the ensemble mean. To ensure that a single inversion product does not overly influence the observed trend, we have applied a Jackknife resampling approach, systematically excluding one inversion product at a time. This analysis confirms that the negative correlation between stand replacement in old forests and NEE trends remains consistent across multiple inversions, reinforcing the robustness of our findings. The revised figure now includes the full range of regression lines for individual inversion products in the background to illustrate their variability transparently. Additionally, we have refined our spatial aggregation strategy by extracting data within larger spatial windows rather than relying on previous binning methods. This change improves the spatial coherence of the analysis and ensures that the inferred relationships are less susceptible to noise from individual grid cells or single outliers. These modifications strengthen confidence in our results, demonstrating that the observed trend is a systematic signal rather than an artefact driven by a single inversion model. A more detailed answer is provided in the responses given to the first reviewer.

- 7. Comment:** For fig 3b, it is unclear whether the age class is based on 2011 or 2020. If it is based on 2011, why isn't there a positive NEE (carbon source), which is expected due to legacy emissions from previous disturbances? This also ties back to my major concern regarding the method used to derive NEE by age class. How was the NEE by age classes derived? The CO₂ flux at one-deg contains multiple age classes.

Response: We appreciate the referee's concern regarding the temporal reference for forest age and NEE fluxes. We agree that deriving age-class-specific NEE at a coarse 1-degree resolution, given the fine-scale spatial heterogeneity of forest age classes, could introduce artefacts and misinterpretation. After careful reconsideration, we have removed Figure 3b from the manuscript to avoid confusion arising from potential legacy emissions or spatial scale mismatches. Instead, our updated analysis (Fig. 3c) explicitly assesses the relationship between the fraction of old forests replaced by young stands and the overall regional trend in NEE using weighted averages at the one-degree resolution. In this revised approach, we calculate weighted-average forest age class fractions within each inversion grid cell and correlate these fractions with their corresponding NEE trends.

This methodology avoids the problematic assumption of direct one-to-one relationships between specific age classes and NEE at the coarse inversion resolution.

8. Comment: In P4, L55, it is unclear how Figure 3b supports the statement being made.

Response: We appreciate the referee's feedback regarding the link between Figure 3b and the statement about factors influencing forest age distribution and carbon sequestration capacity. Figure 3b illustrated the relationship between net CO₂ fluxes and forest age classes, showing how different age groups contribute to carbon uptake or release. Yet, Figure 3b has now been removed from the manuscript to avoid potential misinterpretations due to spatial-scale mismatches. The sentence has been revised.

Revisions Made:

- The sentence was replaced by: *'These factors, in turn, affect the forests' capacity to sequester carbon'* (L 46-47)

9. Comment: In P9, L273-274, the statement “younger forests typically have lower net CO₂ uptake because of emissions related to stand-replacement (Fig. 3b)” is unclear. Does this refer to legacy emissions (CO₂ losses from delayed tree mortality, deadwood decomposition, and soil respiration) from themselves, or to immediate emissions from the aboveground biomass of forested areas surrounding younger forests?

Response: This sentence has been replaced with the following to better align with the literature: *"Despite the initial loss of AGC following stand replacement, young forests often transition quickly to net carbon sinks."* (L299-300)

Additionally, we clarify that this statement refers specifically to CO₂ losses from delayed tree mortality, deadwood decomposition, and soil respiration—commonly known as legacy emissions following stand replacement. These emissions originate from the previously existing forest stand rather than from surrounding areas. This distinction has been made explicit in the revised manuscript to improve clarity.

10. Comment: L9, L277, the data source of unmanaged and managed layers is missing.

Response: As mentioned in our response to other comments, the analysis distinguishing managed and unmanaged forests has been removed from the manuscript. Consequently, there is no longer a need to reference a data source for these layers.

11. Comment: In P9, L284-285, the statement, “The natural regeneration and diverse age structure in these forests may help absorb the impacts of such disturbances without

significantly affecting net CO₂ fluxes,” raises a question about the role of forest tree cover. To what extent is it due to the high tree cover in unmanaged forests? The classification of unmanaged versus managed forests seems to correlate with tree cover. This suggests that, at the spatial scale of Figure 3a and S8, grids classified as unmanaged forests may simply have higher tree cover, which leads to greater carbon uptake, thereby offsetting losses from stand-replacement and making the slope insignificant. This implies that the result may not necessarily be due to a diverse age structure.

Response: This analysis has been removed from the revised manuscript. As a result, we no longer distinguish between managed and unmanaged forests in relation to net CO₂ fluxes. Consequently, discussions have been omitted regarding the role of natural regeneration, diverse age structures, or potential correlations with tree cover at the grid scale. We appreciate the referee’s insightful comment, and while this specific analysis is no longer included, future studies could explore the interaction between forest management, tree cover, and carbon uptake in more detail.

12. Comment: In P11, L374-383, I am not sure how the carbon stocks partition among age classes from L370-383 is related to the former part of this paragraph. And also, providing stock changes over the interval is also very important.

Response: We have revised the text to clearly distinguish between total carbon stocks and carbon losses associated with stand replacement (L257-292). The updated version explicitly separates the discussion of AGC stocks from the processes driving biomass reductions, ensuring a more structured and coherent presentation. Additionally, we have improved the transition between these topics to avoid conflating total carbon storage with stand-replacement-driven losses. Given the limitations of the ESA-CCI biomass product in capturing slow-in processes, such as the gradual growth of mature and old forests, we do not assess carbon accumulation within age classes. Instead, our analysis focuses on the spatial distribution of carbon stock losses resulting from stand replacement while recognising that undisturbed forests continue storing carbon over time.

13. Comment: In P11, L387, how exactly was the BAU scenario defined? Assuming the current rate of deforestation continues? Providing a clearer definition or outlining the key assumptions used to establish the BAU scenario is very necessary.

Response: The methodology section explicitly defines the BAU scenario (L670-707). It assumes that forest management practices and disturbance rates observed between 2010 and 2020 remain unchanged over the next 30 years. This scenario does not take a fixed stand-replacement rate but maintains the proportional distribution of forest age classes observed during this period. In practice, the BAU scenario projects future carbon stocks

by assuming that the total area of each forest age class in 2050 remains the same as in 2020 while the forests within each class continue to age and accumulate biomass. The total carbon stock is then estimated by multiplying the projected area of each age class in 2050 by its respective carbon stock estimate from 2020, normalised by the total area of that age class in 2020.

14. Comment: In P18, L511, it is unclear what "mean" refers to. Mean over what (time or space)? Additionally, how is the standard deviation of aboveground biomass (AGB) derived? Is it calculated over space, and if so, why? Does the spread in fig 3c reflect both spatial variation across pixels within the group as well as AGB variation due to perturbations?

Response: We have clarified the description in the revised manuscript to ensure greater transparency regarding how mean and standard deviation values are used in our analysis. The ESA-CCI biomass v4 dataset provides per-pixel estimates of aboveground biomass (AGB) along with corresponding standard deviation values, representing uncertainty at the pixel level rather than over time or space. To account for this uncertainty, we generated 20 realisations of AGB by introducing controlled perturbations to the mean AGB values using the provided standard deviation as a scaling factor. This approach allowed us to create an ensemble of biomass estimates, capturing a plausible range of biomass variations due to measurement uncertainties. Regarding Figure 3c (now Figure 3a in the revised manuscript), we report the median estimates across these 20 biomass maps rather than a single realisation. This ensures that our analysis is robust to uncertainties in the ESA-CCI biomass dataset. The spread shown in Figure 3c reflects the spatial variation within a given age class rather than uncertainty across ensemble members. This means that the variability in AGB observed within each age class is due to heterogeneity in forest structure and biomass accumulation across different regions. To further improve clarity, we have revised the relevant sections in the Methods and Figure 3c (now Figure 3a in the revised manuscript) caption to explicitly state that the spread in the figure represents the range of AGB values derived from our ensemble approach rather than purely spatial variation.

Revisions made:

- The Methods section has been updated to explicitly state that the ESA-CCI biomass product provides per-pixel mean and standard deviation values and clarify that these uncertainties were incorporated into our ensemble-based analysis.
- The caption of Figure 3c (now Figure 3a) has been revised to indicate that the median values from the 20 biomass realisations are shown and that the spread represents the spatial variation within a given age class.

15. Comment: In P19, L537, it is unclear how the gain layer was used in the age estimation process. As I know that this data does not specify when the gain occurred, so how was this information utilized? Was the gain layer just used to provide a gridded fraction, or for another purpose?

Response: The gain layer was not used to determine the exact timing of regrowth, as it does not provide annual information on when the gain occurred. Instead, we used it as a filtering criterion to distinguish between areas where stand-replacement disturbances led to forest regrowth versus areas where forest loss resulted in a transition to non-forest. Specifically, in cases where the loss layer indicated a stand-replacement event, we checked for the presence of the gain layer. If gain was detected in the corresponding pixel, we assumed the forest regenerated after disturbance and assigned it a younger age based on the estimated time since disturbance. However, if no gain was present in a pixel with forest loss, we interpreted this as a permanent forest-to-non-forest transition, meaning that the area did not recover as a forest and was not considered in the ageing process. By incorporating the gain layer this way, we ensured that our age reconstruction only accounted for regrowing forests, thereby avoiding misclassifying deforested areas as young forests. This approach aligns with previous studies that use gained information to validate post-disturbance recovery processes while acknowledging its limitations in pinpointing the exact year of regrowth.

16. Comment: In P19, L544, the method for estimating “forest age from Landsat-based estimates” is unclear. Are you referring to the "time since disturbance" layer? If so, it's important to note that this measure does not necessarily reflect the year the forest began to recover.

Response: As requested by Referee 2, we have updated the Methods section to clarify how forest age was estimated from Landsat-based disturbance data (L523-574). Specifically, we confirm that our approach relies on the time since disturbance layer, which estimates the time elapsed since the last major disturbance event. We acknowledge that this does not always precisely reflect the exact year in which forest recovery began, as some disturbances (e.g., partial harvesting, fires) may lead to delayed regeneration. However, because we bin our data into broad age classes (e.g., 0-20 years), we are confident that the pixels categorised as young forests belong to this age class, even if there is some variability in the exact timing of regrowth. This classification approach ensures that our analysis captures meaningful trends in stand-replacement dynamics while accounting for uncertainties in forest recovery timing. We hope these clarifications address the reviewer’s concerns and improve the transparency of our methodology.

17. Comment: In P19, L565, it is unclear how the uncertainty data were incorporated into the analysis, particularly in Fig 3a. Could you clarify how these uncertainties were factored into the results, and how they influenced the trends shown in the figure?

Response: We have clarified how the uncertainty from the 20 forest age ensemble members was incorporated into the analysis, particularly in Figure 3a. The 20 age ensemble members represent different realisations of forest age dynamics, capturing the uncertainty in age estimates due to variations in input data and model assumptions. To ensure our results are robust to these uncertainties, all analyses related to forest age and its relationship with CO₂ fluxes (Fig. 3a) were conducted independently for each ensemble member. Figure 3a calculated the relationship between stand-replacement fraction and net CO₂ flux trends separately for each age and NEE member, producing 20 x 9 individual regression slopes. The black regression line in Figure 3a (now Fig. 3c in the revised manuscript) represents the median trend across all 180 members.

18. Comment: In P20, L600-615, what was the consideration for not using the Hansen forest loss and gain layers, which are closer to direct observation, and instead opting for modeled forest age? How consistent are these two approaches? Are there areas where the Hansen data do not indicate loss or gain, but the model shows an age change of less than 10 years, resulting in classification as stand-replaced forests?

Response: Our choice to use a modelled forest age product rather than the Hansen forest loss and gain layers was based on the key consideration that for our approach there was the need to partition stand-replaced forests by different age classes—something that the Hansen dataset cannot provide, as it only tracks forest cover change since 2000 and does not contain granular information on pre-2000 forest age structure. Moreover, no globally consistent long-term forest age dataset exists, making our approach essential for understanding age-class dynamics over extended periods. Our methodology explicitly distinguished between stand-replaced forests and gradually ageing forests across different age classes. This was done by computing the difference in forest age between 2010 and 2020, identifying pixels where age increased by less than 10 years, which we classified as stand-replaced forests. In contrast, pixels aged by precisely 10 years were classified as gradually ageing forests. This allowed us to create two products: (i) an age difference map quantifying the magnitude of stand-replacement and (ii) a binary classification identifying stand-replaced vs. gradually ageing forests. Our approach further stratifies stand-replaced and gradually ageing forests into four age classes (0-20, 21-80, 81-200, >200 years old) using the 2010 GAMIV2.0 forest age product. This level of stratification was essential for assessing how stand replacement influences NEE trends across forest age classes. While useful for detecting recent loss and gain, the Hansen dataset does not

provide information on pre-2000 forest age and, therefore, does not allow us to track stand-replacement dynamics at an age-class level. We hope this clarification addresses the reviewer's concerns.

19. Comment: In P21, L630-640, the method for calculating NEE by age class is not clearly explained. How is the "ratio between the fraction of stand-replaced and gradually aging forests" related to the age classes shown in fig 3b?

Response: We appreciate the referee's comment and acknowledge the need for further clarification. As previously explained, our analysis of NEE and the fraction of stand-replacement has been revised to ensure a more robust interpretation of the data. Instead of relying on a ratio between the fraction of stand-replaced and gradually ageing forests, we now assess NEE trends by evaluating the relationship between stand-replacement extent and changes in net CO₂ fluxes within spatial windows.

General update:

In the revised manuscript, we have applied a consistent forest fraction mask across all analyses, excluding one-degree grid cells with less than 20% forest cover. This adjustment reduces noise from sparsely forested regions. Consequently, figures and associated statistics have been updated, though the study's main findings remain unchanged. Several key supplementary figures (i.e., Fig. S1, S5, S6, S15, S16) have been promoted to Extended Data to enhance the visibility of essential background information and improve access to core results. Fig. S7 is now part of the main text and has been designated as Fig. 3. We have also substantially reduced the main text to comply as much as possible with the *Nature Ecology & Evolution* word limit (~3,500 words), with the current word count at approximately 4,600 words. The final section on the prognostic scenario has been removed as part of these revisions. Figure 4 is now included as an Extended Data figure, and the outcomes of the prognostic exercise are integrated in the concluding remarks of the paper.

Reviewer #1

General response:

We thank the reviewer for acknowledging the substantial revisions and improved clarity and quality.

Major Comments

1. **Comment:** My main concerns are mostly well addressed, and your manuscript is much improved. However, the new results (Figure 3) regarding to NEE based on atmospheric inversion models become quite different from previous results in the original manuscript. For example, NEE from most models now decreases (meaning increasing sinks) with stand replacement extent (Figure 3c) instead of increasing in the previous result. You provided an explanation for this new and different result, which is reasonable as younger stands could indeed grow faster than the replaced old stands, although AGB of younger stands is smaller. However, this large change in the results due to different ways in manipulating the same data causes concern on the reliability of the inversion results for the purpose of your analysis. A more problematic issue is the set of results in Figure 3d, which shows very different responses of NEE to stand replacement extent for different age cohorts. In particular, the responses between maturing (decrease) and mature (increase) age cohorts to stand replacement is opposite, while the old forests show a decreasing trend. Logically, we would expect the mature cohort would be a case between maturing and old forest cohorts. Although you provided a long explanation to these results in a speculative tone (lines 332-349), I feel the explanation is faint and unconvincing. While I fully appreciate your great effort in pooling the data together and producing these new outputs, I feel that parts of the results are too uncertain to publish. I would suggest to scrub all results from the atmospheric inversion models, but it would water down the content of the paper significantly. As a compromise, Figure 3c may be retained if you have some confidence on the results, but the uncertainty associated with NEE results should be emphasized.

Response: We fully acknowledge the reviewer's concerns regarding the inversion-based NEE trends, particularly the disaggregated analysis by age class shown in the revised Figure 3. In response, we have removed panel 3d from the main manuscript and now present the age-class-specific correlations only in the Supplement (Fig. S16), accompanied by explicit caveats about interpretability. In the revised main text, we focus on the more robust and interpretable relationship

shown in Figure 3c—namely, the negative correlation between the extent of old forest replacement and the 10-year trend in net CO₂ uptake. We now clearly state that the spatial resolution of atmospheric inversion models (~1°) limits the capacity to disentangle NEE trends across all four age classes. This constraint is particularly pronounced in regions with mixed-age forests and intensive management, where the inversion signal reflects integrated fluxes. In contrast, more interpretable signals may emerge in tropical regions where large contiguous areas of old-growth forest have been replaced by young stands, resulting in more homogeneous age distributions within grid cells. We have substantially reduced the speculative interpretation previously included in Lines 332–349 and now present age-class disaggregation only as an exploratory analysis. Furthermore, we have revised the discussion to emphasise the ensemble spread and uncertainties in the inversion products. For transparency, we retained all nine inversion models from GCB2023 in our analysis, even though we know that some (e.g., CT-NOAA) show weaker land sinks and stronger ocean sinks than the ensemble average. Others (e.g., MIROC) were excluded from the final GCP2023 ensemble due to quality control issues. The manuscript acknowledges these differences to help contextualise the spread observed in Figure 3c. To further strengthen our interpretation, we note that we observe similar relationships between old forest replacement and NEE trends when using the latest Global Carbon Project (GCP 2024) ensemble (Fig. R1c). This convergence across versions adds confidence in the robustness of the reported patterns, despite the known limitations of inversion-based CO₂ flux estimates. We believe this revised framing offers a more cautious and transparent interpretation of the inversion-based NEE results while maintaining the integrity and value of the dataset. We hope the reviewer agrees this is a constructive way to address the limitations raised.

Revisions Made:

- Figure 3d was removed from the manuscript (now Supplementary Figure S20).
- Discussion in Lines 332–349 shortened and rewritten to focus on Fig. 3c and to acknowledge resolution limits and ensemble uncertainties: *“It is also essential to recognise this analysis's limitations. At the spatial resolution of current atmospheric inversion models (~1°), it is beyond our capacity to robustly disentangle NEE trends across all four age classes. This limitation is particularly pronounced in regions with mixed-age forests and intensive management, where inversion signals integrate fluxes across diverse stand conditions. While exploratory correlations by age class are presented in the Supplement (Extended Data Fig. 8a–c), these should be interpreted cautiously. More interpretable signals may emerge*

in tropical regions, where large areas of old-growth forest have been replaced by younger stands, resulting in more homogeneous age distributions within inversion grid cells. Still, robust separation of age-specific NEE responses remains challenging at this scale. Accordingly, our main interpretation focuses on the consistent NEE signal linked to old-to-young forest transitions in these areas (Extended Data Fig. 3d), which shows a consistent negative correlation with NEE trends across multiple inversion models (Fig. 4c). Still, broader factors such as CO₂ fertilisation, temperature shifts, and climate variability may also influence NEE trends. While we acknowledge these influences, our analysis isolates the relationship between stand replacement and NEE without attempting to disentangle climate-driven effects.” (L. 312–326)

Fig. R1. Above-ground carbon stocks across forest age classes, distinguishing between stand-replaced forests and undisturbed ageing forests, expressed per unit area at a one-degree pixel level. The stand-replaced categories represent the AGC stock of forests at a given age class (Young, Maturing, Mature, or Old) before stand replacement. In (b), net carbon changes for stand-replaced forests across different forest age categories are shown. In (a) and (b), the median values from the 20 biomass realisations are shown, and the spread represents the spatial variation within a given age class. In (c), the relationship between the fraction of old forests replaced by young stands (i.e., stand-replacement extent) and changes in NEE from inversions between circa 2020 (average of 2019–2021) and 2010 (average of 2009–2011) is shown. The dark solid line represents the linear regression on the ensemble estimates, while the dashed grey lines indicate the regressions for the nine individual atmospheric inversion models. To smooth the spatial distribution of net CO₂ fluxes, we applied a Gaussian filter (length = 500 km, equivalent to approximately four one-degree pixels). This smoothing technique reduces noise in the data and helps minimise the influence of local transport errors. To reduce noise and regional variability due to atmospheric transport errors, all spatial data were aggregated using area-weighted averaging over 5×5 grid cells (i.e., a coarsening scale of 5° × 5°).

Minor Comments

- 1. Comment:** L80-82: add “to” to the three objectives;
Revisions made: Thank you for your comment. We added “to” to the three objectives.
- 2. Comment:** L91: change “increase” to “increased”;
Revisions made: Thank you for your comment. We corrected the typo.
- 3. Comment:** L291: change “slightly” to “substantially”

Reviewer #2

General response:

Major Comments

- 1. Comment:** The methods section on GAMI starting on line 523 in the revised manuscript version reads much more clearly. Furthermore, fig S15 that shows non-linear, and more importantly, non-monotonic relationships between biomass and age is highly important, and in my opinion a marked improvement over existing age products at coarser resolutions. While I understand the perspective and audience of this work is primary for the global modeling and carbon dynamics community, there is a lot of skepticism regarding the validity and utility of mean forest age mapping in the ground-based forest ecology community. These results demonstrate an advance in global-scale representations of biomass as a function of age that are more in line with ground-based observations, which is highly promising. If appropriate, the authors should consider highlighting this result more than is currently done in the manuscript.

Response: We thank the reviewer for this thoughtful observation. We agree that the non-linear and non-monotonic relationships between biomass and age (Fig. S15) represent a key strength of the GAMI v2.0 dataset. In response, we now highlight this result more clearly in the main text (L. 105–106) and the method section (L. 508–515) to explicitly state that the observed relationships align with ground-based forest ecological patterns, helping bridge the gap between remote sensing-derived age products and ecological field studies.

- 2. Comment:** I would recommend the authors be explicit and precise in this area of the manuscript however (e.g bottom panels in table 2 of revised manuscript, i.e. “forests experience a mean age difference of ...” and “a uniform ten-year mean age increase ...”). When talking about the forest age transitions at 1 degree, please be explicit you’re referring to the mean age from the higher resolution data.

Response: We appreciate the reviewer’s suggestion. We have revised the relevant text in Table 2 to explicitly refer to "mean age derived from 100 m resolution data."

- 3. Comment:** The methods section starting on line 596 is much improved – the distinction between aging and growth is well explained and the authors do a good job giving context to how variations in modeled age are primarily driven by growth (structural, ecological changes). I suggest some form of the information communicated in lines 609-613 belongs in the main body of the manuscript, perhaps somewhere in the discussion of age transitions (130-160).

Response: Thank you for the suggestion. We agree that the distinction between ageing and growth is important for interpreting the forest age transitions in our study. However, due to strict word limits (our current manuscript already exceeds the 3,500-word cap), we have decided to retain the more detailed explanation in the Methods section rather than duplicating it in the main text. That section now clearly explains that modelled forest age in GAMiv2.0 reflects structural and ecological growth patterns, not only the passage of time (L 592–597).

- 4. Comment:** This is helpful explanation, but I still have questions: How were the data products referenced in lines 634-636 integrated into the analysis? It seems to me quantifying the contribution of both factors (the magnitude of age difference vs the number of pixels with an age change) would be valuable information to report and use to contextualize results. Perhaps this is what is being referenced on lines 625-627, but it isn’t clear how these products feed into the analysis or results.

Response: We thank the reviewer for this helpful observation. We have clarified in the revised Methods that the *age difference magnitude* and *binary classification* products are used to compute the extent and intensity of stand-replacement and undisturbed ageing at the one-degree scale. Specifically, the binary classification (stand-replaced vs. undisturbed) determines the fraction of forest pixels within each grid cell that underwent stand replacement or natural ageing. Meanwhile, the *age difference magnitude* is used to calculate the mean age difference per pixel, capturing the severity of disturbance. These variables are then aggregated across space (1-degree resolution) and across model realisations to generate distributions and summary statistics for each age class and region, as shown in Fig. 1c and 2a-c. We have clarified this in the revised Methods section to better link the data processing with the analyses presented in the Results.

Revisions made: We expanded the method section in L. 618-625: “*Finally, the age difference and binary classification products were resampled to a one-degree resolution using an average resampling method, enabling large-scale analysis. This process was conducted independently for each of the 20 forest age maps to account for model variability. These resampled products served two purposes: the binary classification was used to compute the fraction of forest pixels in each one-degree grid cell that experienced stand replacement or undisturbed ageing. At the same time, the age difference magnitude captured the mean severity of age reduction or gain. These metrics were then used to quantify age transitions' spatial extent and intensity (e.g., Fig. 1c, Fig. 2a-c).*”